# Cardinality Sparsity: Applications in Matrix-Matrix Multiplications and Machine Learning

**Ali Mohaddes**                                          *ali.mohaddes@uni-hamburg.de*
*Department of Mathematics*
*University of Hamburg*

**Johannes Lederer**                                 *johannes.lederer@uni-hamburg.de*
*Department of Mathematics*
*University of Hamburg*

**Reviewed on OpenReview:** *https://openreview.net/forum?id=zoSRSpGu9C*

## Abstract

High-dimensional data has become ubiquitous across the sciences but presents computational and statistical challenges. A common approach to addressing these challenges is through sparsity. In this paper, we introduce a new concept of sparsity, called cardinality sparsity. Broadly speaking, we define a tensor as sparse if it contains only a small number of unique values. We demonstrate that cardinality sparsity can improve deep learning and tensor regression both statistically and computationally. Along the way, we generalize recent statistical theories in these fields. Most importantly, we show that cardinality sparsity has a strikingly powerful application beyond high-dimensional data analysis: it can significantly speed up matrix-matrix multiplications. For instance, we demonstrate that cardinality sparsity leads to algorithms for binary-matrix multiplication that outperform state-of-the-art algorithms by a substantial margin. Additionally, another crucial aspect of this sparsity is minimizing memory usage. By executing matrix multiplication in the compressed domain, we can significantly lower the amount of memory needed to store the input data.

## 1 Introduction

Contemporary data are often large and complex. On one hand, this offers the chance for extremely accurate and detailed descriptions of processes. On the other hand, it inflicts computational obstacles and the risk of overfitting (Cybenko, 1989). These computational and statistical challenges are under much investigation in deep learning: (Cybenko, 1989), (Joshi, 2022), (Srivastava et al., 2014), (Mohaddes & Lederer, 2023), (Ying, 2019), (Zhang et al., 2016) and (Bengio et al., 2015) but also in more traditional regression settings: (Bartlett et al., 2020), (Lederer, 2021), (Eldar & Kutyniok, 2012), (Thrampoulidis et al., 2015) and (Elgohary et al., 2016).

A common remedy for these challenges is sparsity. Sparsity is usually defined as many zero-valued parameters or many groups of zero-valued parameters: (El Karoui, 2008), (Candès et al., 2006), (Hebiri & Lederer, 2020) and (Bakin, 1999). Sparsity has demonstrated its benefits from mathematical: (Barron & Klusowski, 2018), (Neyshabur et al., 2015), (Taheri et al., 2020), (Schmidt-Hieber, 2020), (Taheri et al., 2021), (Bauer & Kohler, 2019), (Golestaneh et al., 2024), (Mohaddes et al., 2014) and (Beknazaryan, 2022a), algorithmic: (Zhu et al., 2019; Lemhadri et al., 2021), and applied perspectives: (Wen et al., 2016), (Dong et al., 2011), (Fu et al., 2015), (Christensen et al., 2009), (Ravishankar et al., 2019), (Sun & Li, 2017) and (Akbari et al., 2022).

However, standard notions of sparsity have certain limitations. The primary drawback is their ineffectiveness in reducing computational costs. For instance, it is well known that algorithms for Boolean and binary matrix multiplications are not faster than those for general matrix multiplications, with the most efficient algorithms

having a complexity of approximately $\mathcal{O}(n^{2.37})$ (Alman & Williams; Lee, 2002). As a result, they do not fully exploit the potential computational benefits of sparsity. Moreover, they ask for zero-valued parameters or data points, while we might target values other than zero as well. A related limitation is that regularizers and constraints (often $\ell_1$-type norms) that induce zero-valued parameters also imply an unwanted bias toward small parameter values (Lederer, 2021, Section 2.3 on Pages 45ff).

This paper introduces a new notion of sparsity, *cardinality sparsity*, which is beneficial for circumventing both computational and statistical challenges. This novel sparsity concept can enable rapid matrix multiplications, which are the primary source of computational issues in deep learning. Cardinality sparsity generalizes the idea of "few non-zero-valued parameters or data points" to "few unique parameter values or data points." Thus, cardinality sparsity provides a more nuanced and comprehensive notion of sparsity. It is inspired by four different lines of research: fusion sparsity (Tibshirani et al., 2005; Land & Friedman, 1997), which pushes adjacent parameter values together; dense-dictionary coding (Elgohary et al., 2016), which optimizes matrix-vector multiplications; layer sparsity (Hebiri & Lederer, 2020), which merges parameters of different layers across neural networks; and function approximations by neural networks with few unique parameter values (Beknazaryan, 2022b). Theorems 1 and 2 show that cardinality sparsity has essentially the same statistical effects in deep learning and tensor regression as the usual sparsity—yet without necessarily pushing parameters toward zero. Section 3 explains that cardinality sparsity can greatly reduce computational costs in pipelines that depend on large tensor-tensor multiplications, including once more deep learning and tensor regression.

Let us briefly elaborate on the computational aspect. The most time-consuming parts of many optimization processes are often matrix-matrix multiplications. The standard multiplication methods for $M \times P$ and $P \times N$ matrices have complexity of order $\mathcal{O}(MPN)$ (Papadimitriou, 2003). However, many types of data enjoy cardinality sparsity: for example, image data often consist of only 256 different values. We can then reduce the complexity; indeed, we show that the number of multiplications in the matrix multiplication process can be reduced to $\mathcal{O}(MN+mPn)$, (The term $MN$ arises because reconstructing the complete output from the compressed computation involves inspecting every element in the resulting matrix.) with $m$ and $n$ the maximal number of unique values in the columns of the first matrix and the maximum number of unique values in the rows of the second matrix, respectively, that is, typically $m \ll M$ and $n \ll N$. It should be noted that we assumed that the structure of the matrices is known. While the fast multiplication algorithm, with a complexity of approximately $\mathcal{O}(n^{2.37})$, (Le Gall, 2014; Alman & Williams), or $\mathcal{O}(n^3/(\log n^1.5))$, (Yu, 2015) remains applicable to any matrix, we focus on matrices with cardinality sparsity and known structure. By "matrix structure," we refer to both the compressed version and the encoding matrix that we will introduce in Section 3. This assumption is relevant in neural networks and machine learning algorithms, as the input matrix, which is the primary source of sparsity, remains consistent across all learning iterations. Therefore, we can determine the input matrix structure through preprocessing before initiating the training process. It is important to note that this preprocessing step only needs to be performed once, incurring minimal time overhead, especially when dealing with a large number of iterations. Additionally, this approach is advantageous for Boolean and binary matrix multiplication when the matrix contains only two distinct values. Moreover, another advantage of our method is that it remains valid even when one of the matrices is binary, whereas Boolean and binary multiplication methods usually require both matrices to satisfy binary conditions. This is particularly beneficial for example for categorical data when dummy encoding is applied. The key insight is to avoid redundant multiplications, allowing us to effectively condense the matrices. Further details are provided in Section 3. For illustration, consider $\boldsymbol{WV}$ with

$$\boldsymbol{W} := \begin{bmatrix} 2.1 & 1.1 \\ 1 & 2.3 \\ 1 & 1.1 \\ 2.1 & 1.1 \\ 3 & 2.3 \\ 3 & 4 \end{bmatrix} \quad \boldsymbol{V} := \begin{bmatrix} a & a & b & a \\ c & d & d & c \end{bmatrix}.$$

A naive multiplication of the two matrices requires $6 \times 2 \times 4 = 48$ multiplications of entries. Using the cardinality trick instead, we can consider the compressed matrices

$$\boldsymbol{C_W} := \left[ \begin{array}{cc} 2.1 & 1.1 \\ 1 & 2.3 \\ 3 & 4 \end{array} \right] \qquad \boldsymbol{C_V} := \left[ \begin{array}{cc} a & b \\ c & d \end{array} \right],$$

where $\boldsymbol{C_W}$ consists of the unique elements of the columns of $\boldsymbol{W}$ and $\boldsymbol{C_V}$ the unique elements of the rows of $\boldsymbol{V}$. We can then compute $\boldsymbol{C_W C_V}$ and finally "decompress" the resulting matrix to obtain $\boldsymbol{WV}$. This reduces the number of elementary multiplications to $3 \times 2 \times 2 = 12$. Furthermore, the compression also reduces the memory requirements because the matrices $\boldsymbol{C_W}$ and $\boldsymbol{C_V}$ are considerably smaller than their original counterparts $\boldsymbol{W}$ and $\boldsymbol{V}$.

Another crucial aspect of our multiplication method is its efficiency for matrices with a cardinality degree of order 2, such as binary (Boolean) matrices. Since the matrix consists of only two distinct values, pre-processing in this case is significantly simpler than in the general scenario. Furthermore, modifying the primary matrix to function as an encoding matrix can further streamline the pre-processing. This approach can substantially accelerate the multiplication process for binary matrices, as we will demonstrate in our simulations.

In addition, another major challenge in neural networks is the high memory demand for storing and processing input data. As we will see, by employing cardinality sparsity, we can perform multiplication in the compressed domain. We only need to save the compressed version of the input data, which can decrease the required storage memory (Elgohary et al., 2016). The main reason is that we avoid saving all real numbers, each of which requires 32 bits to store. Further details will be explored in the simulation experiments (Section 6.1.2, Table 1). Our three main contributions are:

- We introduce a new concept of sparsity, *cardinality sparsity*, defining a tensor as sparse if it contains only a small number of unique values.

- We demonstrate that cardinality sparsity can outperform state-of-the-art algorithms for matrix-matrix multiplications, including binary matrix-matrix multiplications, considerably. We also show that cardinality sparsity can reduce memory usage substantially.

- Introducing a new family of regularizers compatible with our notion of sparsity, and establishing corresponding statistical guarantees, we prove that cardinality sparsity can reduce statistical complexity as well.

These contributions provide a general framework for decreasing computational costs, reducing memory storage requirements, and addressing statistical complexity in machine learning systems and beyond.

The remainder of this manuscript is organized as follows. Section 2 recaps different types of sparsity and defines cardinality sparsity formally. Section 3 highlights the computational advantages of cardinality sparsity. Our statistical results are discussed in Section 4 and Section 5 where we explore cardinality sparsity in the contexts of deep learning and tensor regression, respectively. Simulation results are provided in Section 6. In Appendix A we review matrix-matrix multiplication algorithms, generalize our cardinality sparsity notion and review some technical results. The proofs deferred to the Appendix B.

## 2 Cardinality Sparsity

Sparsity has become a standard concept in statistics and machine learning, arguably most prominently in compressed sensing (Eldar & Kutyniok, 2012) and high-dimensional statistics (Lederer, 2021). This section introduces a concept of sparsity that, broadly speaking, limits the number of different parameter values. In the subsequent sections, we will illustrate the potential of this concept from the perspectives of statistics (Section 4 and Appendix 5) and computations (Section 4.2.1).

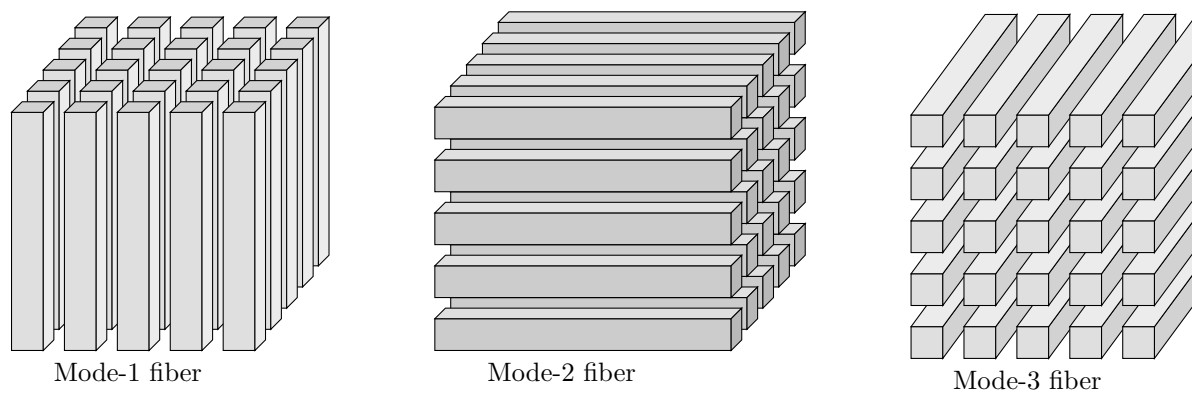

Figure 1: Different fibers for a third-order tensor

We consider a data-generating process indexed by an $N$th order tensor $\mathbb{A} \in \mathbb{R}^{Q_1 \times Q_2 \times \ldots \times Q_N}$. The most standard notion of sparsity concerns the number of nonzero elements: $\mathbb{A}$ is $k$-sparse if

$$\|\mathbb{A}\|_0 := \#\text{supp}[\mathbb{A}] \leq k,$$

where supp denotes the number of non-zero elements of the tensor. Instead, we count the number of different parameter values in each fiber. Fibers are the sub-arrays of a tensor whose indexes, all but one, are fixed. More precisely, the mode-$n$ fibers are the set of all sub-arrays (vectors) that are generated from fixing all but the $n$th index (Kolda & Bader, 2009; Bowen & Wang, 2008). Indeed, by $\mathbb{A}_{q_1, \cdots, q_{n-1}, \bullet, q_{n+1}, \cdots, q_N}$ we mean a vector of length $Q_n$ where $q_i$ is fixed for $\forall i \in \{1, \cdots, N\} \setminus \{n\}$. The fibers of a third-order tensor are illustrated in Figure 1. Now we are ready to provide the definition for our notion of sparsity:

**Definition 1** (Cardinality Sparsity). *We say that the tensor $\mathbb{A} \in \mathbb{R}^{Q_1 \times \ldots \times Q_N}$ is $(n, k)$-sparse if*

$$k \geq \max\Big\{ \#\{v_1, \ldots, v_{Q_n}\} : $$
$$(v_1, \ldots, v_{Q_n})^\top := \mathbb{A}_{q_1, \ldots, q_{n-1}, \bullet, q_{n+1}, \ldots, q_N},$$
$$q_1 \in \{1, \ldots, Q_1\}, \ldots, q_{n-1} \in \{1, \ldots, Q_{n-1}\},$$
$$q_{n+1} \in \{1, \ldots, Q_{n+1}\}, \ldots, q_N \in \{1, \ldots, Q_N\}\Big\}.$$

In other words, a tensor is $(n, k)$-sparse if the mode-$n$ fibers have at most $k$ different values. We call this notion of sparsity, cardinality sparsity.

We can also define cardinality sparsity for vectors, which actually are special cases of tensors. Cardinality sparsity for vectors is related to the fusion sparsity notion (Tibshirani et al., 2005; Land & Friedman, 1997). They assumed that the vectors are pice-wise constant. However, the main difference is that they assumed that the adjacent elements are similar. While in the cardinality sparsity, we look for similarities in the entire vector. Similar to tensors we say that a vector $\boldsymbol{v} := (v_1, \ldots, v_{J_n})^\top$ has cardinality sparsity if:

$$k := \#\{v_1, \ldots, v_{J_n}\} \ll J_n.$$

Another special case that we are interested in is sparsity in the matrices. We say a matrix has column-wise cardinality (mode-1 cardinality) sparsity if it has repeated values in each column. For example, the most sparse case (in the sense of this paper) is a matrix whose columns' elements are unique as follows:

$$\boldsymbol{W} := \begin{bmatrix} a & b & c & \ldots \\ a & b & c & \ldots \\ \vdots & \vdots & \vdots & \ddots \\ a & b & c & \ldots \end{bmatrix}.$$

Similarly, a row-wise (mode-2) sparse matrix is a matrix whose rows have cardinality sparsity property.

We now introduce an alternative definition of sparsity that is more convenient for analysis. This new definition allows us to concentrate on standard sparsity rather than cardinality sparsity. In order to provide this definition we first review some preliminaries. Consider a tensor $\mathbb{A} \in \mathbb{R}^{Q_1 \times Q_2 \times \dots \times Q_N}$. The mode-$n$ product of a tensor $\mathbb{A} \in \mathbb{R}^{Q_1 \times Q_2 \times \dots \times Q_N}$ by a matrix $\boldsymbol{U} \in \mathbb{R}^{J_n \times Q_n}$ is a tensor $\mathbb{B} \in \mathbb{R}^{Q_1 \times \dots \times Q_{n-1} \times J_n \times Q_{n+1} \times \dots \times Q_N}$, denoted as:

$$\mathbb{B} := \mathbb{A} \times_n \boldsymbol{U},$$

where each entry of $\mathbb{B}$ is defined as the sum of products of corresponding entries in $\mathbb{A}$ and $\boldsymbol{U}$ :

$$\mathbb{B}_{q_1, \dots, q_{n-1}, j_n, q_{n+1}, \dots, q_N} := \sum_{q_n} \mathbb{A}_{q_1, \dots, q_N} \cdot \boldsymbol{U}_{j_n, q_n}.$$

Also, we define a difference tensor $\mathbb{A}'$ as follows:

$$\mathbb{A}' := \mathbb{A} \times_n \boldsymbol{A}, \tag{1}$$

where the matrix $\boldsymbol{A}$ is of size $\binom{Q_n}{2} \times Q_n$ such that for each row of $\boldsymbol{A}$, there is one element equals to 1, one element equals to $-1$, and the remaining $(Q_n - 2)$ elements are zero. This means, that instead of studying mode-$n$ cardinality sparsity of a tensor, we can focus on the standard sparsity of the difference tensor.

Similarly, we can say a matrix $\boldsymbol{W} \in \mathbb{R}^{m \times n}$ has cardinality sparsity if the matrix resulting from the difference of each two elements in each column is sparse in the standard sense. Therefore, we concentrate on the difference matrix. The difference matrix could be considered as the product of two matrices $\boldsymbol{A}$ and $\boldsymbol{W}$. Where again, the matrix $\boldsymbol{A}$ is of size $\binom{m}{2} \times m$ such that for each row of $\boldsymbol{A}$, there is one element equal to 1, one element equal to $-1$, and the remaining $(m - 2)$ elements are zero. For more clarification, we provide the below example.

**Example 1.** *As an example, assume that $m = n = 3$ and write*

$$\boldsymbol{W} := \begin{bmatrix} w_{11} & w_{12} & w_{13} \\ w_{21} & w_{22} & w_{23} \\ w_{31} & w_{32} & w_{33} \end{bmatrix},$$

*where $w_{ij} \in \mathbb{R}$ for all $i, j \in \{1, 2, 3\}$. According to the construction of the matrix $\boldsymbol{A}$ mentioned above, we can set*

$$\boldsymbol{A} := \begin{bmatrix} 1 & -1 & 0 \\ 0 & 1 & -1 \\ -1 & 0 & 1 \end{bmatrix},$$

*and hence difference matrix $\boldsymbol{W}'$ is equal to:*

$$\boldsymbol{W}' := \boldsymbol{AW} := \begin{bmatrix} w_{11} - w_{21} & w_{12} - w_{22} & w_{13} - w_{23} \\ w_{21} - w_{31} & w_{22} - w_{32} & w_{23} - w_{33} \\ w_{31} - w_{11} & w_{32} - w_{12} & w_{33} - w_{13} \end{bmatrix}.$$

We can observe that this transformation helps us to measure the similarities between any two rows of each matrix. In particular, if all of the rows of $\boldsymbol{W}$ are the same, then we obtain that $\boldsymbol{AW} = \boldsymbol{0}$. Therefore, cardinality sparsity could be induced by employing matrix $\boldsymbol{A}$.

## 3 Cardinality Sparsity in Computational Cost Reduction

As we mentioned another important aspect of cardinality sparsity is reducing the computational cost. For the sake of completeness, we provide a matrix-matrix multiplication method that reduces computations for matrices with cardinality sparsity. This method is inspired by the matrix-vector multiplication approach studied in (Elgohary et al., 2016). The matrix-matrix multiplication would appear in many applications. In a neural network, we need to perform a chain of matrix multiplication. For the case when the input

is a matrix, we can employ matrix-matrix multiplication in each step to find the output. Moreover, in tensor analysis, tensor-matrix products also could be computed by some matrix products. Therefore, an efficient matrix-matrix multiplication method can reduce computational costs in many applications. In the next part, we employ cardinality sparsity to provide a matrix multiplication method with low computational complexity.

## 3.1 Matrix Multiplication for Sparse Matrices

In this part, we review the necessary background to introduce our matrix-matrix multiplication algorithm. The elements of a matrix $\boldsymbol{W} \in \mathbb{R}^{M \times P}$ which is sparse in our sense, can be described as follows

$$w_{i,j} \in \{a_{1,j}, ..., a_{m,j}\};$$

where $j \in \{1, \cdots, P\}$ is fixed and $m \ll M$ is the maximum number of unique elements in each column (where for simplicity we did not change the order of unique elements in each column). We define an indicator function that denotes the positions of each repeated element in a fixed column as follows

$$\boldsymbol{I}(a_{t,j}) := t; \ \ t \in \{1, \cdots, m\}.$$

If the number of unique elements in a column was $s$ less than $m$ we then put $m-s$ remaining elements are equal to zero. Therefore, we obtain a matrix $\boldsymbol{I} \in \mathbb{Z}^{M \times P}$ whose elements are integers from the set $\{0, 1, \cdots, m\}$. Consider matrix-vector multiplication $\boldsymbol{W} \times \boldsymbol{v}$ where $\boldsymbol{v} = [v_1, \cdots, v_P]^{\top}$. To perform this multiplication it is enough to consider the unique elements of each column of $\boldsymbol{W}$ and compute their multiplication with the corresponding element of $\boldsymbol{v}$. Finally, we use the matrix $\boldsymbol{I}$ to reconstruct the final result.

### 3.1.1 Matrix-Matrix Multiplication Using Cardinality Sparsity

To develop an efficient matrix-matrix multiplication method, we consider two scenarios. We first assume that $P$ is not very large compared with $M$ and $N$. The key point for the matrix-matrix multiplication method (for $P$ is small) is to employ the below definition.

**Definition 2.** *Consider the multiplication between $\boldsymbol{W}_{M \times P}$ and $\boldsymbol{V}_{P \times N}$. In this case, we have*

$$\boldsymbol{W} \times \boldsymbol{V} := \sum_{i=1}^{P} \boldsymbol{W}^{(i)} \otimes \boldsymbol{V}_{(i)}, \tag{2}$$

*where $\boldsymbol{W}^{(i)}$ is the ith column of the matrix $\boldsymbol{W}$ and $\boldsymbol{V}_{(i)}$ is the ith row of $\boldsymbol{V}$, and each $\boldsymbol{W}^{(i)} \otimes \boldsymbol{V}_{(i)}$ is a $M \times N$ rank-one matrix, computed as the tensor (outer) product of two vectors.*

This definition can lead us to a matrix-matrix multiplication method with less number of multiplications. We again assume $w_{i,j} \in \{a_{1,j}, ..., a_{m,j}\}$; when $j \in \{1, \cdots, P\}$ is fixed and $m \ll M$ is the maximum number of unique elements in each column of $\boldsymbol{W}$. Also we assume that $v_{j,k} \in \{b_{j,1}, ..., b_{j,n}\}$; where $j \in \{1, \cdots, P\}$ is fixed and $n$ is the maximum number of unique elements in each row of $\boldsymbol{V}$.

Our goal is to find a method in which we can generate elements in Equation (2) with less number of computations. We then compute the summation over these elements. Similar to the previous method we define two matrices $\boldsymbol{I}$ and $\boldsymbol{J}$ as follows

$$\boldsymbol{I}(a_{i,j}) := i; \ i \in \{1, \cdots, m\}, \boldsymbol{J}(b_{j,k}) := k; \ k \in \{1, \cdots, n\}.$$

Now we compute the tensor product between two vectors $C_{\boldsymbol{W}^{(j)}} = [a_{1,j} \ ... \ a_{m,j}]^{\top}$ and $C_{\boldsymbol{V}_{(j)}} = [b_{j,1} \ ... \ b_{j,n}]$ (vectors obtained from unique elements of the $j$th column of $\boldsymbol{W}$ and $j$th row of $\boldsymbol{V}$). For reconstructing the $j$th element in Equation (2) we use the aforementioned tensor product and the Cartesian product (which in this paper we show by $\times_c$) between $j$th column of $\boldsymbol{I}$ and $j$th row of $\boldsymbol{J}$. This Cartesian product denotes the map which projects elements of $C_{\boldsymbol{W}^{(j)}} \otimes C_{\boldsymbol{V}_{(j)}}$ to the matrices of size $M \times N$. In the below we provide a numerical example to illustrate the procedure.

**Example 2** (Fast multiplication for sparse matrices)**.** *Let*

$$\boldsymbol{W} := \begin{bmatrix} 2.1 & 1.1 \\ 1 & 2.3 \\ 1 & 1.1 \\ 2.1 & 1.1 \\ 3 & 2.3 \\ 3 & 4 \end{bmatrix}; \boldsymbol{V} := \begin{bmatrix} a & a & b & a \\ c & d & d & c \end{bmatrix}$$

*We compute the first element in Equation (2) as an example. We have $\boldsymbol{W}^{(1)} = \begin{bmatrix} 2.1 & 1 & 1 & 2.1 & 3 & 3 \end{bmatrix}^{\top}$ and $\boldsymbol{V}_{(1)} = \begin{bmatrix} a & a & b & a \end{bmatrix}$. In order to implement the proposed method we compute $C_{\boldsymbol{W}^{(1)}} = \begin{bmatrix} 2.1 & 1 & 3 \end{bmatrix}^{\top}$, and $C_{\boldsymbol{V}_{(1)}} = \begin{bmatrix} a & b \end{bmatrix}$. Also by definition of $\boldsymbol{I}$ and $\boldsymbol{J}$ we obtain firs column of $\boldsymbol{I}$ and the first row of $\boldsymbol{J}$ is equal to $\boldsymbol{I}^{(1)} = \begin{bmatrix} 1 & 2 & 2 & 1 & 3 & 3 \end{bmatrix}^{\top}$ and $\boldsymbol{J}_{(1)} = \begin{bmatrix} 1 & 1 & 2 & 1 \end{bmatrix}$ respectively. The tensor product of $C_{\boldsymbol{W}^{(1)}}$ and $C_{\boldsymbol{V}_{(1)}}$ is equal to*

$$\boldsymbol{D}_1 := \begin{bmatrix} 2.1a & 2.1b \\ a & b \\ 3a & 3b \end{bmatrix}$$

*Now we find the Cartesian product of $\boldsymbol{I}^{(1)}$ and $\boldsymbol{J}_{(1)}$ to map $\boldsymbol{D}_1$ to $\boldsymbol{W}^{(1)} \otimes \boldsymbol{V}_{(1)}$ as follows*

$$\boldsymbol{I}^{(1)} \times_c \boldsymbol{J}_{(1)} = \begin{bmatrix} (1,1) & (1,1) & (1,2) & (1,1) \\ (2,1) & (2,1) & (2,2) & (2,1) \\ (2,1) & (2,1) & (2,2) & (2,1) \\ (1,1) & (1,1) & (1,2) & (1,1) \\ (3,1) & (3,1) & (3,2) & (3,1) \\ (3,1) & (3,1) & (3,2) & (3,1) \end{bmatrix}$$

*This helps us to map $\boldsymbol{D}_1$ to $\boldsymbol{W}^{(1)} \otimes \boldsymbol{V}_{(1)}$. The second term in Equation (2) can be obtained similarly.*

The Cartesian product is just concatenation of elements and does not increase computational complexity. This new matrix-matrix multiplication procedure can lead to computations of order $\mathcal{O}(mPn)$ which could be much less than usual matrix multiplication methods ($\mathcal{O}(MPN); m \ll M, n \ll N$).

This multiplication method is particularly useful when $P$ is not a large number. However, if $P$ is large, a different approach is more effective. In this case, we use the usual definition of matrix-matrix multiplication, i.e. we consider each element of the multiplication result as the inner product of rows of the first matrix and columns of the second matrix. This approach lets us reduce computational costs. In this case, we generate matrix $\boldsymbol{I}$ and $C_{\boldsymbol{W}}$ for the first matrix so that we perform encoding row-wise. In order to obtain the $(i,j)$th element of the multiplication result we consider the inner product of the $i$th row of the first matrix and the $j$th column of the second matrix. In order to perform factorization, we employ the $i$th row of matrix $I$ and perform summation over elements of the column of the second matrix based on the $i$th row of $I$. For example, if we have

$$C_{\boldsymbol{W}^{(i)}} := \begin{bmatrix} 1.1 & 2.3 \end{bmatrix}^{\top}; \quad I_i := \begin{bmatrix} 1 & 1 & 2 & 1 & 2 \end{bmatrix}^{\top}$$

Then we sum the first, second, and fourth elements of the $i$th column of the second matrix and multiply by the first element of $C_{\boldsymbol{W}^{(i)}}$. We also sum the third and fifth elements of the $i$th column of the second matrix and multiply by the second element of $C_{\boldsymbol{W}^{(i)}}$ and finally sum these two results. Algorithms 2 and 3 explain these multiplication methods, while the standard multiplication algorithm is reviewed in Algorithm 1.

**Example 3** (Fast binary matrix multiplication)**.** *Now we provide a crucial example of binary matrices. The key point is that in the binary case, since there are only two possible values, the preprocessing process is much simpler than for the general case. To perform preprocessing, we need to find the compressed matrix and the encoding matrix. For matrices with column-wise cardinality sparsity, we store the first row, and the second row of the compressed matrix is simply the complement of the first row, resulting in the compressed*

*matrix. The encoding matrix is obtained by storing columns of the original matrix that start with zero and the complements of those that start with one. This method streamlines the preprocessing process and leads to further computational reductions.*

*For example, in the following, you can observe a binary matrix along with its equivalent encoding and compressed matrix:*

$$A = \begin{bmatrix} 0 & 0 & 1 \\ 0 & 1 & 0 \\ 1 & 1 & 0 \\ 0 & 0 & 1 \\ 1 & 0 & 0 \end{bmatrix} \quad I_A = \begin{bmatrix} 0 & 0 & 0 \\ 0 & 1 & 1 \\ 1 & 1 & 1 \\ 0 & 0 & 0 \\ 1 & 0 & 1 \end{bmatrix} \quad C_A = \begin{bmatrix} 0 & 0 & 1 \\ 1 & 1 & 0 \end{bmatrix}$$

*The rest of the multiplication process is similar to the general case. We omit the multiplication steps to prevent redundancy.*

The experiment for this section are presented in Figures 2, 3, 4, 5, 6, 7, 8, 9, 10, 12, 13, and 14.

## 4   Cardinality Sparsity in Deep Learning

In (Hebiri & Lederer, 2020), the authors investigated the notion of layer sparsity for neural networks. Cardinality sparsity is analogous to layer sparsity, but instead addresses the sparsity in the width of the network. Moreover, recent work on function approximation (Beknazaryan, 2022b) has demonstrated that deep neural networks can effectively approximate a broad range of functions with a smaller set of unique parameters. This provides further motivation to focus on cardinality sparsity as a means of reducing the network's parameter space and increasing its learning speed.

In this section, we will first review the neural network model, as well as some notations and definitions that will be necessary. We will then study a general family of regularizers that includes both the cardinality and $\ell_1$ regularizers as special cases. While the standard $\ell_1$ regularizer promotes sparsity but does not explicitly discourage large values, this broader class of sparsity-inducing regularizers can also help control large magnitudes in the solution. Finally, we will provide a prediction guarantee for the estimators of a network with cardinality sparsity, and study the computational aspects of the the introduced regularizer.

We focus on regression rather than classification simply because regression is the statistically more challenging problem: for example, the boundedness of classification allows for the use of techniques like McDiarmid's inequality (McDiarmid, 1989; Devroye et al., 2013).

### 4.1   Deep Learning Framework

In this part, we briefly discuss some assumptions and notations of deep neural networks. Consider the below regression model:

$$y_i \ := \ \mathfrak{g}[\boldsymbol{x}_i] + u_i, \qquad i \in \{1, \dots, n\},$$

for data $(y_1, \boldsymbol{x}_1), \dots (y_n, \boldsymbol{x}_n) \in \mathbb{R} \times \mathbb{R}^d$. Where $\mathfrak{g} : \mathbb{R}^d \mapsto \mathbb{R}$ is an unknown data-generating function and $u_1, \cdots, u_n \in \mathbb{R}$ is the stochastic noise. Our goal is to fit the data-generating function $\mathfrak{g}$ with a feed-forward neural network $\mathfrak{g}_\Theta : \mathbb{R}^d \mapsto \mathbb{R}$

$$\mathfrak{g}_\Theta[\boldsymbol{x}] \ := \ \boldsymbol{W}^L \mathfrak{f}^L\Big[\dots \boldsymbol{W}^1 \mathfrak{f}^1\big[\boldsymbol{W}^0 \boldsymbol{x}\big]\Big], \qquad \boldsymbol{x} \in \mathbb{R}^d,$$

indexed by $\Theta := (\boldsymbol{W}^L, \dots, \boldsymbol{W}^0)$. Where $L$ is a positive integer representing the number of hidden layers and $\boldsymbol{W}^l \in \mathbb{R}^{p_{l+1} \times p_l}$ are the weight matrices for $l \in \{0, \dots, L\}$ with $p_0 = d$ and $p_{L+1} = 1$. For each $l \in \{0, \dots, L\}$, the functions $\mathfrak{f}^l : \mathbb{R}^{p_{l+1}} \mapsto \mathbb{R}^{p_{l+1}}$ are the activation functions. We assume that the activation functions satisfy $\mathfrak{f}^l[\boldsymbol{0}_{p_l}] = \boldsymbol{0}_{p_l}$ and are $a_{\text{Lip}}$-Lipschitz continuous for a constant $a_{\text{Lip}} \in [0, \infty)$ and with respect to the Euclidean norms on their input and output spaces:

$$\|\mathfrak{f}^l[\boldsymbol{z}] - \mathfrak{f}^l[\boldsymbol{z}']\| \ \leq \ a_{\text{Lip}} \|\boldsymbol{z} - \boldsymbol{z}'\| \qquad \text{for all } \boldsymbol{z}, \boldsymbol{z}' \in \mathbb{R}^{p_{l+1}}.$$

Furthermore, we assume that the activation function $\mathfrak{f}^l : \mathbb{R}^{p_{l+1}} \mapsto \mathbb{R}^{p_{l+1}}$ is a non-negative homogeneous of degree 1 (compared to (Taheri et al., 2021))

$$\mathfrak{f}[s\boldsymbol{z}] \;=\; s\mathfrak{f}[\boldsymbol{z}] \qquad \text{for all } s \in [0,\infty) \text{ and } \boldsymbol{z} \in \mathbb{R}^{p_{l+1}}.$$

A standard example of activation functions that satisfy these assumptions are ReLU functions (Taheri et al., 2021). We also denote by

$$\mathcal{A} \;:=\; \left\{ \Theta := (\boldsymbol{W}^L, \ldots, \boldsymbol{W}^0) : \boldsymbol{W}^l \in \mathbb{R}^{p_{l+1} \times p_l} \right\},$$

as the collection of all possible weight matrices for $\mathfrak{g}_\Theta$. Taking into account the high dimensionality of $\mathcal{A}$ ($\sum_{l=0}^{L} p_{l+1}p_l \in [n,\infty)$), and following (Taheri et al., 2021) ; we can employ the below estimator:

$$(\hat{\kappa}, \hat{\Omega}) \in \underset{\substack{\kappa \in [0,\infty) \\ \Omega \in \mathcal{A}_\mathfrak{h}}}{\arg\min} \left\{ \frac{1}{n} \sum_{i=1}^{n} \left( y_i - \kappa \mathfrak{g}_\Omega[\boldsymbol{x}_i] \right)^2 + \lambda\kappa \right\},$$

where $\lambda \in [0,\infty)$ is a tuning parameter, $\mathfrak{h} : \mathcal{A} \mapsto \mathbb{R}$ indicates the penalty term and the corresponding unit ball denotes by $\mathcal{A}_\mathfrak{h} = \{\Theta | \mathfrak{h}[\Theta] < 1\}$. This regularization enables us to focus on the effective noise: $z_\mathfrak{h} := \sup_{\Omega \in \mathcal{A}_\mathfrak{h}} |2/n \sum_{i=1}^{n} \mathfrak{g}_\Omega[\boldsymbol{x}_i]u_i|$, which is related to the Gaussian and Rademacher complexities of the function class $\mathcal{G}_\mathfrak{h} := \{\mathfrak{g}_\Omega : \Omega \in \mathcal{A}_\mathfrak{h}\}$. We need to show that the effective noise is controlled by the tuning parameter with high probability. In order to consider the control on the effective noise similar to (Taheri et al., 2021) we define:

$$\lambda_{\mathfrak{h},t} \in \min \left\{ \delta \in [0,\infty) : \mathbb{P}(z_\mathfrak{h} \leq \delta) \geq 1 - t \right\},$$

which is the smallest tuning parameter for controlling the effective noise at level $t$. We consider the prediction error as

$$\text{err}[\kappa \mathfrak{g}_\Omega] := \sqrt{ \frac{1}{n} \sum_{i=1}^{n} \left( \mathfrak{g}[\boldsymbol{x}_i] - \kappa \mathfrak{g}_\Omega[\boldsymbol{x}_i] \right)^2 }$$

$$\text{for } \kappa \in [0,\infty), \Omega \in \mathcal{A}_\mathfrak{h}.$$

Furthermore, we assume that noise variables $u_i$ are independent, centered, and uniformly sub-Gaussian for constants $K, \gamma \in (0,\infty)$ (van de Geer, 2000, page 126); (Vershynin, 2018, Section 2.5):

$$\max_{i \in \{1,\ldots,n\}} K^2 \left( \mathbb{E}e^{\frac{|u_i|^2}{K^2}} - 1 \right) \leq \gamma^2.$$

In addition, we give some definitions which are convenient for illustrating the main results of this section. For a vector $\boldsymbol{z}$ and a weight matrix $\boldsymbol{W}^l$ we say

$$\|\boldsymbol{z}\|_n \;:=\; \sqrt{ \sum_{i=1}^{n} \|z_i\|_2^2 / n },$$

and

$$\|\boldsymbol{W}^l\|_1 \;:=\; \sum_{k=1}^{p_{l+1}} \sum_{j=1}^{p_l} \left| \boldsymbol{W}_{kj}^l \right|. \tag{3}$$

We also need the usual matrix norm (Geijn, 2014, page 5):

**Definition 3.** *Suppose that $\|\cdot\|_\alpha : \mathbb{R}^n \to \mathbb{R}$ and $\|\cdot\|_\tau : \mathbb{R}^m \to \mathbb{R}$ are vector norms. For $\boldsymbol{B} \in \mathbb{R}^{m \times n}$ we define:*

$$\|\boldsymbol{B}\|_{\alpha \to \tau} \;:=\; \max_{\boldsymbol{x} \in \mathbb{R}^n \setminus \{0\}} \frac{\|\boldsymbol{B}\boldsymbol{x}\|_\tau}{\|\boldsymbol{x}\|_\alpha}.$$

Let $\|\cdot\|_\alpha$ is a norm on $\mathbb{R}^n$, $\|\cdot\|_\tau$ is a norm on $\mathbb{R}^m$ and $\|\cdot\|_\zeta$ is a norm on $\mathbb{R}^p$ then for any $\boldsymbol{x} \in \mathbb{R}^n$:

$$\|\boldsymbol{BCx}\|_\zeta \ \leq \ \|\boldsymbol{B}\|_{\tau \to \zeta} \|\boldsymbol{Cx}\|_\tau \ \leq \ \|\boldsymbol{B}\|_{\tau \to \zeta} \|\boldsymbol{C}\|_{\alpha \to \tau} \|\boldsymbol{x}\|_\alpha.$$

We now introduce some other parameters which are used in the main result of this section. We set

$$\boldsymbol{G} \ := \ (\boldsymbol{G}^L, ..., \boldsymbol{G}^0), \tag{4}$$

and

$$M_{\boldsymbol{G}} \ := \ \max_j \|\bar{\boldsymbol{G}}^{j^{-1}}\|_{1 \to 1}, \tag{5}$$

where $\boldsymbol{G}^j$; $j \in \{0, \cdots, L\}$ are non-invertible matrices and $\bar{\boldsymbol{G}}^j$ is the invertible matrix induced by the matrix $\boldsymbol{G}^j$, and its inverse matrix denotes by $\bar{\boldsymbol{G}}^{-1} \ : \ \mathrm{Im}(\boldsymbol{G}^j) \to \ker(\boldsymbol{G})^\perp$.

## 4.2 A General Statistical Guarantee for Regularized Estimators

Regularization plays an essential role in sparse regimes. Sparsity may lead to degenerate or unstable solutions, since multiple sparse parameterizations can yield the same mapping. In this section we propose a regularizer which resolves this by stabilizing the solution space, reducing redundancy, and enforcing structured sparsity.

To induce cardinality sparsity, as shown in Equation (1), one can promote standard sparsity in $\boldsymbol{A}\Theta$ for $\Theta \in \mathcal{A}$. Where we define $\boldsymbol{A} \ := \ (\boldsymbol{A}^L, ..., \boldsymbol{A}^0)$, for

$$\Theta \ := \ (\boldsymbol{W}^L, \ldots, \boldsymbol{W}^0) \text{ for } \boldsymbol{W}^l \in R^{p_{l+1} \times p_l},$$

which belongs to parameter space:

$$\mathcal{A} \ := \ \{(\boldsymbol{W}^L, \ldots, \boldsymbol{W}^0) : \boldsymbol{W}^l \in R^{p_{l+1} \times p_l}\}.$$

Indeed in this case, we again consider a neural network with $L$ hidden layers. For $l \in \{0, \ldots, L\}$, our first step is to transform each weight matrix into difference matrix, $\boldsymbol{W}^l$ to $\boldsymbol{A}^l \boldsymbol{W}^l$. Where similar to Equation (1), we define $\boldsymbol{A}^l \in \mathbb{R}^{\binom{p_{l+1}}{2} \times p_{l+1}}$ so that each row of $\boldsymbol{A}^l$ corresponds to a distinct pair of indices $(i,j)$ with $1 \leq i < j \leq p_{l+1}$. In the row associated with $(i,j)$, the entry in column $i$ is set to 1, the entry in column $j$ is set to $-1$, and all remaining $p_{l+1} - 2$ entries are zero. Thus, $\boldsymbol{A}^l$ enumerates all possible index pairs, yielding $\binom{p_{l+1}}{2}$ rows in total.

The term $\|\boldsymbol{A}\Theta\|_1$ penalizes absolute differences, encouraging many of them to be zero and thereby promoting equality among paired elements. Consequently, this term enforces cardinality sparsity in the matrices $\Theta$, meaning that each column of $\Theta$ contains repeated elements. The sparsest case occurs when each column contains only a single unique value.

However, a limitation arises because $\|\boldsymbol{A}\Theta\|_1$ alone is not a norm: it vanishes for any constant-per-column matrix (i.e., $\boldsymbol{A}\Theta = 0$ for all such $\Theta$ in the kernel of $\boldsymbol{A}$). This allows arbitrarily large constants to pass without penalty, potentially leading to degenerate solutions. Moreover, since the measure is based on differences of rows, distinct matrices $\Theta$ can yield the same value of $\boldsymbol{A}\Theta$.

To address these issues, we introduce a second term, $\nu \|\Pi_{\boldsymbol{A}}\Theta\|_1$, where $\Pi_{\boldsymbol{A}}$ is the projection onto $\ker(\boldsymbol{A})$. This term ensures that the projection of $\Theta$ onto the kernel, namely, the component corresponding to a matrix with a single repeated element per column, is also sparse. In addition, the inclusion of this term guarantees that the overall regularizer is a norm (Proposition 1).

This family of regularizers can be viewed as a generalization of the standard $\ell_1$ regularizer. While the usual $\ell_1$ penalty does not prevent large values, this form of sparsity control also mitigates the effect of large magnitudes. Accordingly, the regularizer can be expressed as follows:

$$\mathfrak{h}[\Theta] \ := \ \|\boldsymbol{A}\Theta\|_1 + \nu \|\Pi_{\boldsymbol{A}}\Theta\|_1, \tag{6}$$

where $\Pi_{\boldsymbol{A}} : \mathcal{A} \to \ker(\boldsymbol{A})$ is the projection to $\ker(\boldsymbol{A})$ and $\nu$ is a positive constant.

In this paper, we focus on a more general statistical guarantee for regularized estimators, of which cardinality-based sparsity and the standard notion of sparsity are only two special cases. We consider regularizers of the form

$$\mathfrak{h}[\Theta] \; := \; \|\boldsymbol{G}\Theta\|_1 + \nu\|\Pi_{\boldsymbol{G}}\Theta\|_1 \quad \text{for } \Theta \in \mathcal{A}, \tag{7}$$

where $\boldsymbol{G} \in \mathbb{R}^{N \times P_r}$; ($N \geq 2$; $P_r := \sum_{l=1}^{L+1} p_l$) is an (invertible or non-invertible) matrix, $\nu \in (0, \infty)$ is a constant, and $\Pi_{\boldsymbol{G}} \in \mathbb{R}^{\dim(\ker(\boldsymbol{G})) \times P_r}$ is the projection onto $\ker(\boldsymbol{G})$, the kernel of $\boldsymbol{G}$. One special case is $\ell_1$-regularization, where $\boldsymbol{G} = \boldsymbol{0}$. Another special case of the regularizer is the one that induces cardinality sparsity, where $\boldsymbol{G} = \boldsymbol{A}$.

Studying the effective noise introduced in the previous section enables us to prove the main theorem of this section. We show that the $\lambda_{\mathfrak{h},t}$ must satisfy $\lambda_{\mathfrak{h},t} \leq 2\delta$. Where $\delta$ will be explicitly defined in Equation (11). Combining this with Theorem 2 of (Taheri et al., 2021) we can prove the below theorem for prediction guarantee of the introduced regularizer.

**Theorem 1** (Prediction guarantee for deep learning). *Assume that*

$$\lambda \geq \; C(M_{\boldsymbol{G}} + 1)(a_{\mathrm{Lip}})^{2L}\|\boldsymbol{x}\|_n^2 \left(\frac{\sqrt{2}}{L}\right)^{2L-1}$$
$$\times \; \sqrt{\log(2P)}\,\frac{\log(2n)}{\sqrt{n}}, \tag{8}$$

*where $C \in (0, \infty)$ is a constant that depends only on the sub-Gaussian parameters $K$ and $\gamma$ of the noise. Then, for $n$ large enough, and $\nu \geq 1/(1 - M_{\boldsymbol{G}})$*

$$\mathrm{err}^2[\widehat{\kappa}_{\mathfrak{h}}\mathfrak{g}_{\tilde{\Omega}_{\mathfrak{h}}}] \; \leq \; \inf_{\substack{\kappa \in [0,\infty) \\ \Omega_{\mathfrak{h}} \in \mathcal{A}_{\mathfrak{h}}}} \left\{\mathrm{err}^2[\kappa\mathfrak{g}_{\Omega_{\mathfrak{h}}}] + 2\lambda\kappa\right\},$$

*with probability at least $1 - 1/n$.*

The bound in Theorem 1 establishes essentially a $(1/\sqrt{n})$-decrease of the error in the sample size $n$, a logarithmic increase in the number of parameters $P$, and an almost exponential decrease in the number of hidden layers $L$ if everything else is fixed. In addition, we know that $a_{\mathrm{Lip}}$ is equal to one for ReLU activation functions. This theorem is valid for any matrix in general and we just need to adjust parameter $M_{\boldsymbol{G}}$. We continue this part with two corollaries that result from Theorem 1.

**Corollary 1** (Standard $\ell_1$-sparsity). *Let $\boldsymbol{G} = \boldsymbol{0}$ then $M_{\boldsymbol{0}} = 0$, and for $\nu = 1$ the regularizer in (7) would be changed to $\ell_1$-regularizer. In this case, the Equation (11) for a large number of parameters could be written as follows*

$$\lambda \; \geq \; C(a_{\mathrm{Lip}})^{2L}\|\boldsymbol{x}\|_n^2 \left(\frac{\sqrt{2}}{L}\right)^{2L-1} \sqrt{\log 2P}\,\frac{\log 2n}{\sqrt{n}}, \tag{9}$$

*and then, for $n$ large enough,*

$$\mathrm{err}^2[\widehat{\kappa}_{\mathfrak{h}}\mathfrak{g}_{\tilde{\Omega}_{\mathfrak{h}}}] \; \leq \; \inf_{\substack{\kappa \in [0,\infty) \\ \Omega_{\mathfrak{h}} \in \mathcal{A}_{\mathfrak{h}}}} \left\{\mathrm{err}^2[\kappa\mathfrak{g}_{\Omega_{\mathfrak{h}}}] + 2\lambda\kappa\right\},$$

*with probability at least $1 - 1/n$.*

This corollary states that $\ell_1$-regularizer is a special case of regularizer in Equation (7). This corresponds to the setup in (Taheri et al., 2021). Another exciting example of matrix $\boldsymbol{G}$ is the matrix $\boldsymbol{A}$. Before illustrating the next corollary for cardinality sparsity, we provide the below lemma.

**Lemma 1** (Upper bound for $M_{\boldsymbol{A}}$). *Let $\boldsymbol{G} = \boldsymbol{A}$. Then we have*

$$M_{\boldsymbol{A}} \; \leq \; \sup_{j \in \{0,\dots,L\}} \frac{2}{\sqrt{P_j}}, \tag{10}$$

*where $P_j$ denotes the number of parameters of each weight matrix.*

This lemma implies that, the value of $M_{\boldsymbol{A}}$ is always less than two. Moreover, in practice, deep learning problems contains many parameters and as a result $\sqrt{P_j}$ is a large number thus $M_{\boldsymbol{A}}$ is a real number close to zero ($M_{\boldsymbol{A}} \ll 1$).

**Corollary 2** (Cardinality sparsity). *Let $\boldsymbol{G} = \boldsymbol{A}$ then $M_{\boldsymbol{A}} \ll 1$, and in this case, the Equation (11) for large number of parameters could be written as follows*

$$\lambda \ \geq \ C(a_{\text{Lip}})^{2L}\|\boldsymbol{x}\|_n^2 \left(\frac{\sqrt{2}}{L}\right)^{2L-1} \sqrt{\log 2P}\frac{\log 2n}{\sqrt{n}}, \tag{11}$$

*and then, for $n$ large enough, and $\nu \ \geq \ 1/(1 - M_{\boldsymbol{A}})$*

$$\text{err}^2[\widehat{\kappa}_{\mathfrak{h}}\mathfrak{g}_{\tilde{\Omega}_{\mathfrak{h}}}] \ \leq \ \inf_{\substack{\kappa \in [0,\infty) \\ \Omega_{\mathfrak{h}} \in \mathcal{A}_{\mathfrak{h}}}} \left\{\text{err}^2[\kappa\mathfrak{g}_{\Omega_{\mathfrak{h}}}] + 2\lambda\kappa\right\},$$

*with probability at least $1 - 1/n$.*

### 4.2.1 Further Insights into Cardinality Sparsity Regularization

Now, we further investigate another aspect of the introduced regularizer. Recall that $\mathfrak{h}[\Theta] \ := \ \|\boldsymbol{A}\Theta\|_1 + \nu\|\Pi_{\boldsymbol{A}}\Theta\|_1$, where, $\Theta \ = \ (\boldsymbol{W}^L, \ldots, \boldsymbol{W}^0)$ for $\boldsymbol{W}^l \in R^{p_{l+1} \times p_l}$. In the above equation the first term is $\ell_1$-regularizer, the second term however, $(\Pi_{\boldsymbol{A}}\Theta)$ needs more investigation. We know that the kernel space of $\boldsymbol{A}^l$ is a matrix as follows (elements in each column are the same)

$$\begin{bmatrix} \beta_1 & \beta_2 & \cdots & \beta_n \\ \beta_1 & \beta_2 & \cdots & \beta_n \\ \vdots & \vdots & \ddots & \vdots \\ \beta_1 & \beta_2 & \cdots & \beta_n \end{bmatrix}$$

Therefore, the projection of a matrix $\boldsymbol{W}^l$ on this space, i.e. $\Pi_{\boldsymbol{A}}\Theta$, would be of the below form

$$\begin{bmatrix} \frac{w_{11}+w_{21}+\cdots+w_{n1}}{n} & \frac{w_{12}+w_{22}+\cdots+w_{n2}}{n} & \cdots \\ \frac{w_{11}+w_{21}+\cdots+w_{n1}}{n} & \frac{w_{12}+w_{22}+\cdots+w_{n2}}{n} & \cdots \\ \vdots & \vdots & \ddots \end{bmatrix},$$

where $w_{ij}$s are entries of matrix $\boldsymbol{W}^l$. As a result, calculating $\Pi_{\boldsymbol{A}}\Theta$ is not computationally expensive and we just need to compute mean of each column. Statistical benefits experiments are presented in Figure 11.

Also note that from Equation (6), we apply the regularizer below in the cardinality-inducing case,

$$\mathfrak{h}[\Theta] \ := \ \|\boldsymbol{A}\Theta\|_1 + \nu\|\Pi_{\boldsymbol{A}}\Theta\|_1.$$

Since according to Proposition 1 the above equation is norm, it is consequently convex. Therefore, our regularizer remains convex, in contrast to direct cardinality penalties, which are indeed non-convex and combinatorial. Our approach leverages norm regularization to promote sparsity in a tractable manner, avoiding the computational difficulties of solving combinatorial problems.

In addition, to further mitigate computational complexity, one approach is to apply the projection $P_k$ onto the space of matrices with cardinality degree $k$. Within the learning algorithm, we can enforce cardinality sparsity by mapping the updated weight matrix of the hidden layer onto a cardinality-sparse matrix via a structured projection process. Specifically, for each column of the matrix, we first sort the values in ascending or descending order. Then, we partition these sorted values into $k$ distinct groups, where $k$ denotes the predetermined sparsity degree, controlling the level of sparsity in the resulting matrix. Subsequently, we compute the mean of the values within each partition and replace all original values in that partition with this mean, effectively making all values within each partition uniform. This projection ensures that the matrix retains a sparse structure while preserving essential information.

## 5 Sparsity in Tensor Regression

In this section, we first provide a brief review of tensors and tensor regression. Then, we study cardinality sparsity for tensor regression. For two tensors $\mathbb{X} \in \mathbb{R}^{I_1 \times \cdots \times I_K \times P_1 \cdots P_L}$ ( $I_1, \ldots, I_K, P_1, \ldots, P_L$ are positive integers) and $\mathbb{Y} \in \mathbb{R}^{P_1 \times \cdots \times P_L \times Q_1 \times \cdots \times Q_M}$ ($P_1, \ldots, P_L, Q_1, \ldots, Q_M$ are positive integers) the contracted tensor product

$$\langle \mathbb{X}, \mathbb{Y} \rangle_L \in \mathbb{R}^{I_1 \times \cdots \times I_K \times Q_1 \times \cdots \times Q_M},$$

can be defined as follows

$$(\langle \mathbb{X}, \mathbb{Y} \rangle_L)_{i_1, \ldots i_K, q_1, \ldots, q_M} :=$$
$$\sum_{p_1=1}^{P_1} \cdots \sum_{p_L=1}^{P_L} \mathbb{X}_{i_1, \ldots i_K, p_1, \ldots, p_L} \mathbb{Y}_{p_1, \ldots p_L, q_1, \ldots, q_M}.$$

In the special case for matrices $\boldsymbol{X} \in \mathbb{R}^{N \times P}$ and $\boldsymbol{Y} \in \mathbb{R}^{P \times Q}$, we have

$$\langle \boldsymbol{X}, \boldsymbol{Y} \rangle_1 := \boldsymbol{X}\boldsymbol{Y}.$$

Assume that we have $n = 1, \ldots, N$ observations, and consider predicting a tensor $\mathbb{Y} \in \mathbb{R}^{N \times Q_1 \times \cdots \times Q_M}$ from a tensor $\mathbb{X} \in \mathbb{R}^{N \times P_1 \times \cdots \times P_L}$ with the model

$$\mathbb{Y} := \langle \mathbb{X}, \mathbb{B} \rangle_L + \mathbb{U},$$

where $\mathbb{B} \in \mathbb{R}^{P_1 \times \cdots \times P_L \times Q_1 \times \cdots \times Q_M}$ is a coefficient array and $\mathbb{U} \in \mathbb{R}^{N \times Q_1 \times \cdots \times Q_M}$ is an error array. The least-squares solution is as follows

$$\widehat{\mathbb{B}} \in \arg\min_{\mathbb{B}} \|\mathbb{Y} - \langle \mathbb{X}, \mathbb{B} \rangle_L\|_F^2,$$

where $\|\cdot\|_F$ denotes the Frobenius norm. However, this solution is still prone to over-fitting. In order to address this problem an $L_2$ penalty on the coefficient tensor $\mathbb{B}$, could be applied

$$\widehat{\mathbb{B}} \in \arg\min_{\mathbb{B}} \|\mathbb{Y} - \langle \mathbb{X}, \mathbb{B} \rangle_L\|_F^2 + \lambda\|\mathbb{B}\|_F^2, \tag{12}$$

where $\lambda$ controls the degree of penalization. This objective is equivalent to ridge regression when $\mathbb{Y} \in \mathbb{R}^{N \times 1}$ is a vector and $\mathbb{X} \in \mathbb{R}^{N \times P}$ is a matrix. Similar to the neural networks we modify Equation (12) so that it is proper for cardinality sparsity as follows

$$\widehat{\mathbb{B}} \in \arg\min_{\mathbb{B}} \|\mathbb{Y} - \langle \mathbb{X}, \mathbb{B} \rangle_L\|_F^2 + \lambda\mathfrak{h}[\mathbb{B}], \tag{13}$$

and similar to the previous case we again define

$$\mathfrak{h}[\mathbb{B}] := \|\mathbb{B} \times_n \boldsymbol{G}\|_1 + \nu\|\Pi_{\boldsymbol{G}}\mathbb{B}\|_1, \tag{14}$$

where again $\boldsymbol{G}$ is any matrix, $\Pi_{\boldsymbol{G}}$ is the projection to $\ker(\boldsymbol{G})$ and $\nu$ is a constant. Then we have the below theorem for tensors.

**Theorem 2** (Prediction bound for tensors and boundedness of effective noise). *Consider the Equation (13). Let $\mathbb{U}_{\bullet, p_1, \cdots, p_l} \sim \mathcal{N}(0, \sigma^2)$. Assume effective noise $2\|\langle \mathbb{U}, \mathbb{X} \rangle_1\|_\infty$ is bounded, then for every $\mathbb{B} \in \mathbb{R}^{P_1 \times \cdots \times P_L \times Q_1 \times \cdots \times Q_M}$ it holds that*

$$\|\langle \mathbb{X}, \mathbb{B} \rangle_L - \langle \mathbb{X}, \widehat{\mathbb{B}} \rangle_L\|_F^2 \leq \inf_{\mathbb{A}} \left\{ \|\langle \mathbb{X}, \mathbb{B} \rangle_L - \langle \mathbb{X}, \mathbb{A} \rangle_L\|_F^2 + 2\lambda\mathfrak{h}[\mathbb{A}] \right\}$$

*Also let $\mathbb{U}_{\bullet, p_1, \cdots, p_l} \sim \mathcal{N}(0, \sigma^2)$. Then the effective noise $2\|\langle \mathbb{X}, \mathbb{U} \rangle_1\|_\infty$ is bounded with high probability, i.e.,*

$$\mathbb{P}\left\{ \lambda_t \geq 2\|\langle \mathbb{X}, \mathbb{U} \rangle_1\|_\infty \right\} \geq 1 - P_1 \cdots P_L Q_1 \cdots Q_M e^{-\left(\frac{\lambda_t}{2\sigma\sqrt{Ns}}\right)^2/2},$$

*where $s$ is defined as the follows*

$$s := \sup_{q_1 \in \{1, \ldots, Q_1\} \cdots q_M \in \{1, \ldots, Q_M\}} 2|\left(\langle \mathbb{X}, \mathbb{X} \rangle_1\right)_{q_1, \cdots, q_M, q_1, \cdots, q_M}|/N$$

Note that, for proving this theorem we just employed the norm property of $\mathfrak{h}[\mathbb{B}]$, which is irrelated to form of matrix $\boldsymbol{G}$. Therefore, we can immediately result the below corollaries.

**Corollary 3** (Prediction bound for tensors and boundedness of effective noise for $\ell_1$-regularizer). *Let $\boldsymbol{G} = \boldsymbol{0}$ then regularizer in (14) changes to $\ell_1$-regularizer and by assumptions of Theorem 2 it holds that*

$$\|\langle \mathbb{X}, \mathbb{B} \rangle_L - \langle \mathbb{X}, \widehat{\mathbb{B}} \rangle_L \|_F^2 \;\leq\; \inf_{\mathbb{A}} \left\{ \|\langle \mathbb{X}, \mathbb{B} \rangle_L - \langle \mathbb{X}, \mathbb{A} \rangle_L \|_F^2 + 2\lambda \mathfrak{h}[\mathbb{A}] \right\},$$

*and the effective noise $2\|\langle \mathbb{X}, \mathbb{U} \rangle_1 \|_\infty$ is bounded with high probability.*

**Corollary 4** (Prediction bound and boundedness of effective noise for cardinality sparsity). *Let $\boldsymbol{G} = \boldsymbol{A}$ then regularizer in (14) induces cardinality sparsity and by assumptions of Theorem 2 it holds that*

$$\|\langle \mathbb{X}, \mathbb{B} \rangle_L - \langle \mathbb{X}, \widehat{\mathbb{B}} \rangle_L \|_F^2 \leq \inf_{\mathbb{A}} \left\{ \|\langle \mathbb{X}, \mathbb{B} \rangle_L - \langle \mathbb{X}, \mathbb{A} \rangle_L \|_F^2 + 2\lambda \mathfrak{h}[\mathbb{A}] \right\},$$

*and the effective noise $2\|\langle \mathbb{X}, \mathbb{U} \rangle_1 \|_\infty$ is bounded with high probability.*

**Corollary 5.** *In the special case, for $\boldsymbol{Y} := \boldsymbol{X}\boldsymbol{\beta} + \boldsymbol{U}$; where $\boldsymbol{X} \in \mathbb{R}^{N \times P_1}, \boldsymbol{U} \in \mathbb{R}^{P_1 \times 1}$, the Theorem 2 would change to*

$$\mathbb{P}\left\{ \lambda_t \;\geq\; 2\|\langle \boldsymbol{X}, \boldsymbol{U} \rangle_1 \|_\infty \right\} \;\geq\; 1 - P_1 e^{-\left( \frac{\lambda_t}{2\sigma\sqrt{Ns}} \right)^2 /2},$$

*and $s$ changes to*

$$s \;:=\; \sup_{p_1 \in \{1,\ldots,P_1\}} 2|\left( \langle \boldsymbol{X}, \boldsymbol{X} \rangle_1 \right)_{p_1,p_1} |/N.$$

This result matches Lemma 4.2.1 of (Lederer, 2021).

# 6 Expremintal Support

In this section, we initially present experimental evidence demonstrating the advantages of sparsity in reducing multiplication and memory costs. Subsequently, we implement cardinality sparsity in machine learning systems. Our results underscore the substantial benefits of cardinality sparsity, demonstrating its advantageous impact.

## 6.1 Exprimental Support (Part I)

This section validates our results. First, we test the methods for matrix-matrix multiplication introduced in Section 3.1. We evaluate our approach for both simulated and real-world datasets.

### 6.1.1 Matrix-Matrix Multiplication

This part evaluates the performance of the proposed multiplication methods. We consider the matrix multiplication task $\boldsymbol{AB}$; $\boldsymbol{A} \in \mathbb{R}^{M \times P}$, $\boldsymbol{B} \in \mathbb{R}^{P \times N}$. We generate random matrices $\boldsymbol{A}, \boldsymbol{B}$ with varying degrees of sparsity, that is, varying number of unique elements of columns of $\boldsymbol{A}$ and rows of $\boldsymbol{B}$. Specifically, we populate the coordinates of the matrices with integers sampled uniformly from $\{1, \ldots, k\}$, where $k$ is the sparsity degree, and then subtract a random standard-normally generated value (the same for all coordinates) from all coordinates (we did this subtraction to obtain a general form for the matrix and avoid working with a matrix with just integer values). We set the sparsity degree to 10 and generated square matrices of different sizes. We then compared our method (Algorithm 3) with both the Strassen algorithm (Strassen, 1969) and the naive approach (Algorithm1) for matrix multiplication. The empirical results demonstrate large gains in speed especially for matrices of large size. As illustrated in Figure 2, our multiplication technique surpasses both Strassen's and the conventional algorithms when dealing handling large matrices. For smaller matrices, the preprocessing cost is relatively high. However, for larger matrices, the significant reduction in redundant multiplications makes our algorithm more efficient. Consequently, our algorithm outperforms those that do not optimize for redundant multiplications.

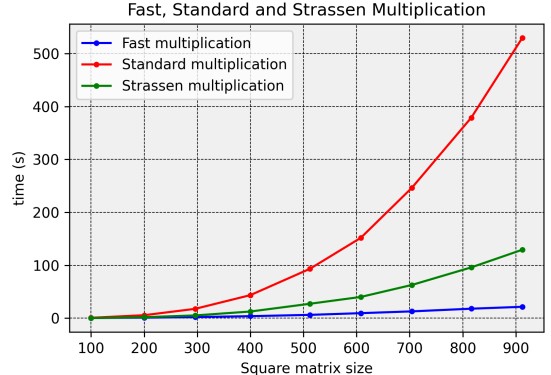 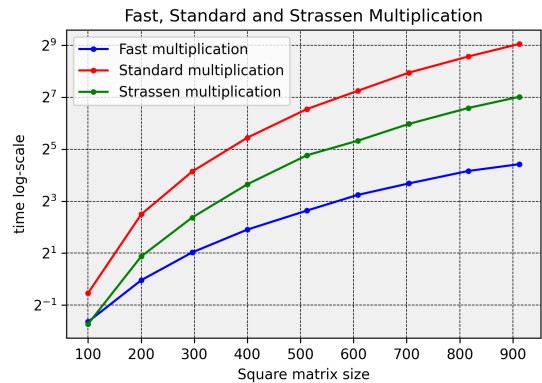

(a) Square nonbinary matrices multiplication without scaling.

(b) Square nonbinary matrices multiplication with logarithmic scaling.

Figure 2: Algorithm 3 compared to the naive Algorithm 1 and Strassen Algorithm for nonbinary matrices of sparsity degree equal to 10. Our algorithms are faster than the naive and Strassen approach, especially for large sizes.

### 6.1.2 Binary Matrix Multiplication

In this section, we will concentrate on the binary matrix multiplication of square matrices, implementing preprocessing similar to example 3. We compare our approach with the method described by Strassen (Strassen, 1969) and the standard multiplication technique. Similar to the previous part we randomly generated square matrices of sizes suitable for Strassen's algorithm. We utilized the algorithm referenced in 3 to perform our matrix multiplication. As shown in Figure 3, our multiplication method significantly outperforms both Strassen's and the standard algorithms.

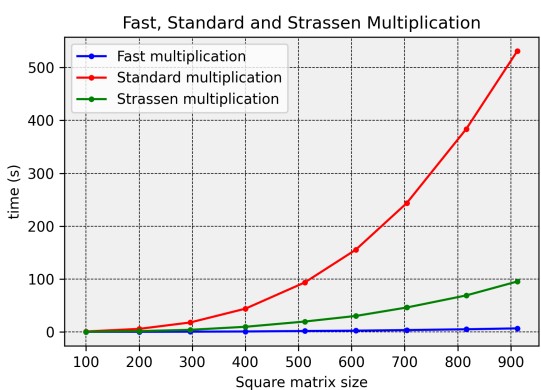 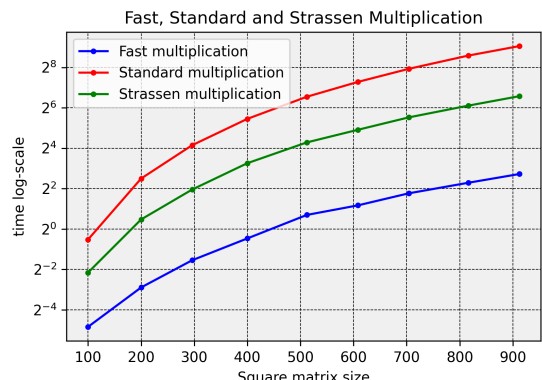

(a) Square binary matrices multiplication without scaling.

(b) Square binary matrices multiplication with logarithmic scaling.

Figure 3: Algorithm 3 compared to the naive Algorithm 1 and Strassen Algorithm for binary matrices. Our algorithms are considerably faster than the naive and Strassen approach.

### 6.2 Matrix Multiplcation for Real-world Datasets

In this section we will annalize cardinality sparsity for real-world datasets. We conducted additional experiments on real-world datasets to further validate the efficiency of our approach. Specifically, we applied our multiplication method to the real datasets (Leter Recognition Dataset, Letter Digits Dataset, and Firm

Teacher Clave Direction Classification (Slate, 1991; Alpaydin, 1998; Vurka, 2011)) and compared the results with the standard and Strassen multiplication method.

Matrix multiplication for real datasets commonly occurs in applications such as neural networks, where input data must be multiplied by weight matrices. To evaluate our method, we selected a dataset as input and multiplied it with randomly generated weight matrices. For example, the size of the input dataset in the Letter Digits is $1797 \times 64$. Using our matrix multiplication algorithm, we performed operations to multiply the input dataset ($1797 \times 64$) with weight matrices of size $64 \times n$. We varied the value of "$n$" to test our method under different conditions, simulating various numbers of columns (or number of nodes in neural networks). Subsequently, we measured the runtime, including preprocessing time.

Our results reveal a significant gap between the runtime of our method and that of standard matrix multiplication algorithms. We also applied our method to the "Firm Teacher Clave Direction Classification" dataset (Vurka, 2011), where dummy encoding is used. We demonstrated that datasets utilizing dummy encoding could achieve substantial performance improvements with our matrix multiplication algorithm. Since matrix multiplications are the primary computational workload in many applications like neural networks, optimizing them can significantly reduce energy consumption when processing real data.

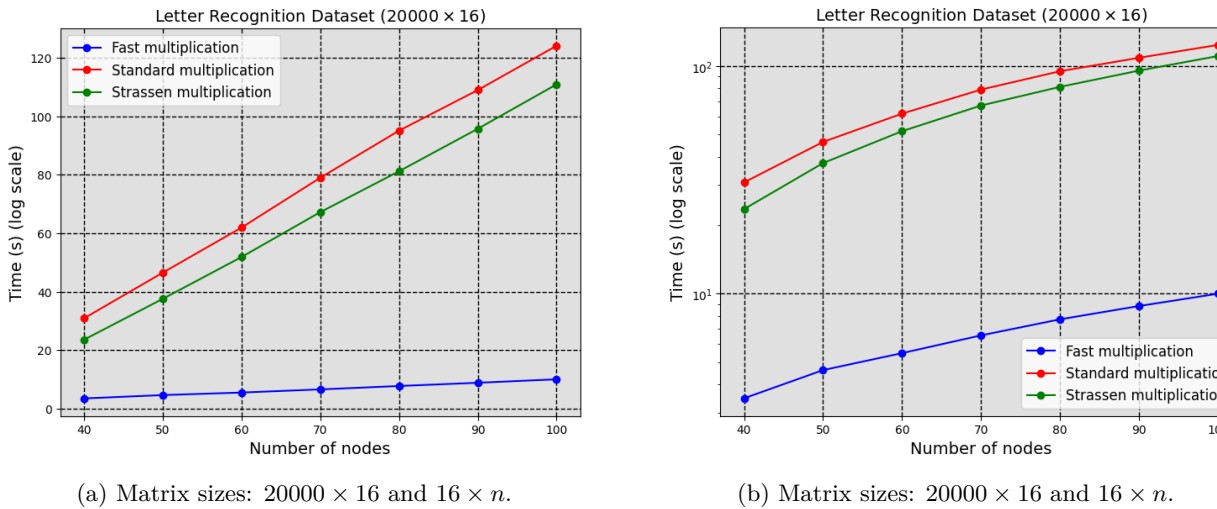

(a) Matrix sizes: $20000 \times 16$ and $16 \times n$.      (b) Matrix sizes: $20000 \times 16$ and $16 \times n$.

Figure 4: Panel (a) displays the normal scale, while Panel (b) shows the log scale. Our algorithms significantly outperform both the naive and Strassen methods in terms of speed for the Letter Recognition Dataset.

Moreover, we carried out more simulation experiments to evaluate our "binary multiplication algorithm" against some modern state-of-the-art methods. Note that many modern state-of-the-art algorithms are primarily theoretical, with no publicly available implementations. We selected two approaches that are more practical to implement—namely, the Four Russians algorithm and the bit-packed method—for direct comparison. As shown in Figure 7, our algorithm outperforms the Four Russians algorithm and achieves comparable performance to the bit-packed approach, which leverages CPU-level parallelism and low-level optimizations. This further demonstrates that our algorithm can significantly reduce CPU usage and hardware resource consumption.

We also conducted additional ablation experiments by varying the sparsity degree while keeping the matrix size fixed at $512 \times 80$ and $80 \times 512$. Specifically, we increased the cardinality (i.e., the number of unique elements per row or column) to observe its impact on computation speed. As expected, performance decreased with higher degrees of sparsity. Figure 8 illustrates the results.

### 6.2.1 Memory reduction using cardinality sparsity

In this part, we explore memory optimization through cardinality sparsity in real-world datasets. Since we can perform multiplication in the compressed domain, only the compressed version of the input data needs

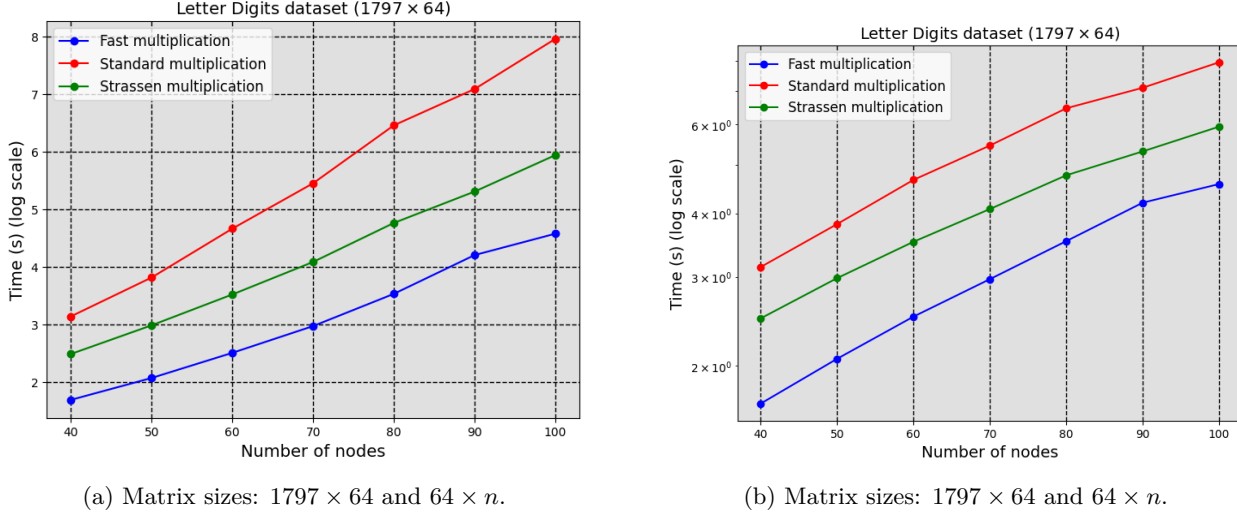

(a) Matrix sizes: $1797 \times 64$ and $64 \times n$.   (b) Matrix sizes: $1797 \times 64$ and $64 \times n$.

Figure 5: Panel (a) displays the normal scale, while Panel (b) shows the log scale. Our algorithms significantly outperform both the naive and Strassen methods in terms of speed for the Letter Digits Dataset.

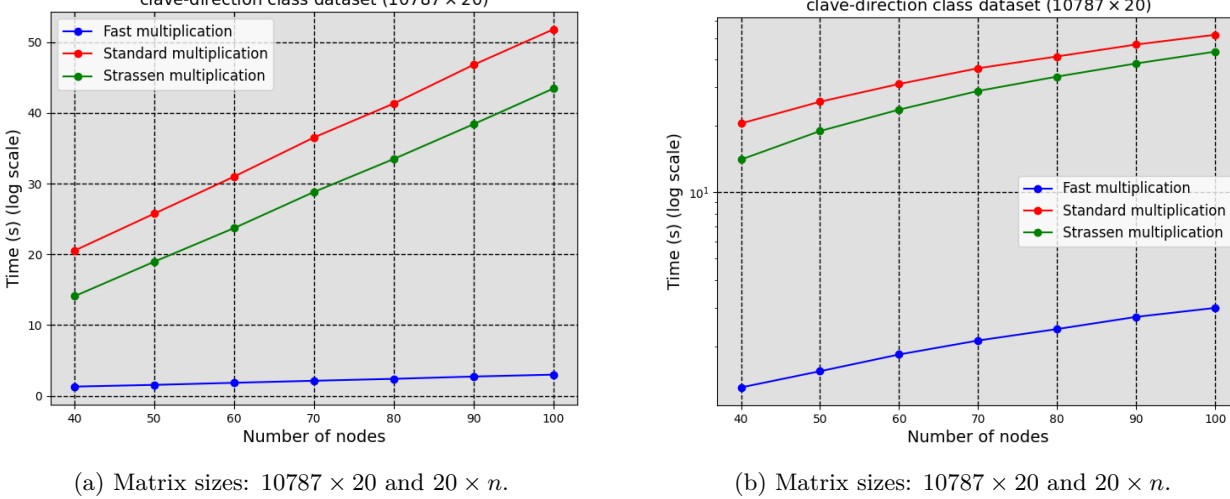

(a) Matrix sizes: $10787 \times 20$ and $20 \times n$.   (b) Matrix sizes: $10787 \times 20$ and $20 \times n$.

Figure 6: Panel (a) displays the normal scale, while Panel (b) shows the log scale. Our algorithms significantly outperform both the naive and Strassen methods in terms of speed for the Clave-direction Class Dataset.

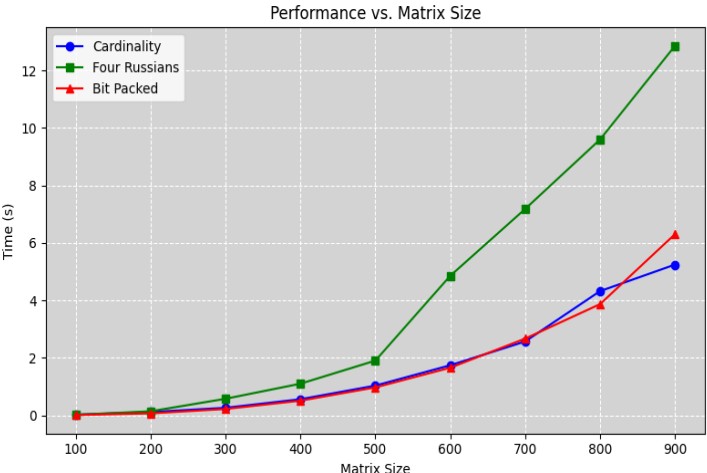

Figure 7: Runtime comparison of our binary multiplication algorithm with the Four Russians and bit-packed methods on square binary matrices of varying sizes.

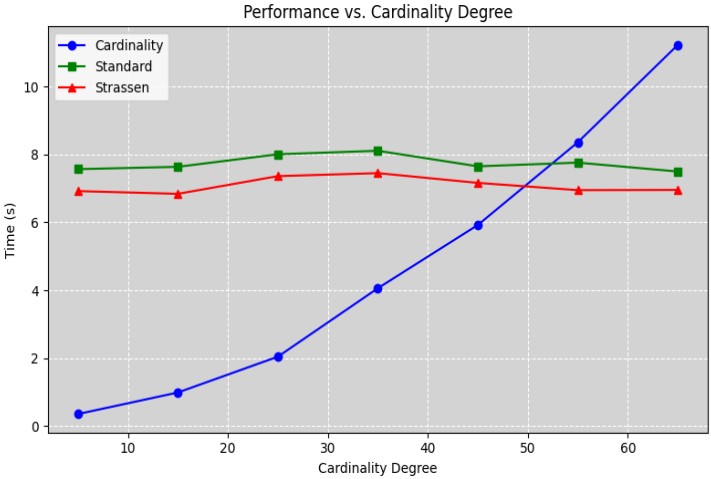

Figure 8: Computation time vs. cardinality for fixed $512 \times 80$ and $80 \times 512$ matrices. As the number of unique elements increases, traditional methods begin to outperform the cardinality sparsity approach.

to be stored, significantly reducing memory requirements. This is because we no longer need to store all the real numbers, which typically require 32 bits each. By eliminating the duplication of real numbers, we only need integers in the encoding matrix to reference the locations of these unique values. we provide examples using real datasets: the *Yeast* dataset (Nakai, 1991), the *Concrete Compressive Strength* dataset (Yeh, 1998), and the *AI4I 2020 Predictive Maintenance Dataset* (ai4, 2020), to demonstrate the efficiency of cardinality sparsity in reducing memory usage in neural networks. These datasets are referred to as data 1, data 2, and data 3, respectively, in the table.

## 6.3 Additional Empirical Results

We continue this section by presenting additional empirical results that further substantiate our methods and theories. In particular, we provide further experiments to investigate the performance of the matrix-matrix multiplication techniques introduced in Section 3.1. We then apply cardinality sparsity to neural networks and tensor regression. Finally, we illustrate the regularizer statistical benefits.

Table 1: We compared the memory requirements in bits with and without the use of cardinality sparsity in bytes. This table illustrates that implementing cardinality sparsity significantly reduces memory costs in a neural network.

| Input data | Using sparsity | Without sparsity |
|---|---|---|
| data 1 [1] | 9752.5 | 47488 |
| data 2 | 19662.8 | 37080 |
| data 3 | 125496.0 | 480000 |

Again consider the matrix multiplication task $AB$; $A \in \mathbb{R}^{M \times P}$, $B \in \mathbb{R}^{P \times N}$. We generate random matrices $A, B$ with varying degrees of sparsity, that is, varying number of unique elements of columns of $A$ and rows of $B$. Specifically, we populate the coordinates of the matrices with integers sampled uniformly from $\{1, \ldots, k\}$, where $k$ is the sparsity degree, and then subtract a random standard-normally generated value (the same for all coordinates) from all coordinates (we did this subtraction to obtain a general form for the matrix and avoid working with a matrix with just integer values). We then compare our methods (Algorithms 2 and 3) with the naive approach (Algorithm 1) for the corresponding matrix multiplication. The factor by which our methods improve on the naive approach's speed averaged over 10 times generating the matrices are presented in Figure 9. There are two cases: (i) $P \ll M, N$, where we use our Algorithm 3 (Figure 9 Panel a); and (ii) $P \gg M, N$, where we use our Algorithm 2 (Figure 9 Panel b). The empirical results demonstrate large gains in speed especially for small sparsity degrees.

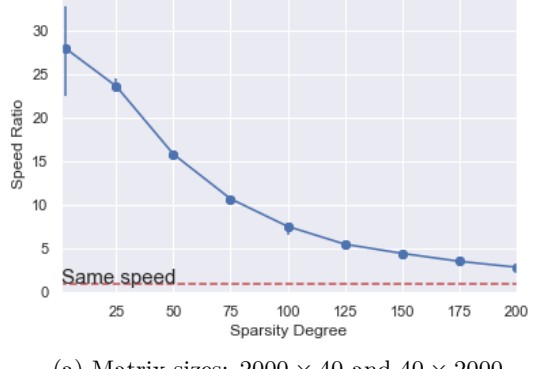

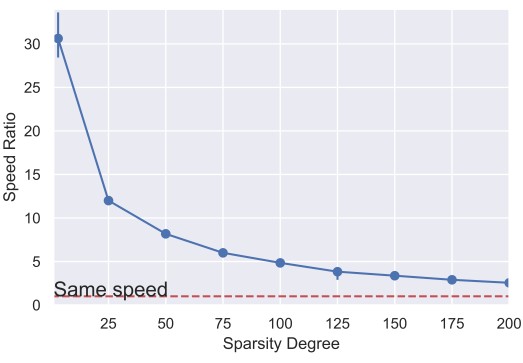

(a) Matrix sizes: $2000 \times 40$ and $40 \times 2000$.

(b) Matrix sizes: $100 \times 10000$ and $10000 \times 100$.

Figure 9: Algorithm 3 (Panel (a)) and Algorithm 2 (Panel (b)) compared to the naive Algorithm 1. Our algorithms are considerably faster than the naive approach, especially for small degrees of sparsity.

### 6.3.1 Cardinality Sparsity in Neural Networks and Tensor Regression

We study cardinality sparsity in machine learning systems using both real and simulated datasets. For the neural-networks application, we use the *Optical-Recognition-of-Handwritten-Digits* data (Dua & Graff, 2017). We train a two-layer relu network with 40 nodes in the hidden layer with gradient descent. For backpropagation, we use Algorithm 2; for forward operations; we use Algorithm 3. After each weight update, we project the weight matrix of the hidden layer onto a cardinality-sparse matrix: We sort and split each column into $k$ partitions, where $k$ is the sparsity degree. We then replace the old values by the mean of the values in each partition (see our Section 4.2.1), so that all values in each partition are equal after the projection. The accuracy of the neural networks is illustrated in Figure 10. This result shows that, cardinality sparsity training accelerates the parameter training substantially. It is important to understand that it is possible to conduct the experiment without using weight matrix projection. This is because the input data, which has a high dimension, is sparse, and most of the computational efficiency comes from

the sparsity of the input data. Although weight projection may further decrease computations, it's not mandatory.

We also investigated the accuracy versus time in two cases. As we observe in Figure 10, our method obtains high accuracy rates faster than the usual training methods.

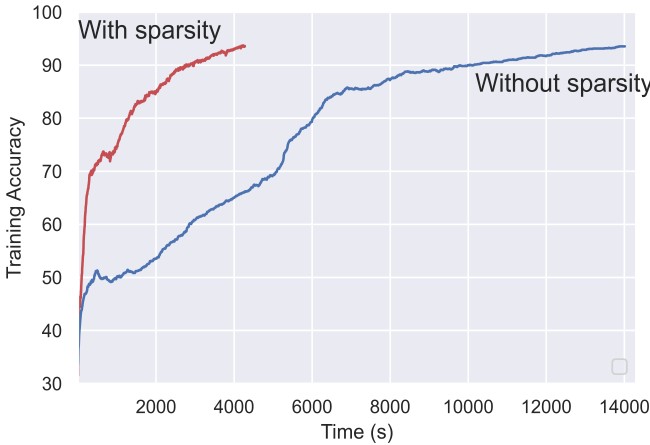

Figure 10: Training accuracy of the network trained with (red) and without (blue) cardinality sparsity as a function of training time. The cardinality sparsity accelerates the parameter training.

We also consider the tensor regression. We applied the `Tensorly` package for this simulation (Kossaifi et al., 2016). In order to employ our matrix-matrix multiplication method, we reshape the tensor into a matrix. We assumed that the input matrix is random and sparse in our sense (data are generated similar to section 6.1.1). The input is a tensor of size $1000 \times 16 \times 16$ and we applied 10 iterations for learning. Regression with sparse multiplication and standard multiplication takes 18.2 and 35 seconds respectively. Therefore, employing sparsity can lead to a faster regression process. Note that, in the tensor regression process we only substitute standard multiplication with sparse multiplication therefore, the accuracy rate is the same in both experiments.

Now we examine the performance of the regularizer and use experiments to certify the introduced regularizer can increase network performance. We consider least-squares minimization complemented by the regularizer which induces cardinality sparsity. We trained a 4 layers neural network where the number of the first, second, third, and fourth layer are equal to 2, 10, 15, and 20 respectively, where we employed *Inverse Square Root Unit ($x/\sqrt{1+x^2}$), identity, arctan,* and again *identity* as activation functions. The samples are generated by a standard normal distribution labeled by weight parameters which are sparse in our sense, plus a standard Gaussian noise. We set the target weights' sparsity equal to 4. We trained the network with 1000 iterations. As shown in Figure 11, incorporating a regularizer driven by cardinality sparsity can enhance regression performance compared to Lasso regression. We also computed the Frobenius norm of the difference between trained weights and the target weight. The difference in networks with and without regularization are equal to 64.24 and 76.12 respectively which is less value in a network with regularization as we expected. Note that all simulations are performed by Macbook Pro laptop with 8 cores of CPU and 16 Gigabytes of RAM.

To further assess the generalizability and practical benefits of our approach, we conducted experiments using the letter recognition dataset as input to a neural network. In this setting, we evaluated performance using the mean squared error as the primary metric. Our objective was to determine whether the advantages of cardinality sparsity—previously demonstrated in matrix multiplication and simulation tasks—would also extend to real-world machine learning workflows. The experimental results in Figure 12 indicate that incorporating cardinality sparsity not only preserves model accuracy but also leads to notable improvements in

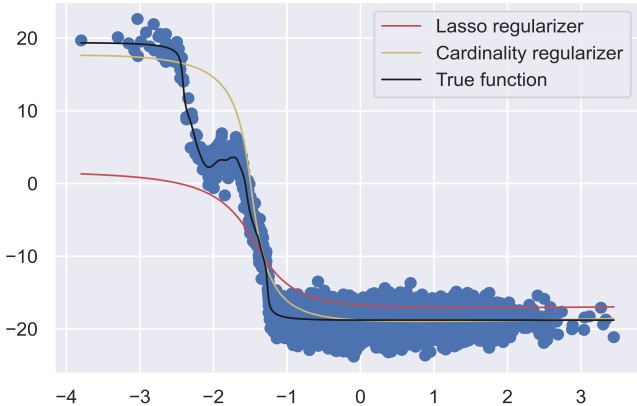

Figure 11: This figure compares deep learning with cardinality and standard sparsity. Integrating a regularizer based on cardinality sparsity can improve regression performance in comparison to Lasso regression. Cardinality sparsitiy (orange line) makes the vanilla estimator (red line) smooth and brings it closer to the true function (black line) especially in the first part of the domain.

computational speed. This suggests that the method is effective in accelerating neural network training and inference across different types of datasets.

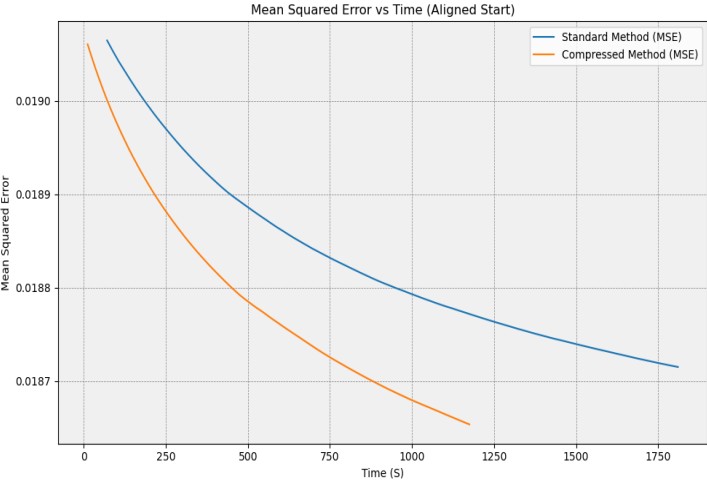

Figure 12: We also evaluated our method on the letter recognition dataset by feeding it into a neural network and measuring the mean squared error. The results show that cardinality sparsity can enhance processing speed across different datasets while maintaining performance.

### 6.4 Cardinality Sparsity Beyond Machine Learning

This section provides additional simulations to further illustrate the effectiveness of our approach, considering the setting where the matrix structure is assumed to be given. We emphasize that preprocessing time is always included in our simulations, and we never rely on the assumption that the matrix structure is known beforehand.

One of the most important cases where this assumption is natural is when we need to compute *powers of a matrix*. In this setting, preprocessing only needs to be done once, after which the encoded structure

can be reused for every multiplication. This property appears in many applications, including dynamical systems (Hirsch et al., 1974) and graph theory (West, 2001), making it a particularly strong use case for our *cardinality sparsity multiplication algorithm*, especially when the matrix is sparse in the cardinality sense.

For instance, in *graph theory*, the $k$-th power of an adjacency matrix reveals the number of paths of length $k$ (West, 2001). To test this, we generated a random adjacency matrix with 200 nodes and compared our cardinality sparsity multiplication algorithm against the standard dense multiplication. As shown in Figure 13, our algorithm achieved a much faster runtime.

Another important application where the structure of the matrix can be assumed fixed is in the *construction of multivariate Markov chain models*. Here, the transition matrix remains constant across iterations. We applied our algorithm in this context as well, and as illustrated in Figure 14, the cardinality sparsity multiplication method again performed very well, highlighting its usefulness beyond machine learning settings (Ching & Ng, 2006).

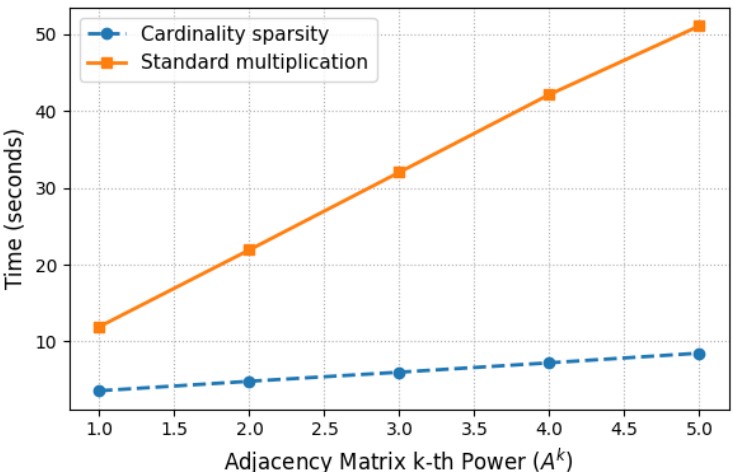

Figure 13: Comparison of cumulative runtime between the proposed cardinality sparsity multiplication algorithm and the standard dense matrix multiplication for successive powers of a random adjacency matrix with 200 nodes. The adjacency matrix is sparse in the cardinality sense, and the experiment demonstrates that the proposed method achieves significant computational savings.

## Acknowledgments

This research was partially funded by grant 520388526 (TRR391) by the Deutsche Forschungsgemeinschaft (DFG, German Research Foundation).

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

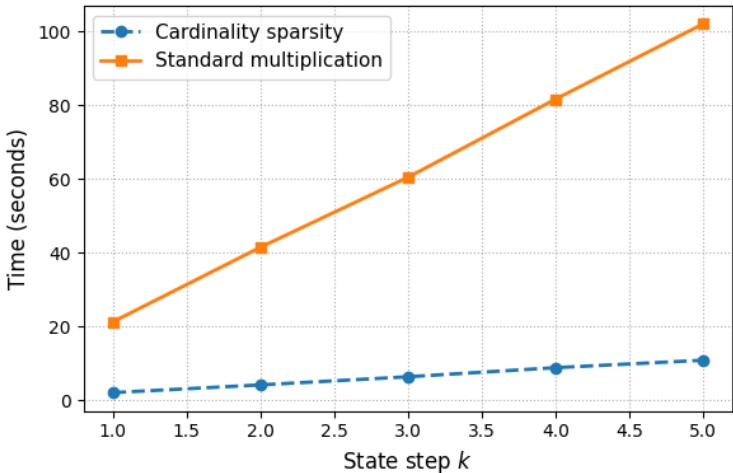

Figure 14: Application of the cardinality sparsity multiplication algorithm to a multivariate Markov chain model with a fixed transition matrix. The proposed method again outperforms the standard dense multiplication in terms of runtime.

S. Bakin. *Adaptive regression and model selection in data mining problems.* Ph.D. thesis, The Australian National University, Canberra, Australia, 1999.

A. Barron and J. Klusowski. Approximation and estimation for high-dimensional deep learning networks. *arXiv:1809.03090*, 2018.

P. Bartlett, P. Long, G. Lugosi, and A. Tsigler. Benign overfitting in linear regression. *Proc. Natl. Acad. Sci. USA*, 117(48):30063–30070, 2020.

B. Bauer and M. Kohler. On deep learning as a remedy for the curse of dimensionality in nonparametric regression. *Ann. Stat.*, 47(4):2261–2285, 2019.

A. Beknazaryan. Neural networks with superexpressive activations and integer weights. *Intell. Comput. Comput. Conf.*, pp. 445–451, 2022a.

A. Beknazaryan. Function approximation by deep neural networks with parameters $\{0, \pm\frac{1}{2}, \pm1, 2\}$. *J. Stat. Theory Pract.*, 2022b.

E. Bengio, P. Bacon, J. Pineau, and D. Precup. Conditional computation in neural networks for faster models. *arXiv:1511.06297*, 2015.

R. Bowen and C. Wang. *Introduction to Vectors and Tensors.* Courier Corporation, Chelmsford, Massachusetts, 2008.

E. Candès, J. Romberg, and T. Tao. Robust uncertainty principles: Exact signal reconstruction from highly incomplete frequency information. *IEEE Trans. Inf. Theory*, 52(2):489–509, 2006.

W. Ching and M. Ng. *Markov Chains: Models, Algorithms and Applications.* Springer, New York, 2006.

M. Christensen, J. Østergaard, and S. Jensen. On compressed sensing and its application to speech and audio signals. *Conf. Record 43rd Asilomar Conf. Signals, Systems, Comput.*, pp. 356–360, 2009.

G. Cybenko. Approximation by superpositions of a sigmoidal function. *Math. Control Signals Systems*, 2 (4):303–314, 1989.

L. Devroye, L. Györfi, and G. Lugosi. *A Probabilistic Theory of Pattern Recognition*, volume 31. Springer Sci. Bus. Media., New York, USA, 2013.

W. Dong, X. Li, L. Zhang, and G. Shi. Sparsity-based image denoising via dictionary learning and structural clustering. *CVPR*, pp. 457–464, 2011.

D. Dua and C. Graff. UCI machine learning repository. *University of California, Irvine, School of Information and Computer Sciences*, 2017. URL http://archive.ics.uci.edu/ml.

N. El Karoui. Operator norm consistent estimation of large-dimensional sparse covariance matrices. *IEEE Trans. Inf. Theory*, 36(6):2717–2756, 2008.

Y. Eldar and G. Kutyniok. Compressed sensing: theory and applications. *Cambridge University Press*, 2012.

A. Elgohary, M. Boehm, P. Haas, F. Reiss, and B. Reinwald. Compressed linear algebra for large-scale machine learning. *Proc. VLDB Endow.*, 9(12):960–971, 2016.

M. Fu, B. Zhao, C. Carignan, R. Shosted, J. Perry, D. P. Kuehn, Z. Liang, and B. Sutton. High-resolution dynamic speech imaging with joint low-rank and sparsity constraints. *Magn. Reson. Med.*, 73(5):1820–1832, 2015.

R. Geijn. Notes on vector and matrix norms. *Dept. of Computer Sci., Univ. of Texas at Austin*, 2014.

P. Golestaneh, M. Taheri, and J. Lederer. How many samples are needed to train a deep neural network? *arXiv:2405.16696*, 2024.

M. Hebiri and J. Lederer. Layer sparsity in neural networks. *arXiv:2006.15604*, 2020.

M. W. Hirsch, R. L. Devaney, and S. Smale. *Differential Equations, Dynamical Systems, and Linear Algebra*, volume 60. Academic Press, New York, USA, 1974.

A. Joshi. Perceptron and neural networks. *Machine Learn. Artif. Intell.*, pp. 57–72, 2022.

T. Kolda and B. Bader. Tensor decompositions and applications. *SIAM Rev.*, 51(3):455–500, 2009.

J. Kossaifi, Y. Panagakis, A. Anandkumar, and M. Pantic. Tensorly: Tensor learning in python. *arXiv:1610.09555*, 2016.

S. Land and J. Friedman. Variable fusion: A new adaptive signal regression method. *Tech. Rep. 656, Dept. Stat., Carnegie Mellon Univ.*, 1997.

F. Le Gall. Powers of tensors and fast matrix multiplication. *Proc. 39th Int. Symp. Symbolic Algebraic Comput.*, pp. 296–303, 2014.

J. Lederer. *Fundamentals of High-Dimensional Statistics*. Springer, New York, 2021.

J. Lederer. Statistical guarantees for sparse deep learning. *arXiv:2212.05427*, 2022.

L. Lee. Fast context-free grammar parsing requires fast boolean matrix multiplication. *J. ACM*, 49(1):1–15, 2002.

I. Lemhadri, F. Ruan, and R. Tibshirani. Lassonet: Neural networks with feature sparsity. *Int. Conf. Artif. Intell. Stat.*, pp. 10–18, 2021.

C. McDiarmid. On the method of bounded differences. *Surv. Comb.*, 141(1):148–188, 1989.

A. Mohaddes and J. Lederer. Affine invariance in continuous-domain convolutional neural networks. *arXiv:2311.09245*, 2023.

Mohamad M. Mohades, Ali Mohades, and Aliakbar Tadaion. A Reed–Solomon Code Based Measurement Matrix with Small Coherence. *IEEE Signal Processing Letters*, 21(7):839–843, Jul 2014.

Kenta Nakai. Yeast. UCI Machine Learning Repository, 1991. DOI: https://doi.org/10.24432/C5KG68.

B. Neyshabur, R. Tomioka, and N. Srebro. Norm-based capacity control in neural networks. In *Proceedings of the 28th Conference on Learning Theory*, pp. 1376–1401, Jun 2015.

C. Papadimitriou. Computational complexity. *Encycl. Comput. Sci.*, pp. 260–265, 2003.

S. Ravishankar, J. Ye, and J. Fessler. Image reconstruction: From sparsity to data-adaptive methods and machine learning. *Proc. IEEE*, 108(1):86–109, 2019.

J. Schmidt-Hieber. Nonparametric regression using deep neural networks with relu activation function. *Annals of Statistics*, 48(4):1875–1897, 2020.

D. Slate. Letter recognition. UCI Machine Learning Repository, 1991. URL https://archive.ics. uci.edu/ml/datasets/letter+recognition.

N. Srivastava, G. Hinton, A. Krizhevsky, I. Sutskever, and R. Salakhutdinov. Dropout: a simple way to prevent neural networks from overfitting. *J. Mach. Learn. Res.*, 15(1):1929–1958, 2014.

V. Strassen. Gaussian elimination is not optimal. *Numer. Math.*, 13(4):354–356, 1969.

W. Sun and L. Li. Store: sparse tensor response regression and neuroimaging analysis. *J. Mach. Learn. Res.*, 18(1):4908–4944, 2017.

M. Taheri, N. Lim, and J. Lederer. Efficient feature selection with large and high-dimensional data. *arXiv:1609.07195*, 2020.

M. Taheri, F. Xie, and J. Lederer. Statistical guarantees for regularized neural networks. *Neural Netw.*, 142: 148–161, 2021.

C. Thrampoulidis, S. Oymak, and B. Hassibi. Regularized linear regression: A precise analysis of the estimation error. In *Proceedings of the 28th Conference on Learning Theory*, pp. 1683–1709, Jun 2015.

R. Tibshirani, M. Saunders, S. Rosset, J. Zhu, and K. Knight. Sparsity and smoothness via the fused lasso. *J. R. Stat. Soc. Ser. B*, 67(1):91–108, 2005.

A. Vaart and J. Wellner. Weak convergence. In *Weak Convergence and Empirical Processes*, pp. 16–28. Springer, New York, USA, 1996.

S. van de Geer. *Empirical Processes in M-Estimation*, volume 6. Cambridge Univ. Press, Cambridge, UK, 2000.

R. Vershynin. *High-Dimensional Probability: An Introduction with Applications in Data Science*, volume 47. Cambridge Univ. Press, Cambridge, UK, 2018.

M. Vurka. Firm-teacher clave-direction classification. UCI Machine Learning Repository, 2011. URL https://archive.ics.uci.edu/ml/datasets/firm-teacher_clave-direction_ classification.

W. Wen, C. Wu, Y. Wang, Y. Chen, and H. Li. Learning structured sparsity in deep neural networks. *Adv. Neural Inf. Process. Syst.*, 29:2074–2082, 2016.

D. West. *Introduction to Graph Theory*. Prentice Hall, New Jersey, 2001.

C. Xu and Z. Zhang. Random tensors and their normal distributions. *arXiv:1908.01131*, 2019.

C. Yeh. Concrete Compressive Strength. UCI Machine Learning Repository, 1998. DOI: https://doi.org/10.24432/C5PK67.

X. Ying. An overview of overfitting and its solutions. *J. Phys. Conf. Ser.*, 1168:022022, 2019.

Huacheng Yu. An improved combinatorial algorithm for boolean matrix multiplication. In *International Colloquium on Automata, Languages, and Programming (ICALP)*, volume 9134 of *Lecture Notes in Computer Science*, pp. 1094–1105, 2015.

Y. Zhang, J. Lee, and M. Jordan. L1-regularized neural networks are improperly learnable in polynomial time. In *Proceedings of the 33rd International Conference on Machine Learning (ICML)*, pp. 993–1001, Jun 2016.

M. Zhu, T. Zhang, Z. Gu, and Y. Xie. Sparse tensor core: Algorithm and hardware co-design for vector-wise sparse neural networks on modern gpus. *Proc. IEEE/ACM Int. Symp. Microarch.*, pp. 359–371, 2019.

# A  Appendix

In this section, we first explore matrix-matrix multiplication algorithms. We then broaden our concept of sparsity. Subsequently, we provide additional technical results. The implementation of our main algorithm (Algorithm 3) is provided in the final section.

## A.1  Algorithms

In this section we review matrix-matrix multiplication algorithms.

---

**Algorithm 1** Standard Multiplication

---

**Require:** $\boldsymbol{A} \in \mathbb{R}^{M \times P}; \boldsymbol{B} \in \mathbb{R}^{P \times N}$
**Ensure:** $\boldsymbol{AB}$

    **function** STANDARDMULTIPLICATION($\boldsymbol{A}, \boldsymbol{B}$)
        **for** $i \leftarrow 1$ to $M$ **do**
            **for** $j \leftarrow 1$ to $N$ **do**
                Output$[i, j] \leftarrow 0$
                **for** $k \leftarrow 1$ to $P$ **do**
                    Output$[i, j] \mathrel{+}= \boldsymbol{A}[i, k] * \boldsymbol{B}[k, j]$
                **end for**
            **end for**
        **end for**
        **return** Output
    **end function**

---

**Algorithm 2** Multiplication by Cardinality Sparsity ($P > M, N$)

---

**Require:** $\boldsymbol{W} \in \mathbb{R}^{M \times P}; \boldsymbol{V} \in \mathbb{R}^{P \times N}$
**Ensure:** $\boldsymbol{WV}$

1:  **function** SPARSEMULTIPLICATION($\boldsymbol{W}, \boldsymbol{V}$)
2:     **for** $j \leftarrow 1$ to $P$ **do**
3:        $C_{\boldsymbol{W}^{(j)}} \leftarrow$ unique elements of column $\boldsymbol{W}^{(j)}$ ($[w_{1,j} \ \ldots \ w_{m,j}]^{\top}$)
4:        $\boldsymbol{I}[w_{i,j}] \leftarrow i; \ \ i \in \{1, \cdots, m\}$
5:     **end for**
6:     **for** $k \leftarrow 1$ to $M$ **do**
7:        **for** $j \leftarrow 1$ to $N$ **do**
8:            **for** $i \leftarrow 1$ to $P$ **do**
9:               Output$[k, j] \leftarrow \text{sum}(\text{sum}(\boldsymbol{V}[\boldsymbol{I}[k, :] == i, j]) * C_{\boldsymbol{W}}[k, i])$
10:          **end for**
11:      **end for**
12:     **end for**
13:     **return** Output
14: **end function**

---

---

**Algorithm 3** Multiplication by Cardinality Sparsity ($P < M, N$)

---

**Require:** $\boldsymbol{W} \in \mathbb{R}^{M \times P}; \boldsymbol{V} \in \mathbb{R}^{P \times N}$
**Ensure:** $\boldsymbol{WV}$

    **function** SPARSEMULTIPLICATION($\boldsymbol{W}, \boldsymbol{V}$)
        **for** $j \leftarrow 1$ to $P$ **do**
            $C_{\boldsymbol{W}^{(j)}} \leftarrow$ unique elements of column $\boldsymbol{W}^{(j)}$ ($[w_{1,j} \ \ldots \ w_{m,j}]^\top$)
            $\boldsymbol{I}[w_{i,j}] \leftarrow i; \ \ i \in \{1, \cdots, m\}$
        **end for**
        **for** $j \leftarrow 1$ to $P$ **do**
            $C_{\boldsymbol{V}_{(j)}} \leftarrow$ unique elements of row $\boldsymbol{V}_{(j)}$ ($[v_{j,1} \ \ldots \ v_{j,n}]$)
            $\boldsymbol{I}[v_{j,i}] \leftarrow i; \ \ i \in \{1, \cdots, n\}$
        **end for**
        **for** $j \leftarrow 1$ to $P$ **do**
            $\boldsymbol{D}_j \leftarrow C_{\boldsymbol{W}^{(j)}} \otimes C_{\boldsymbol{V}_{(j)}}$
            **for** $i \leftarrow 1$ to $m$ **do**
                **for** $k \leftarrow 1$ to $n$ **do**
                    Auxiliary$[\boldsymbol{I}[: \ , \ j] == i, \boldsymbol{J}[j \ , \ :] == k] \leftarrow \boldsymbol{D}_j[i, k]$
                **end for**
            **end for**
            Output $\leftarrow$ Output + Auxiliary
        **end for**
        **return** Output
    **end function**

---

## A.2 Extending Our Notion of Sparsity

Now we recall some notations and definitions. We then provide a generalized setting for our notion of sparsity.

### A.2.1 Some Notations and Definitions

We recall that

$$M \ := \ M_{\boldsymbol{G}} \ := \ \max_j \|\bar{\boldsymbol{G}}^{j^{-1}}\|_{1 \to 1}, \tag{15}$$

where $\bar{\boldsymbol{G}}^j$ is the invertible matrix induced by $\boldsymbol{G}^j$. We also define below auxiliary parameter for our computations

$$b \ := \ \max \{M, 1\}. \tag{16}$$

Furthermore, for the sake of simplicity from now on, we depict $\Pi_{\boldsymbol{G}}\Theta$ and $\Pi_{\boldsymbol{G}}\boldsymbol{W}$ by $\widetilde{\Theta}$ and $\widetilde{\boldsymbol{W}}$ respectively. Indeed

$$\Theta \ := \ \widetilde{\Theta} + \overline{\Theta} \ := \ (\widetilde{\boldsymbol{W}}^L, \ldots, \widetilde{\boldsymbol{W}}^0) + (\overline{\boldsymbol{W}}^L, \ldots, \overline{\boldsymbol{W}}^0), \tag{17}$$

where $\widetilde{\Theta} \in \ker(\boldsymbol{G})$ and $\overline{\Theta}$ belong to the space which is orthogonal to the $\ker(\boldsymbol{G})$. Subsequently, for each $l \in \{0, \ldots, L\}$, we decompose each weight matrix $\boldsymbol{W}^l$ according to the kernel of $\boldsymbol{G}^l$ denoted by $\ker(\boldsymbol{G}^l)$. Specifically, we can write $\boldsymbol{W}^l := \widetilde{\boldsymbol{W}}^l + \overline{\boldsymbol{W}}^l$, where all columns of $\widetilde{\boldsymbol{W}}^l$ belong to $\ker(\boldsymbol{G}^l)$ and all columns of $\overline{\boldsymbol{W}}^l$ are within the space, which is orthogonal to $\ker(\boldsymbol{G}^l)$ denoted by $\ker(\boldsymbol{G}^l)^\perp$. This implies that $\boldsymbol{W}'^l = \boldsymbol{G}^l\boldsymbol{W}^l = \boldsymbol{G}^l\overline{\boldsymbol{W}} + \boldsymbol{G}^l\widetilde{\boldsymbol{W}} = \boldsymbol{G}^l\overline{\boldsymbol{W}}$. We can also rewrite (7) like the below:

$$\mathfrak{h}[\overline{\Theta}, \widetilde{\Theta}] \ := \ \|\boldsymbol{G}\overline{\Theta}\|_1 + \nu\|\widetilde{\Theta}\|_1.$$

Employing Equation (3), equivalent to (7) we can write the penalty term as

$$\mathfrak{h}[\Theta] \; := \; \sum_{i=0}^{L}\sum_{k=1}^{p_{l+1}}\sum_{j=1}^{p_l}\left|(\boldsymbol{G}^l\boldsymbol{W}^l)_{kj}\right| + \sum_{i=0}^{L}\sum_{k=1}^{p_{l+1}}\sum_{j=1}^{p_l}\left|\widetilde{\boldsymbol{W}}_{kj}^l\right|. \tag{18}$$

Note that, we could also study the performance of regularizer in neural networks away with the scaling parameter (see (Lederer, 2022)). In the next part, we provide a generalization of our sparsity concept.

### A.2.2 Sparsity Notion Extension

In a more general setting, we can fix a partition for the weight matrix so that elements with the same value are in the same group. Indeed, we assume that in the sparsest case each partition contains one unique row. While in the previous case there is one individual row in the entire matrix in the sparsest case. Indeed, the generalized setting concerns block sparsity, while in the normal case, the entire matrix is considered as one single block. The group sparsity concept can be employed to study this setting. In this section, we apply the regularizer based on the penalty term defined in equation 7 to the analog of group sparsity.

First of all, for $l \in \{0,\dots,L\}$, we let $\mathcal{I} := \{\mathcal{B}_1^l,\dots,\mathcal{B}_{d_l}^l\}$ be a partition of $\{1,\dots,p_{l+1}\}$, where $d_l \in \{1,2,\dots\}$ is the number of partitions within $\mathcal{I}$. We further write the network weights row-wise

$$\boldsymbol{W}^l \; := \; \left(\left(\boldsymbol{W}_1^l\right)^\top \dots \left(\boldsymbol{W}_{p_{l+1}}^l\right)^\top\right)^\top,$$

and define

$$\boldsymbol{W}_{\mathcal{B}_i^l}^l \; := \; \left(\left(\boldsymbol{W}_{j_1}^l\right)^\top \dots \left(\boldsymbol{W}_{j_{|\mathcal{B}_i^l|}}^l\right)^\top\right)^\top \qquad i \in \{1,\dots,d_l\},$$

where $\{j_1,\dots,j_{|\mathcal{B}_i^l|}\} = \mathcal{B}_i^l$ and $|\mathcal{B}_i^l|$ is the cardinality of $\mathcal{B}_i^l$ for all $l \in \{0,\dots,L\}$. As before, there exist matrices $\boldsymbol{A}_i^j \in \mathbb{R}^{\binom{|\mathcal{B}_i^j|}{2}\times|\mathcal{B}_i^j|}$, such that $\boldsymbol{A}_i^j\boldsymbol{W}_{\mathcal{B}_i^j}$ is a matrix whose rows are row-wise subtractions of elements of $\boldsymbol{W}_{\mathcal{B}_i^j}$. Let $\boldsymbol{A}^j$ be the matrix constructed by these sub-matrices in such a way that

$$\|\boldsymbol{A}^j\boldsymbol{W}^j\|_1 \; = \; \sum_{i=1}^{d_j}\left\|\sqrt{P_{i,j}}\,\boldsymbol{A}_i^j\boldsymbol{W}_{\mathcal{B}_i^j}\right\|_1 \; = \; \sum_{i=1}^{d_j}\sum_{s,t\in\mathcal{B}_i^j}\frac{\sqrt{P_{i,j}}}{2}\|\boldsymbol{W}_s - \boldsymbol{W}_t\|_1,$$

where $\boldsymbol{A}^j : \mathcal{M}_{p_{j+1}\times p_j} \to \mathcal{M}_{l_j\times p_j}$ and $\mathcal{M}_{m\times n}$ denotes the space of $m \times n$ matrices. Moreover, $P_{i,j}$, is the number of parameters in group $i$, where $l_j$ is calculated from summing over all 2-combinations of the cardinality of partitions, i.e.

$$l_j \; := \; \sum_{i=1}^{d_j}\frac{|\mathcal{B}_i^j| \times (|\mathcal{B}_i^j| - 1)}{2}.$$

In order to clarify more consider the below example.

**Example 4.** *Let $\mathcal{B}_1^1 = \{1,2,3\}, \mathcal{B}_2^1 = \{4,5\}$ and let $\boldsymbol{W}^j \in \mathbb{R}^{5\times 5}$ then $P_{1,j} = 15$ and $P_{2,j} = 10$ and $\boldsymbol{A}^j\boldsymbol{W}^j$ is as follows*

$$
\begin{bmatrix}
\sqrt{15} & -\sqrt{15} & 0 & 0 & 0 \\
0 & \sqrt{15} & -\sqrt{15} & 0 & 0 \\
-\sqrt{15} & 0 & \sqrt{15} & 0 & 0 \\
0 & 0 & 0 & \sqrt{10} & -\sqrt{10}
\end{bmatrix}
$$

$$
\times
\begin{bmatrix}
w_{11} & w_{12} & w_{13} & w_{14} & w_{15} \\
w_{21} & w_{22} & w_{23} & w_{24} & w_{25} \\
w_{31} & w_{32} & w_{33} & w_{34} & w_{35} \\
w_{41} & w_{42} & w_{43} & w_{44} & w_{45} \\
w_{51} & w_{52} & w_{53} & w_{54} & w_{55}
\end{bmatrix}
$$

$$
=
\begin{bmatrix}
\sqrt{15}(\boldsymbol{w}_1 - \boldsymbol{w}_2) \\
\sqrt{15}(\boldsymbol{w}_2 - \boldsymbol{w}_3) \\
\sqrt{15}(\boldsymbol{w}_1 - \boldsymbol{w}_3) \\
\sqrt{10}(\boldsymbol{w}_4 - \boldsymbol{w}_5)
\end{bmatrix},
$$

where $\boldsymbol{w}_1, \cdots, \boldsymbol{w}_5$ denote the rows of the second matrix. In addition, we have

$$
\boldsymbol{A}_1^j := \begin{bmatrix}
\sqrt{15} & -\sqrt{15} & 0 \\
0 & \sqrt{15} & -\sqrt{15} \\
-\sqrt{15} & 0 & \sqrt{15}
\end{bmatrix} ; \boldsymbol{A}_2^j := \begin{bmatrix} \sqrt{10} & -\sqrt{10} \end{bmatrix} .
$$

Note that the computations for this setting are similar to the previous case, where we considered a non-invertible matrix $\boldsymbol{G}$ for our analysis in the last Section. The only change is related to the computation of the upper bound for the parameter $M_{\boldsymbol{A}}$, which is addressed in the next corollary.

**Corollary 6.** *For the matrix $\boldsymbol{A}$ with generalized cardinality sparsity the value of $M_{\boldsymbol{A}}$ is bounded as the below*

$$
M_{\boldsymbol{A}} := \sup_j \|\boldsymbol{A}^{j^{-1}}\|_{1 \to 1} \leq \sup_j \sup_i \|\boldsymbol{A}_i^{j^{\dagger}}\|_{1 \to 1} =
$$

$$
\sup_j \sup_i \frac{2}{\sqrt{P_{i,j}}|\mathcal{B}_i^j|}.
$$

### A.3 Technical Results

In this part we provide some technical results which are required to prove the results of this paper.

#### Convexity

We first show that $\mathfrak{h}$ defined in (7) is a norm and $\mathcal{A}_{\mathfrak{h}}$ which is defined as the below is a convex set.

$$
\mathcal{A}_{\mathfrak{h}} := \{\Theta | \mathfrak{h}(\Theta) \leq 1\}. \tag{19}
$$

**Proposition 1** (Convexity). *$\mathcal{A}_{\mathfrak{h}}$ defined in above is a convex set and $\mathfrak{h}$ defined in (7) is a norm.*

#### Lipschitz Property

The Lipschitz property can be employed to show the boundedness of networks over typical sets that are derived from the presented regularization scheme. The Lipschitz property of neural networks on $\mathcal{A}_{\mathfrak{h}}$ can be stated as follows.

**Theorem 3** (Lipschitz property on $\mathcal{A}_{\mathfrak{h}}$). *if $\Omega, \Gamma \in \mathcal{A}_{\mathfrak{h}}$ and activation functions are $a_{\text{Lip}}$-Lipchitz then we get*

$$
\|\mathfrak{g}_{\Omega} - \mathfrak{g}_{\Gamma}\|_n \leq 2(a_{\text{Lip}})^L \sqrt{L} \|\boldsymbol{x}\|_n \left( \frac{1}{L}(M + \frac{1}{\nu}) \right)^L \|\Omega - \Gamma\|_F, \tag{20}
$$

where we set $M = M_{\boldsymbol{G}} = \max_j \|\bar{\boldsymbol{G}}^{j^{-1}}\|_{1\to1}$. $\bar{\boldsymbol{G}}^j$ is a matrix induced by $\boldsymbol{G}^j$ and $L$ is the number of hidden layers. We can rewrite (20) as follows

$$\|\mathfrak{g}_\Omega - \mathfrak{g}_\Gamma\|_n \leq c_{\mathrm{Lip}}\|\Omega - \Gamma\|_F,$$

where we define

$$c_{\mathrm{Lip}} := 2(a_{\mathrm{Lip}})^L\sqrt{L}\|\boldsymbol{x}\|_n\left(\frac{2}{L}\left(M + \frac{1}{\nu}\right)\right)^L.$$

The above theorem results in boundedness on $\mathcal{A}_\mathfrak{h}$. This property proves useful for bounding the quantiles of empirical processes. Specifically, it helps demonstrate that the networks are Lipschitz continuous and bounded over typical sets defined by our regularization strategy ($\mathcal{A}_\mathfrak{h}$). The following lemma formalizes this result.

**Theorem 4** (Boundedness on $\mathcal{A}_\mathfrak{h}$)**.** *The set* $(\{\mathfrak{g}_\Omega | \Omega \in \mathcal{A}_\mathfrak{h}\}, \|\cdot\|_n)$ *is bounded.*

### Dudley Integral

To analyze the complexity characteristics, we introduce the covering number $\mathcal{N}(r, \mathscr{A}, \|\cdot\|)$ and define the corresponding entropy as

$$H(r, \mathscr{A}, \|\cdot\|) := \log\mathcal{N}(r, \mathscr{A}, \|\cdot\|),$$

Where $\mathcal{N}(r, \mathscr{A}, \|\cdot\|)$ is the covering number, $r \in (0, \infty)$, and $\|\cdot\|$ is a norm on an space $\mathscr{A}$ ((Vaart & Wellner, 1996) Page 98). These quantities are used to define a complexity measure for a class of neural networks $\mathcal{G}_h := \{g_\Omega : \Omega \in \mathcal{A}_h\}$.

We also define the Dudley integration of the collection of networks $\mathcal{G}_\mathfrak{h} := \{\mathfrak{g}_\Omega : \Omega \in \mathcal{A}_\mathfrak{h}\}$ by

$$J(\delta, \sigma, \mathcal{A}_\mathfrak{h}) := \int_{\delta/(8\sigma)}^{\infty} H^{1/2}(r, \mathcal{G}_\mathfrak{h}, \|\cdot\|_n)\,dr,$$

for $\delta, \sigma \in (0, \infty)$ ((van de Geer, 2000) Section 3.3). Dudley integration can be used to bound the complexity of the corresponding neural network class. Now we try to provide the elements which are needed to find $\lambda_{\mathfrak{h},t}$ in Theorem 2 of (Taheri et al., 2021). For this purpose, we apply Lemma 10 from (Taheri et al., 2021) which enables us to bound entropies over the parameter space rather than the network space. Moreover, we think of $\mathcal{A}_\mathfrak{h}$ as a set in $\mathbb{R}^P$ and we obtain

$$H(r, \mathcal{G}_\mathfrak{h}, \|\cdot\|_n) \leq H\left(\frac{r}{2(a_{\mathrm{Lip}})^L\sqrt{L}\left(\frac{2}{L}\left(M + \frac{1}{\nu}\right)\right)^L\|\boldsymbol{x}\|_n}, \mathcal{A}_\mathfrak{h}, \|\cdot\|_F\right)$$

$$= H\left(\frac{r}{2(a_{\mathrm{Lip}})^L\sqrt{L}\left(\frac{2}{L}\left(M + \frac{1}{\nu}\right)\right)^L\|\boldsymbol{x}\|_n}, \mathcal{A}_\mathfrak{h} \subset \mathbb{R}^P, \|\cdot\|_2\right).$$

We also need the below auxiliary Lemma which relates $\mathcal{A}_\mathfrak{h}$ and the $l_1-$sphere.

**Lemma 2** (Relation between $\mathcal{A}_\mathfrak{h}$ and $l_1-$sphere of radius $b$)**.** *We have* $\mathcal{A}_\mathfrak{h} \subset B_1(b)$*, where* $B_1(b)$ *is the* $l_1 -$ sphere *of radius* $b$*.*

Bounding the Dudley entropy integral is a key step toward establishing an upper bound on the Rademacher complexity. In the next step, we provide an upper bound for the Dudley integration.

**Lemma 3** (Entropy and Dudley integration upper bound)**.** *Assume that the activation functions* $\mathfrak{f}^l : \mathbb{R}^{p_l} \to \mathbb{R}^{p_l}$ *are* $a_{\mathrm{Lip}}$*-Lipschitz continuous with respect to the Euclidean norms on their input and output spaces. Then, it holds for every* $r \in (0, \infty)$ *and* $\delta, \sigma \in (0, \infty)$ *which satisfy* $\delta \leq 8\sigma R$ *that*

$$H\left(\frac{r}{2(a_{\mathrm{Lip}})^L\sqrt{L}\left(\frac{2}{L}\left(M + \frac{1}{\nu}\right)\right)^L\|\boldsymbol{x}\|_n}, \mathcal{A}_\mathfrak{h} \subset \mathbb{R}^P, \|\cdot\|_2\right) \leq$$
$$\frac{24b^2c_{\mathrm{Lip}}^2}{r^2}\log(\frac{ePr^2}{4b^2c_{\mathrm{Lip}}^2} \vee 2e),$$

*and*

$$J\left(\delta, \sigma, \mathcal{A}_{\mathfrak{h}}\right) \leq 5(M+1)c_{\text{Lip}}\sqrt{\log\left(eP\left(M+\frac{1}{\nu}\right)^2 \vee 2e\right)}$$
$$\times \log\frac{8\sigma R}{\delta},$$

where

$$R' := 2(a_{\text{Lip}})^L\sqrt{L}\|\boldsymbol{x}\|_n\left(\frac{2}{L}\right)^L\left(M+\frac{1}{\nu}\right)^{L+1},$$

and

$$R := \max\{R', 1\}.$$

## B    Appendix: Proofs

In this part, we provide proof for the main results of the paper.

**Proof of Proposition 1**

*Proof.* Because for $\Theta_1, \Theta_2 \in \mathcal{A}_{\mathfrak{h}}$ we get

$$\mathfrak{h}[\alpha\Theta_1 + (1-\alpha)\Theta_2] =$$
$$\|\boldsymbol{G}(\alpha\Theta_1 + (1-\alpha)\Theta_2)\|_1 + \nu\|\alpha\widetilde{\Theta}_1 + (1-\alpha)\widetilde{\Theta}_2\|_1,$$

where $\Theta_1 = \overline{\Theta}_1 + \widetilde{\Theta}_1$ and $\Theta_2 = \overline{\Theta}_2 + \widetilde{\Theta}_2$. Using triangular inequality, we obtain

$$\|\boldsymbol{G}(\alpha\Theta_1 + (1-\alpha)\Theta_2)\|_1 + \nu\|\alpha\widetilde{\Theta}_1 + (1-\alpha)\widetilde{\Theta}_2\|_1$$
$$\leq \alpha\|\boldsymbol{G}\Theta_1\|_1 + (1-\alpha)\|\boldsymbol{G}\Theta_2\|_1 + \alpha\nu\|\widetilde{\Theta}_1\|_1 + (1-\alpha)\nu\|\widetilde{\Theta}_2\|_1$$
$$= \alpha(\|\boldsymbol{G}\Theta_1\|_1 + \nu\|\widetilde{\Theta}_1\|_1) + (1-\alpha)(\|\boldsymbol{G}\Theta_2\|_1 + \nu\|\widetilde{\Theta}_2\|_1)$$
$$= \alpha\mathfrak{h}[\Theta_1] + (1-\alpha)\mathfrak{h}[\Theta_2].$$

For the second part of proposition, we have

$$\mathfrak{h}[\alpha\Theta] = \|\boldsymbol{G}\alpha\Theta\|_1 + \nu\|\alpha\widetilde{\Theta}\|_1 = |\alpha|\,\mathfrak{h}[\Theta].$$

Moreover,

$$\mathfrak{h}[\Theta] = 0 \Rightarrow \|\boldsymbol{G}\Theta\|_1 + \nu\|\widetilde{\Theta}\|_1 = 0$$
$$\Rightarrow \begin{cases} \text{I.} & \|\boldsymbol{G}\overline{\Theta}\|_1 = 0 \\ \text{II.} & \|\widetilde{\Theta}\|_1 = 0 \end{cases},$$

from (I) we obtain that $\boldsymbol{G}\overline{\Theta} = 0$. Since $\overline{\Theta} \in \ker(\boldsymbol{G})^\perp$ we obtain that $\overline{\Theta} = 0$. Therefore, (II) results that $\Theta = 0$. Furthermore, by convexity we obtain

$$\mathfrak{h}[\Theta_1 + \Theta_2] = 2\mathfrak{h}\left[\frac{\Theta_1 + \Theta_2}{2}\right]$$
$$\leq 2\mathfrak{h}\left[\frac{\Theta_1}{2}\right] + 2\mathfrak{h}\left[\frac{\Theta_2}{2}\right]$$
$$= \mathfrak{h}[\Theta_1] + \mathfrak{h}[\Theta_2].$$

$\square$

**Proof of Theorem 3**

*Proof.* Similar to (Taheri et al., 2021) we employ Proposition 6 of (Taheri et al., 2021) to prove Theorem 20. Compared with (Taheri et al., 2021) we can use the below equation.

$$\max_{l \in \{0,\dots,L\}} \prod_{\substack{j \in \{0,\dots,L\} \\ j \neq l}} \left( \|\boldsymbol{W}^j\|_2 \vee \|\boldsymbol{V}^j\|_2 \right) \leq \left( \frac{1}{L} \sum_{\substack{j=0 \\ j \neq l}}^{L} \left( \|\boldsymbol{W}^j\|_2 \vee \|\boldsymbol{V}^j\|_2 \right) \right)^L.$$

Our goal is to find an upper bound for $\sum_j \left( \|\boldsymbol{W}^j\|_1 \vee \|\boldsymbol{V}^j\|_1 \right)$ in terms of $\mathfrak{h}$. Considering Equation (17) we get

$$\boldsymbol{W}^j := \overline{\boldsymbol{W}}^j + \widetilde{\boldsymbol{W}}^j, \widetilde{\boldsymbol{W}}^j \in \ker(\boldsymbol{G}).$$

As a result, employing 1. triangle inequality, 2. invertible matrix $\bar{\boldsymbol{G}}^j$ induced from $\boldsymbol{G}^j$, 3. norm property in Definition (3), 4. factorizing the largest norm value and using Equation (18), 5. using $\mathfrak{h}[\Omega] \leq 1$ yield

$$
\begin{aligned}
\sum_j \|\boldsymbol{W}^j\|_1 &= \sum_j \|\overline{\boldsymbol{W}}^j + \widetilde{\boldsymbol{W}}^j\|_1 \\
&\leq \sum_j \|\overline{\boldsymbol{W}}^j\|_1 + \|\widetilde{\boldsymbol{W}}^j\|_1 \\
&\leq \sum_j \|\bar{\boldsymbol{G}}^{j^{-1}} \bar{\boldsymbol{G}}^j \overline{\boldsymbol{W}}^j\|_1 + \|\widetilde{\boldsymbol{W}}^j\|_1 \\
&\leq \sum_j \|\bar{\boldsymbol{G}}^{j^{-1}}\|_{1 \to 1} \|\bar{\boldsymbol{G}}^j \overline{\boldsymbol{W}}^j\|_1 + \|\widetilde{\boldsymbol{W}}^j\|_1 \\
&\leq \max_j \|\bar{\boldsymbol{G}}^{j^{-1}}\|_{1 \to 1} \mathfrak{h}(\bar{\Theta}, \widetilde{\Theta}) + \|\widetilde{\Theta}\|_1 \\
&\leq \max_j \|\bar{\boldsymbol{G}}^{j^{-1}}\|_{1 \to 1} + \|\widetilde{\Theta}\|_1,
\end{aligned}
\tag{21}
$$

where in the last inequality we restricted ourselves to the $\mathcal{A}_{\mathfrak{h}}$. Also since $\mathfrak{h}[\Omega] \leq 1$ we obtain

$$\nu \|\widetilde{\Theta}\|_1 \leq 1.$$

Then we can write inequality (21) as follows

$$\sum_j \|\boldsymbol{W}^j\|_1 \leq M + \frac{1}{\nu}, \tag{22}$$

where $M$ is defined in Equation (15). As a result equivalent to Lemma 7 of (Taheri et al., 2021) we get

$$
\begin{aligned}
\sum_{j=0}^{L} \left( \|\boldsymbol{W}^j\|_2 \vee \|\boldsymbol{V}^j\|_2 \right) &\leq \sum_{j=0}^{L} \left( \|\boldsymbol{W}^j\|_2 + \|\boldsymbol{V}^j\|_2 \right) \\
&\leq \sum_{j=0}^{L} \left( \|\boldsymbol{W}^j\|_1 + \|\boldsymbol{V}^j\|_1 \right) \\
&\leq 2M + \frac{2}{\nu},
\end{aligned}
$$

and therefore

$$\max_{\substack{l \in \{0,\ldots,L\}}} \prod_{\substack{j \in \{0,\ldots,L\} \\ j \neq l}} \left( \|\boldsymbol{W}^j\|_2 \vee \|\boldsymbol{V}^j\|_2 \right)$$

$$\leq \left( \frac{1}{L} \sum_{\substack{j=0 \\ j \neq l}}^{L} \left( \|\boldsymbol{W}^j\|_2 \vee \|\boldsymbol{V}^j\|_2 \right) \right)^L$$

$$\leq \left( \frac{2}{L} \left( M + \frac{1}{\nu} \right) \right)^L.$$

According to Lemma 6 of (Taheri et al., 2021) for $\Theta, \Gamma \in \mathcal{A}_\mathfrak{h}$ we get

$$\|\mathfrak{g}_\Theta - \mathfrak{g}_\Gamma\|_n \leq 2(a_{\mathrm{Lip}})^L \sqrt{L} \|\boldsymbol{x}\|_n \times$$
$$\max_{\substack{l \in \{0,\ldots,L\}}} \prod_{\substack{j \in \{0,\ldots,L\} \\ j \neq l}} \left( \|\boldsymbol{W}^j\|_2 \vee \|V^j\|_2 \right) \|\Theta - \Gamma\|_F.$$

Therefore, if $\Omega, \Gamma \in \mathcal{A}_\mathfrak{h}$ we obtain

$$\|\mathfrak{g}_\Omega - \mathfrak{g}_\Gamma\|_n \leq 2(a_{\mathrm{Lip}})^L \sqrt{L} \|\boldsymbol{x}\|_n \left( \frac{2}{L} \left( M + \frac{1}{\nu} \right) \right)^L \|\Omega - \Gamma\|_F,$$

where we define,

$$c_{\mathrm{Lip}} := 2(a_{\mathrm{Lip}})^L \sqrt{L} \|\boldsymbol{x}\|_n \left( \frac{2}{L} \left( M + \frac{1}{\nu} \right) \right)^L.$$

$\square$

**Proof of Theorem 4**

*Proof.* Because according to Theorem (3)

$$\|\mathfrak{g}_\Omega - \mathfrak{g}_0\|_n \leq 2(a_{\mathrm{Lip}})^L \sqrt{L} \|\boldsymbol{x}\|_n \left( \frac{2}{L} (M + \frac{1}{\nu}) \right)^L \|\Omega - 0\|_F, \tag{23}$$

and according to (22) we find that

$$\|\Omega\|_F \leq \sum_j \|\boldsymbol{W}^j\|_1 \leq M + \frac{1}{\nu}. \tag{24}$$

$\square$

**Proof of Lemma 2**

*Proof.* Again employing 1. triangle inequality, 2. invertible matrix $\bar{\boldsymbol{G}}^j$ induced from $\boldsymbol{G}^j$, 3. norm property in Definition (3) 4. factorizing the largest norm value, applying this definition that $b := \max\{M, 1\}$, and using Equation (18), yield

$$\boldsymbol{W} \in \mathcal{A}_{\mathfrak{h}} \Rightarrow \|\boldsymbol{W}\|_1 \leq \sum_{j=1}^{L} \|\overline{\boldsymbol{W}}^j\|_1 + \|\widetilde{\boldsymbol{W}}^j\|_1$$

$$= \sum_{j=1}^{L} \|\bar{\boldsymbol{G}}_j^{-1} \bar{\boldsymbol{G}}_j \overline{\boldsymbol{W}}^j\|_1 + \|\widetilde{\boldsymbol{W}}^j\|_1$$

$$\leq \sum_{j=1}^{L} \|\bar{\boldsymbol{G}}_j^{-1}\|_{1 \to 1} \|\bar{\boldsymbol{G}}_j \overline{\boldsymbol{W}}^j\|_1 + \|\widetilde{\boldsymbol{W}}^j\|_1$$

$$\leq b \left( \sum_{j=1}^{L} \|\bar{\boldsymbol{G}}_j \overline{\boldsymbol{W}}^j\|_1 + \|\widetilde{\boldsymbol{W}}^j\|_1 \right).$$

$\square$

**Proof of Lemma 3**

*Proof.* By employing Lemma 2 and using the fact that If $\mathscr{A} \subset \mathscr{B}$ then $\mathcal{N}(\varepsilon, \mathscr{A}, \|.\|) \leq \mathcal{N}(\varepsilon, \mathscr{B}, \|.\|)$, we get

$$\mathcal{N}(\epsilon, \mathcal{A}_{\mathfrak{h}}, \|.\|_2) \leq \mathcal{N}(\epsilon, (B_1(b) \subset \mathbb{R}^P), \|.\|_2)$$
$$= \mathcal{N}(\frac{\epsilon}{b}, (B_1(1) \subset \mathbb{R}^P), \|.\|_2). \tag{25}$$

Then using 1. the definition of the entropy 2. Equation (25) 3. definition of $c_{\text{Lip}}$ we obtain

$$H \left( \frac{r}{2(a_{\text{Lip}})^L \sqrt{L} \left( \frac{2}{L} \left( M + \frac{1}{\nu} \right) \right)^L \|\boldsymbol{x}\|_n}, \mathcal{A}_{\mathfrak{h}} \subset \mathbb{R}^P, \|\cdot\|_2 \right)$$

$$= \log \mathcal{N} \left( \frac{r}{2(a_{\text{Lip}})^L \sqrt{L} \left( \frac{2}{L} \left( M + \frac{1}{\nu} \right) \right)^L \|\boldsymbol{x}\|_n}, \mathcal{A}_{\mathfrak{h}} \subset \mathbb{R}^P, \|\cdot\|_2 \right)$$

$$\leq \log \mathcal{N} \left( \frac{r}{2b(a_{\text{Lip}})^L \sqrt{L} \left( \frac{2}{L} \left( M + \frac{1}{\nu} \right) \right)^L \|\boldsymbol{x}\|_n}, B_1(1) \subset \mathbb{R}^P, \|\cdot\|_2 \right)$$

$$= \log \mathcal{N} \left( \frac{r}{2bc_{\text{Lip}}}, B_1(1) \subset \mathbb{R}^P, \|\cdot\|_2 \right).$$

From (Taheri et al., 2021) we obtain

$$H \left( \sqrt{2}\mu, B_1(1) \subset \mathbb{R}^P, \|\cdot\|_2 \right)$$

$$= \log \mathcal{N} \left( \sqrt{2}\mu, B_1(1) \subset \mathbb{R}^P, \|\cdot\|_2 \right)$$

$$\leq \frac{3}{\mu^2} \log(2eP\mu^2 \vee 2e),$$

where $\mu := \frac{r}{2b\sqrt{2}c_{\text{Lip}}}$. This yields

$$H \left( \frac{r}{2(a_{\text{Lip}})^L \sqrt{L} \left( \frac{2}{L} \left( M + \frac{1}{\nu} \right) \right)^L \|\boldsymbol{x}\|_n}, \mathcal{A}_{\mathfrak{h}} \subset \mathbb{R}^P, \|\cdot\|_2 \right)$$

$$\leq H \left( \sqrt{2}\mu, B_1(1) \subset \mathbb{R}^P, \|\cdot\|_2 \right)$$

$$\leq \frac{24b^2 c_{\text{Lip}}^2}{r^2} \log(\frac{ePr^2}{4b^2 c_{\text{Lip}}^2} \vee 2e).$$

We need to find a constant $R \in [0, \infty)$ so that $\sup_{\Omega \in \mathcal{A}_{\mathfrak{h}}} \|\mathfrak{g}_\Omega\|_n \leq R$. employing Equations (23) and (24) we find that

$$R' := 2(a_{\text{Lip}})^L \sqrt{L} \|\boldsymbol{x}\|_n \left(\frac{2}{L}\right)^L (M + \frac{1}{\nu})^{L+1},$$

and

$$R := \max\{R', 1\}.$$

Therefore, for all $\Gamma \in \mathcal{A}_{\mathfrak{h}}$, it holds that $\mathcal{N}(r, \mathcal{G}_{\mathfrak{h}}, \|\cdot\|_n) = 1$ for all $r > R$ and, consequently, $H(r, \mathcal{G}_{\mathfrak{h}}, \|\cdot\|_n) = 0$ for all $r > R$. Thus we assume that $r \leq R$, as a result

$$J(\delta, \sigma, \mathcal{A}_{\mathfrak{h}}) = \int_{\frac{\delta}{8\sigma}}^{R} H^{\frac{1}{2}}(r, \mathcal{G}_{\mathfrak{h}}, \|\cdot\|_2) \, dr$$

$$\leq \int_{\frac{\delta}{8\sigma}}^{R} \left(\frac{24b^2 c_{\text{Lip}}^2}{r^2} \log\left(\frac{ePr^2}{4b^2 c_{\text{Lip}}^2} \vee 2e\right)\right)^{\frac{1}{2}} dr$$

$$\leq 5b c_{\text{Lip}} \sqrt{\log\left(\frac{ePR^2}{4b^2 c_{\text{Lip}}^2} \vee 2e\right)} \int_{\frac{\delta}{8\sigma}}^{R} \frac{1}{r} \, dr$$

$$= 5b c_{\text{Lip}} \sqrt{\log\left(\frac{ePR^2}{4b^2 c_{\text{Lip}}^2} \vee 2e\right)} \log \frac{8\sigma R}{\delta}$$

$$= \sqrt{\log\left(\frac{eP(2(a_{\text{Lip}})^L \sqrt{L}\|\boldsymbol{x}\|_n(\frac{1}{L})^L(M+\frac{1}{\nu})^{L+1})^2}{4b^2(2(a_{\text{Lip}})^L \sqrt{L}\|\boldsymbol{x}\|_n\left(\frac{2}{L}(M+\frac{1}{\nu})^L\right)^2} \vee 2e\right)}$$

$$\times 5b c_{\text{Lip}} \log \frac{8\sigma R}{\delta}$$

$$= 5b c_{\text{Lip}} \sqrt{\log\left(\frac{eP(M+\frac{1}{\nu})^2}{b^2} \vee 2e\right)} \log \frac{8\sigma R}{\delta}$$

$$\leq 5(M+1) c_{\text{Lip}} \sqrt{\log\left(eP(M+\frac{1}{\nu})^2 \vee 2e\right)} \log \frac{8\sigma R}{\delta}.$$

$\square$

### Proof of the main result of Section 4 (Theorem 1)

*Proof.* We recall that

$$\|\mathfrak{g}_\Omega - \mathfrak{g}_\Gamma\|_n \leq 2(a_{\text{Lip}})^L \sqrt{L} \|\boldsymbol{x}\|_n \left(\frac{2}{L}(M+\frac{1}{\nu})\right)^L \|\Omega - \Gamma\|_F$$

$$\|\mathfrak{g}_\Omega\|_n \leq 2(a_{\text{Lip}})^L \sqrt{L} \|\boldsymbol{x}\|_n \left(\frac{1}{L}\right)^L (M+\frac{1}{\nu})^{L+1}$$

$$J(\delta, \sigma, \mathcal{A}_{\mathfrak{h}}) \leq 5(M+1) c_{\text{Lip}} \sqrt{\log\left(eP\left(M+\frac{1}{\nu}\right)^2 \vee 2e\right)} \times$$

$$\log \frac{8\sigma R}{\delta},$$

As we see in the Section (4.2.1), in practice $M$ is a parameter whose value is less than one and close to zero. We set $\nu \geq \frac{1}{1-M}$ As a result we see that the above upper bound can be simplified as follows

$$J(\delta, \sigma, \mathcal{A}_{\mathfrak{h}}) \leq 5(M+1) c_{\text{Lip}} \sqrt{\log(2P)} \log \frac{8\sigma R}{\delta}$$

$$R' = 2(a_{\text{Lip}})^L \sqrt{L} \|x\|_n (\frac{2}{L})^L.$$

which is the first ingredient we need to employ corollary 8.3 of (van de Geer, 2000). In order to apply this corollary we also need to choose parameters $\delta, \sigma \in (0, \infty)$ so that

$$\delta < \sigma R,$$

and

$$\sqrt{n} \geq \frac{a_{\text{sub}}}{\delta} (J(\delta, \sigma, \mathcal{A}_{\mathfrak{h}}) \vee R).$$

Now we will try to find proper values for them. For simplicity we define:

$$\eta(M, L, P) := 5(M + 1)c_{\text{Lip}}\sqrt{\log(2P)}.$$

Also we set

$$\delta := 4a_{\text{sub}}R \times f \frac{\log(2n)}{\sqrt{n}},$$

where $f$ is a parameter related to $M, L, P$ that we will find later.

In this step we first find an upper bound for $\frac{8\sigma R}{\delta}$ that we need in our computations. We define $\sigma := 2\delta/R \vee \sqrt{\gamma}$. 1. by definition of $\sigma$ and $\delta$, 2. assuming $4\sqrt{2}a_{\text{sub}} \geq \gamma/f$ we get

$$
\begin{aligned}
\frac{8\sigma R}{\delta} &= \frac{8R\left(\frac{8a_{\text{sub}}R \times f \frac{\log(2n)}{\sqrt{n}}}{R} \vee \sqrt{2}\gamma\right)}{4a_{\text{sub}}R \times f \frac{\log(2n)}{\sqrt{n}}} \\
&= 16 \vee \left(\frac{2\sqrt{2}\gamma\sqrt{n}}{fa_{\text{sub}}\log(2n)}\right) \leq 16 \vee \frac{16\sqrt{n}}{\log(2n)} \leq 16\sqrt{n}.
\end{aligned}
\tag{26}
$$

As a result, using 1. Equation (26), 2. the fact that $\log$ is an increasing function and $R \geq 1$:

$$
\begin{aligned}
\frac{a_{\text{sub}}(R \vee J)}{\delta} &= \frac{\sqrt{n}a_{\text{sub}}\left[R \vee \eta(M, L, P)\log\left(\frac{8\sigma R}{\delta}\right)\right]}{4a_{\text{sub}}R \times f(M, L, P)\log(2n)} \\
&\leq \frac{\sqrt{n}\left[1 \vee \eta(M, L, P)\log(16\sqrt{n})\right]}{4f(M, L, P)\log(2n)} \\
&= \sqrt{n}\left(\frac{1}{4f(M, L, P)\log(2n)} \vee \frac{\eta(M, L, P)}{2f(M, L, P)}\right).
\end{aligned}
$$

If we assume that $n$ is enough large and by selecting constant $f = \frac{\eta}{2}$ we get

$$\frac{a_{\text{sub}}(J \vee R)}{\delta} \leq \sqrt{n},$$

which is satisfying the second condition as desired.

Therefore, using 1. Lemma 11 of (Taheri et al., 2021) (where $v := \sigma^2$), 2. the fact that $\mathbb{P}(\mathcal{C} \cap \mathcal{D}) \leq \alpha$ results $\mathbb{P}\left(\mathcal{C}^{\complement} \cup \mathcal{D}^{\complement}\right) \geq 1 - \alpha$ and $\mathbb{P}\left(\mathcal{C}^{\complement}\right) \geq \mathbb{P}\left(\mathcal{C}^{\complement} \cup \mathcal{D}^{\complement}\right) - \mathbb{P}\left(\mathcal{D}^{\complement}\right)$ we obtain

$$
\mathbb{P}\left(\left\{\sup_{\Omega \in \mathcal{A}_{\mathfrak{h}}} |\frac{1}{n}\sum_{i=1}^{n}\mathfrak{g}_{\Omega}(\boldsymbol{x}_i)u_i| \geq \delta\right\} \cap \left\{\frac{1}{n}\sum_{i=1}^{n}u_i^2 \leq \sigma^2\right\}\right)
$$

$$
\leq a_{\mathrm{sub}}e^{\frac{-n\delta^2}{(a_{\mathrm{sub}}R)^2}}
$$

$$
\Rightarrow \mathbb{P}\left(\left\{\sup_{\Omega \in \mathcal{A}_{\mathfrak{h}}} |\frac{1}{n}\sum_{i=1}^{n}\mathfrak{g}_{\Omega}(\boldsymbol{x}_i)u_i| \leq \delta\right\}\right) \tag{27}
$$

$$
\geq 1 - a_{\mathrm{sub}}e^{\frac{-n\delta^2}{(a_{\mathrm{sub}}R)^2}} - e^{\frac{n\sigma^2}{12K^2}}
$$

$$
\Rightarrow \mathbb{P}\left(\left\{\sup_{\Omega \in \mathcal{A}_{\mathfrak{h}}} |\frac{1}{n}\sum_{i=1}^{n}\mathfrak{g}_{\Omega}(\boldsymbol{x}_i)u_i| \leq \delta\right\}\right) \geq 1 - \frac{1}{n},
$$

where in the last line we used the fact that

$$
\delta = a_{\mathrm{sub}}R \times \frac{\eta}{2}\frac{\log(2n)}{\sqrt{n}}
$$

$$
= a_{\mathrm{sub}}R \times \frac{5(M+1)c_{\mathrm{Lip}}\sqrt{\log(2P)}}{2}\frac{\log(2n)}{\sqrt{n}},
$$

and

$$
\frac{-n\delta^2}{(a_{\mathrm{sub}}R)^2} = \frac{-na_{\mathrm{sub}}^2 R^2 \times \frac{25(M+1)^2 c_{\mathrm{Lip}}^2 \log(2P)}{4}\left(\frac{\log(2n)}{\sqrt{n}}\right)^2}{(a_{\mathrm{sub}}R)^2}
$$

$$
= -\frac{25}{4}(M+1)^2 c_{\mathrm{Lip}}^2 \log(2P)\left(\log(2n)\right)^2.
$$

Equation (27) states that $\lambda_{\mathfrak{h},t} \leq 2\delta$ for $t = 1/n$. Then the main result can be implied from Theorem 2 of (Taheri et al., 2021) by setting $\lambda \geq 2\delta = 2a_{\mathrm{sub}}R \times f\frac{\log(2n)}{\sqrt{n}}$. $\qquad\square$

**Proof of Corollary 1**

*Proof.* We know that

$$
\bar{\boldsymbol{G}}^{-1} \ : \ \mathrm{Im}(\boldsymbol{G}^j) \to \ker(\boldsymbol{G})^{\perp}.
$$

for $\boldsymbol{G} = 0$ we have $\mathrm{Im}(\boldsymbol{G}^j) = \boldsymbol{0}$ and $\ker(\boldsymbol{G})^{\perp} = \boldsymbol{0}$ (as the $\ker(\boldsymbol{0})$ contains all the space). Therefore, $\bar{\boldsymbol{G}}^{-1}$ is also equal to zero. As a result $M_{\boldsymbol{0}} = 0$ and Theorem 1 holds for every $\nu \geq 1$. $\qquad\square$

**Proof of Upper Bound for Parameter $M$ (Lemma 1)**

To prove this lemma we first prove Corollary 6. This lemma is a direct result when we have just one partition.

*Proof.* We know that $\boldsymbol{A}^j : \mathcal{M}_{p_{j+1} \times p_j} \to \mathcal{M}_{l_j \times p_j}$ where $\mathcal{M}_{p_{j+1} \times p_j} = \ker(\boldsymbol{A}^j) \oplus \ker(\boldsymbol{A}^j)^{\perp}$ ($\oplus$ denotes the direct sum) and $l_j = \sum_i \binom{|\mathcal{B}_i^j|}{2}$. We also have

$$
\ker \boldsymbol{A}^j = \left\{\boldsymbol{W}^j : \text{consist of same rows in each partition}\right\}.
$$

In fact

$$\boldsymbol{A}^j \ : \ \ker(\boldsymbol{A}^j) \oplus \ker(\boldsymbol{A}^j)^\perp \to \mathrm{Im}(\boldsymbol{A}^j) \subset \mathcal{M}_{l_j \times p_j},$$

and by definition $\bar{\boldsymbol{A}}^j : \ker(\boldsymbol{A}^j)^\perp \to \mathrm{Im}(\boldsymbol{A}^j)$ is the restriction of $\boldsymbol{A}^j$ to the subspace $\ker(\boldsymbol{A}^j)^\perp$. Therefore, it is invertible and we obtain

$$\bar{\boldsymbol{A}}^{j^{-1}} \ : \ \mathrm{Im}(\boldsymbol{A}^j) \to \ker(\boldsymbol{A}^j)^\perp.$$

We know that groups provide a partition for rows therefore $\boldsymbol{A}^j = \bigoplus_i \sqrt{P_{i,j}} \boldsymbol{A}_i^j$ where $\boldsymbol{A}_i^j$ is the matrix associated to each group. We also know

$$\boldsymbol{A}^{j^\dagger} \ : \ \mathcal{M}_{l_j \times p_j} \to \ker(\boldsymbol{A}^j)^\perp,$$

by the geometric interpretation of $\boldsymbol{A}^{j^\dagger}$ (pseudo-inverse of $\boldsymbol{A}$) we know that it is an extension of $\bar{\boldsymbol{A}}^{j^{-1}}$ : $\mathrm{Im}(\boldsymbol{A}^j) \to ker(\boldsymbol{A}^j)^\perp$. We need below auxiliary Lemma for the rest of computations.

**Lemma 4** (Equivalence of norm on two spaces). *$\| \cdot \|_{1 \to 1}$ of $\boldsymbol{A}^j$ as a linear map from $\mathbb{R}^{p_{j+1}} \to \mathbb{R}^{l_j}$ is the same as the $\| \cdot \|_{1 \to 1}$ of $\boldsymbol{A}^j$ as a linear map from $\mathcal{M}_{p_{j+1} \times p_j}$ to $\mathcal{M}_{l_j \times p_j}$.*

*Proof.* Let $\boldsymbol{D} \in \mathcal{M}_{p_{j+1} \times p_j}$ and let denote its columns by $\boldsymbol{d}_1, ..., \boldsymbol{d}_{p_j}$. Therefore, $\boldsymbol{A}^j(\boldsymbol{D}) = [\boldsymbol{A}^j \boldsymbol{d}_1, ..., \boldsymbol{A}^j \boldsymbol{d}_{p_j}]$. By (28, 29), $\| \cdot \|_{1 \to 1}$ of $\boldsymbol{A}^j$ as a linear map $\mathcal{M}_{p_{j+1} \times p_j}$ to $\mathcal{M}_{l_j \times p_j}$ is equal to $\| \cdot \|_{1 \to 1}$ of $\boldsymbol{A}^j$ as a linear map from $\mathbb{R}^{p_{j+1}} \to \mathbb{R}^{l_j}$. $\square$

Also by definition

$$\|\bar{\boldsymbol{A}}^{j-1}\|_{1 \to 1} \ = \ \sup_{\boldsymbol{x} \in \mathrm{Im}(\boldsymbol{A}^j)} \frac{\|\bar{\boldsymbol{A}}^{j^{-1}} \boldsymbol{x}\|_1}{\|\boldsymbol{x}\|_1},$$

and

$$\|\boldsymbol{A}^{j^\dagger}\|_{1 \to 1} \ = \ \sup_{\boldsymbol{x} \in \mathcal{M}_{l_j \times p_j}} \frac{\|\boldsymbol{A}^{j^\dagger} \boldsymbol{x}\|_1}{\|\boldsymbol{x}\|_1}.$$

Since the sup is taking over a larger set we get

$$\|\bar{\boldsymbol{A}}^{j^{-1}}\|_{1 \to 1} \ \le \ \|\boldsymbol{A}^{j^\dagger}\|_{1 \to 1}.$$

Note that $\boldsymbol{A}^j$ is a linear map from the linear space $\mathcal{M}_{p_{j+1} \times p_j}$ to the linear space $\mathcal{M}_{l_j \times p_j}$. ( If $\boldsymbol{D} \in \mathcal{M}_{l_j \times p_j}$ then:$\boldsymbol{A}^{j^\dagger}(D) = (\bar{\boldsymbol{A}}^j)^{-1}$(orthogonal projection of $D$ on $\mathrm{I}m(\boldsymbol{A}^j)$) which shows $\ker \boldsymbol{A}^{j^\dagger}$ is an extension of $\bar{\boldsymbol{A}}^{j^{-1}}$). Because of special structure of matrix $\boldsymbol{A}^j$ we can see that

$$\boldsymbol{A}_i^{j^\dagger} \ = \ \frac{1}{\sqrt{P_{i,j}}|\mathcal{B}_i^j|} \times \boldsymbol{A}_i^{j^\top},$$

therefore:

$$\|\boldsymbol{A}_i^{j^\dagger}\|_{1 \to 1} \ = \ \frac{2}{\sqrt{P_{i,j}}|\mathcal{B}_i^j|}.$$

On the other hand, if

$$\boldsymbol{B} \ = \ \boldsymbol{B}_1 \oplus \boldsymbol{B}_2 \oplus \ldots, \tag{28}$$

then

$$\|\boldsymbol{B}\|_{1 \to 1} \ = \ \sup_i \|\boldsymbol{B}_i\|_{1 \to 1}. \tag{29}$$

As a result

$$
\begin{aligned}
M \;=\; \sup_j \|\boldsymbol{A}^{j^{-1}}\|_{1\to 1} &\leq \sup_j \sup_i \|\boldsymbol{A}_i^{j^{\dagger}}\|_{1\to 1} \\
&= \sup_j \sup_i \frac{2}{\sqrt{P_{i,j}}|\mathcal{B}_i^j|}.
\end{aligned}
$$

$\square$

**Proof of Prediction Bound for Tensors and Boundedness of Effective Noise (Theorem 4)**

*Proof.* We have

$$
\|\mathbb{Y} - \langle \mathbb{X}, \widehat{\mathbb{B}} \rangle_L\|_F^2 + \lambda\mathfrak{h}[\widehat{\mathbb{B}}] \leq \|\mathbb{Y} - \langle \mathbb{X}, \mathbb{A} \rangle_L\|_F^2 + \lambda\mathfrak{h}[\mathbb{A}].
$$

Adding a zero-valued term in the $\ell_2$-norms on both sides then gives us

$$
\begin{aligned}
&\|\mathbb{Y} - \langle \mathbb{X}, \mathbb{B} \rangle_L + \langle \mathbb{X}, \mathbb{B} \rangle_L - \langle \mathbb{X}, \widehat{\mathbb{B}} \rangle_L\|_F^2 + \lambda\mathfrak{h}[\widehat{\mathbb{B}}] \\
&\leq \|\mathbb{Y} - \langle \mathbb{X}, \mathbb{B} \rangle_L + \langle \mathbb{X}, \mathbb{B} \rangle_L - \langle \mathbb{X}, \mathbb{A} \rangle_L\|_F^2 + \lambda\mathfrak{h}[\mathbb{A}].
\end{aligned}
$$

The fact that $\|\mathbb{C}\|_F^2 = \langle \mathbb{C}, \mathbb{C} \rangle_D$ and expanding $\|\mathbb{C} + \mathbb{D}\|_F^2 = \langle \mathbb{C}, \mathbb{C} \rangle_D + \langle \mathbb{D}, \mathbb{D} \rangle_D + 2\langle \mathbb{C}, \mathbb{D} \rangle_D$, yields

$$
\begin{aligned}
&\|\mathbb{Y} - \langle \mathbb{X}, \mathbb{B} \rangle_L\|_F^2 + \|\langle \mathbb{X}, \mathbb{B} \rangle_L - \langle \mathbb{X}, \widehat{\mathbb{B}} \rangle_L\|_F^2 + \\
&2\left\langle \mathbb{Y} - \langle \mathbb{X}, \mathbb{B} \rangle_L, \langle \mathbb{X}, \mathbb{B} \rangle_L - \langle \mathbb{X}, \widehat{\mathbb{B}} \rangle_L \right\rangle_{M+1} + \lambda\mathfrak{h}[\widehat{\mathbb{B}}] \\
&\leq \|\mathbb{Y} - \langle \mathbb{X}, \mathbb{B} \rangle_L\|_F^2 + \|\langle \mathbb{X}, \mathbb{B} \rangle_L - \langle \mathbb{X}, \mathbb{A} \rangle_L\|_F^2 + \\
&2\left\langle \mathbb{Y} - \langle \mathbb{X}, \mathbb{B} \rangle_L, \langle \mathbb{X}, \mathbb{B} \rangle_L - \langle \mathbb{X}, \mathbb{A} \rangle_L \right\rangle_{M+1} + \lambda\mathfrak{h}[\mathbb{A}].
\end{aligned}
$$

Then we can then derive the below inequality.

$$
\begin{aligned}
\|\langle \mathbb{X}, \mathbb{B} \rangle_L - \langle \mathbb{X}, \widehat{\mathbb{B}} \rangle_L\|_F^2 &\leq \|\langle \mathbb{X}, \mathbb{B} \rangle_L - \langle \mathbb{X}, \mathbb{A} \rangle_L\|_F^2 \\
&+ 2\langle \mathbb{U}, \langle \mathbb{X}, \widehat{\mathbb{B}} \rangle_L - \langle \mathbb{X}, \mathbb{A} \rangle_L \rangle_{M+1} \\
&- \lambda\mathfrak{h}[\widehat{\mathbb{B}}] + \lambda\mathfrak{h}[\mathbb{A}].
\end{aligned} \tag{30}
$$

This inequality separates the prediction error of the estimator from other parts of the problem. We now try to control the inner product term on the right-hand side. Employing the properties of the inner product we can write this inner product as follows.

$$
\begin{aligned}
\langle \mathbb{U}, \langle \mathbb{X}, \widehat{\mathbb{B}} \rangle_L - \langle \mathbb{X}, \mathbb{A} \rangle_L \rangle_{M+1} &= \langle \mathbb{U}, \langle \mathbb{X}, \widehat{\mathbb{B}} - \mathbb{A} \rangle_L \rangle_{M+1} \\
&= \langle \langle \mathbb{X}, \mathbb{U} \rangle_1, \widehat{\mathbb{B}} - \mathbb{A} \rangle_{M+L} = \langle \langle \mathbb{X}, \mathbb{U} \rangle_1, \widehat{\mathbb{B}} \rangle_{M+L} + \langle \langle \mathbb{X}, \mathbb{U} \rangle_1, -\mathbb{A} \rangle_{M+L}.
\end{aligned}
$$

Then Holder's inequality gives

$$
\langle \langle \mathbb{X}, \mathbb{U} \rangle_1, \widehat{\mathbb{B}} \rangle_{M+L} \leq \|\langle \mathbb{X}, \mathbb{U} \rangle_1\|_\infty \|\widehat{\mathbb{B}}\|_1,
$$

and

$$
\langle \langle \mathbb{X}, \mathbb{U} \rangle_1, \mathbb{A} \rangle_{M+L} \leq \|\langle \mathbb{X}, \mathbb{U} \rangle_1\|_\infty \|\mathbb{A}\|_1.
$$

By replacing them in Equation (30) we obtain

$$
\begin{aligned}
\|\langle \mathbb{X}, \mathbb{B} \rangle_L - \langle \mathbb{X}, \widehat{\mathbb{B}} \rangle_L\|_F^2 &\leq \|\langle \mathbb{X}, \mathbb{B} \rangle_L - \langle \mathbb{X}, \mathbb{A} \rangle_L\|_F^2 \\
&+ 2\|\langle \mathbb{X}, \mathbb{U} \rangle_1\|_\infty \|\mathbb{A}\|_1 + 2\|\langle \mathbb{X}, \mathbb{U} \rangle_1\|_\infty \|\widehat{\mathbb{B}}\|_1 - \lambda\mathfrak{h}[\widehat{\mathbb{B}}] + \lambda\mathfrak{h}[\mathbb{A}].
\end{aligned} \tag{31}
$$

Similar to the previous case we know that the regularizer is a norm. Employing the inequality between norms we have $\|\mathbb{A}\|_1 \leq \mathcal{C}\mathfrak{h}[\mathbb{A}]$ where $\mathcal{C}$ is a constant. By replacing in (31) we get

$$
\|\langle \mathbb{X}, \mathbb{B} \rangle_L - \langle \mathbb{X}, \widehat{\mathbb{B}} \rangle_L\|_F^2 \leq \|\langle \mathbb{X}, \mathbb{B} \rangle_L - \langle \mathbb{X}, \mathbb{A} \rangle_L\|_F^2
$$
$$
+ 2\|\langle \mathbb{X}, \mathbb{U} \rangle_1\|_\infty \mathcal{C}\mathfrak{h}[\mathbb{A}] + 2\|\langle \mathbb{X}, \mathbb{U} \rangle_1\|_\infty \mathcal{C}\mathfrak{h}[\widehat{\mathbb{B}}] - \lambda\mathfrak{h}[\widehat{\mathbb{B}}] + \lambda\mathfrak{h}[\mathbb{A}].
$$

Assuming $2\|\langle \mathbb{U}, \mathbb{X} \rangle_1\|_\infty \mathcal{C} \leq \lambda$ we obtain

$$
\|\langle \mathbb{X}, \mathbb{B} \rangle_L - \langle \mathbb{X}, \widehat{\mathbb{B}} \rangle_L\|_F^2 \leq \|\langle \mathbb{X}, \mathbb{B} \rangle_L - \langle \mathbb{X}, \mathbb{A} \rangle_L\|_F^2 + 2\lambda\mathfrak{h}[\mathbb{A}].
$$

Since $\mathbb{A}$ was selected arbitrarily, we get

$$
\|\langle \mathbb{X}, \mathbb{B} \rangle_L - \langle \mathbb{X}, \widehat{\mathbb{B}} \rangle_L\|_F^2 \leq
$$
$$
s \inf_{\mathbb{A}} \left\{ \|\langle \mathbb{X}, \mathbb{B} \rangle_L - \langle \mathbb{X}, \mathbb{A} \rangle_L\|_F^2 + 2\lambda\mathfrak{h}[\mathbb{A}] \right\}.
$$

We now show boundedness of effective noise in the tensors. By definition of the sup-norm, it holds that

$$
2\|\langle \mathbb{U}, \mathbb{X} \rangle_1\|_\infty
$$
$$
= \sup_{\substack{p_1 \in \{1,...,P_1\} \cdots p_L \in \{1,...,P_L\} \\ q_1 \in \{1,...,Q_1\} \cdots q_M \in \{1,...,Q_M\}}} 2 \left| (\langle \mathbb{X}, \mathbb{U} \rangle_1)_{p_1,\cdots,p_L,q_1,\cdots,q_M} \right|.
$$

Therefore, the complement of the event is as follows

$$
\{\lambda_t \geq 2\|\langle \mathbb{X}, \mathbb{U} \rangle_1\|_\infty\}^{\complement} = \{2\|\langle \mathbb{X}, \mathbb{U} \rangle_1\|_\infty > \lambda_t\}
$$
$$
= \bigcup_{p_1=1}^{P_1} \cdots \bigcup_{q_M=1}^{Q_M} \left\{ 2 \left| (\langle \mathbb{X}, \mathbb{U} \rangle_1)_{p_1,\cdots,p_l,q_1,\cdots,q_M} \right| > \lambda_t \right\}.
$$

Applying the union bound, yields

$$
\mathbb{P}\left\{ 2\|\langle \mathbb{X}, \mathbb{U} \rangle_1\|_\infty > \lambda_t \right\} \leq
$$
$$
\sum_{p_1=1}^{P_1} \cdots \sum_{q_M=1}^{Q_M} \mathbb{P}\left\{ 2 \left| (\langle \mathbb{X}, \mathbb{U} \rangle_1)_{p_1,\cdots p_L,q_1,\cdots,q_M} \right| > \lambda_t \right\},
$$

and we obtain

$$
\mathbb{P}\left\{ 2\|\langle \mathbb{X}, \mathbb{U} \rangle_1\|_\infty > \lambda_t \right\} \leq P_1 \cdots P_L Q_1 \cdots Q_M \times
$$
$$
\sup_{\substack{p_1 \in \{1,...,P_1\} \cdots p_L \in \{1,...,P_L\} \\ q_1 \in \{1,...,Q_1\} \cdots q_M \in \{1,...,Q_M\}}} \mathbb{P}\left\{ 2 \left| (\langle \mathbb{X}, \mathbb{U} \rangle_1)_{p_1,\cdots,p_L,q_1,\cdots,q_M} \right| > \lambda_t \right\}.
$$

We define

$$
s := \sup_{q_1 \in \{1,...,Q_1\} \cdots q_M \in \{1,...,Q_M\}} 2 \left| (\langle \mathbb{X}, \mathbb{X} \rangle_1)_{q_1,\cdots,q_M,q_1,\cdots,q_M} \right| / N,
$$

then we get

$$
\mathbb{P}\left\{2\|\langle \mathbb{X},\mathbb{U}\rangle_1\|_\infty\right\}
$$

$$
\leq P_1\cdots P_L Q_1\cdots Q_M \sup_{\substack{p_1\in\{1,\dots,P_1\}\cdots p_L\in\{1,\dots,P_L\}\\ q_1\in\{1,\dots,Q_1\}\cdots q_M\in\{1,\dots,Q_M\}}}
$$

$$
\mathbb{P}\left\{2\left|(\langle\mathbb{X},\mathbb{U}\rangle_1)_{p_1,\cdots,p_L,q_1,\cdots,q_M}\right|>\lambda_t\right\}
$$

$$
= P_1\cdots P_L Q_1\cdots Q_M \sup_{\substack{p_1\in\{1,\dots,P_1\}\cdots p_L\in\{1,\dots,P_L\}\\ q_1\in\{1,\dots,Q_1\}\cdots q_M\in\{1,\dots,Q_M\}}}
$$

$$
\mathbb{P}\left\{\frac{\left|(\langle\mathbb{X},\mathbb{U}\rangle_1)_{p_1,\cdots,p_L,q_1,\cdots,q_M}\right|}{2\sigma\sqrt{Ns}}>\frac{\lambda_t}{2\sigma\sqrt{Ns}}\right\}
$$

$$
= P_1\cdots P_L Q_1\cdots Q_M \sup_{\substack{p_1\in\{1,\dots,P_1\}\cdots p_L\in\{1,\dots,P_L\}\\ q_1\in\{1,\dots,Q_1\}\cdots q_M\in\{1,\dots,Q_M\}}}
$$

$$
\mathbb{P}\left\{\frac{\left|(\langle\mathbb{X},\mathbb{U}\rangle_1)_{p_1,\cdots,p_L,q_1,\cdots,q_M}\right|}{\sigma\sqrt{\left|(\langle\mathbb{X},\mathbb{X}\rangle_1)_{q_1,\cdots,q_M,q_1,\cdots,q_M}\right|}}>\frac{\lambda_t}{2\sigma\sqrt{Ns}}\right\},
$$

where $\dfrac{\left|(\langle\mathbb{U},\mathbb{X}\rangle_1)_{p_1,\cdots,p_L,q_1,\cdots,q_M}\right|}{\sigma\sqrt{\left|(\langle\mathbb{X},\mathbb{X}\rangle_1)_{q_1,\cdots,q_M,q_1,\cdots,q_M}\right|}}\sim\mathcal{N}(0,1)$.

Because, $\mathbb{U}_{\bullet,p_1,\cdots,p_L}\sim\mathcal{N}(0,\sigma^2)$ and $\mathbb{X}_{\bullet,q_1,\cdots,q_M}/\sqrt{\left|(\langle\mathbb{X},\mathbb{X}\rangle_1)_{q_1,\cdots,q_M,q_1,\cdots,q_M}\right|}$ belongs to the unit sphere (Lemma 3.1 of (Xu & Zhang, 2019)). Therefore, we can employ the tail bound

$$
\mathbb{P}\{|z|\geq a\}\leq e^{-\frac{a^2}{2}}\quad\text{for all }a\geq 0.
$$

Combining this tail bound evaluated at $a=\lambda_t/(2\sigma\sqrt{Ns})$ we get

$$
\mathbb{P}\left\{2\|\langle\mathbb{X},\mathbb{U}\rangle_1\|_\infty>\lambda_t\right\}\leq P_1\cdots P_L Q_1\cdots Q_M e^{-\left(\frac{\lambda_t}{2\sigma\sqrt{Ns}}\right)^2/2}.
$$

As a result

$$
\mathbb{P}\left\{\lambda_t\geq 2\|\langle\mathbb{X},\mathbb{U}\rangle_1\|_\infty\right\}
$$
$$
= 1-\mathbb{P}\left\{2\|\langle\mathbb{X},\mathbb{U}\rangle_1\|_\infty>\lambda_t\right\}
$$
$$
\geq 1-P_1\cdots P_L Q_1\cdots Q_M e^{-\left(\frac{\lambda_t}{2\sigma\sqrt{Ns}}\right)^2/2}.
$$

$\square$

## C  Implementation

This section presents the implementation of our main algorithm (Algorithm 3).

```
1   import numpy as np
2   import matplotlib.pyplot as plt
3   import time
4   import math
5
6   def encode_columns_by_first_element(matrix):
7       num_rows = len(matrix)
8       num_cols = len(matrix[0])
9       encoded_matrix = [[0 for _ in range(num_cols)] for _ in range(num_rows)]
10      for col in range(num_cols):
11          is_complement = matrix[0][col] == 1
```

```
12          for row in range(num_rows):
13          value = matrix[row][col]
14          encoded_matrix[row][col] = (value + 1) % 2 if is_complement else value
15          return np.array(encoded_matrix)
16
17          def compress_matrix_by_first_row(matrix):
18          first_row = matrix[0]
19          return [[val, (val + 1) % 2] for val in first_row]
20
21          def encode_rows_by_first_element(matrix):
22          for row in matrix:
23          if row[0] == 1:
24          for i in range(len(row)):
25          row[i] = (row[i] + 1) % 2
26          return matrix
27
28          def compress_matrix_by_first_column(matrix):
29          first_col = [row[0] for row in matrix]
30          return [[val, (val + 1) % 2] for val in first_col]
31
32          def compressed_matrix_multiply(U_list_A, index_matrix_A, U_list_B, index_matrix_B):
33          num_products = len(index_matrix_A)
34          num_rows = len(index_matrix_A[0])
35          num_cols = len(index_matrix_B[0])
36          compressed_products = []
37          for i in range(num_products):
38          product = compute_outer_product(U_list_A[i], U_list_B[i])
39          compressed_products.append(product)
40          result_matrix = np.zeros((num_rows, num_cols))
41          for k in range(num_products):
42          num_rows_sub = len(compressed_products[k])
43          num_cols_sub = len(compressed_products[k][0])
44          for i in range(num_rows_sub):
45          row_indices = [idx for idx, val in enumerate(index_matrix_A[:, k]) if val == i]
46          for j in range(num_cols_sub):
47          col_indices = [idx for idx, val in enumerate(index_matrix_B[k, :]) if val == j]
48          result_matrix[np.ix_(row_indices, col_indices)] += compressed_products[k][i][j]
49          return result_matrix % 2
50
51          def compute_outer_product(vector_a, vector_b):
52          return [[vector_a[i] * vector_b[j] for j in range(len(vector_b))] for i in range(len(
                   vector_a))]
53
54          def standard_binary_matrix_multiply(matrix_a, matrix_b):
55          result = np.zeros((len(matrix_a), len(matrix_b[0])))
56          for i in range(len(matrix_a)):
57          for j in range(len(matrix_b[0])):
58          for k in range(len(matrix_b)):
59          result[i][j] += matrix_a[i][k] * matrix_b[k][j]
60          return result % 2
61
62          def compute_frobenius_norm(matrix):
63          return math.sqrt(sum(element ** 2 for row in matrix for element in row))
64
65          fast_times = []
66          standard_times = []
67          matrix_sizes = [30, 100, 200, 300]
68
69          print("Comparing fast (compressed) vs standard binary matrix multiplication:")
70          print("=" * 60)
71
72          for size in matrix_sizes:
73          matrix_A = np.random.randint(2, size=(size, size))
74          matrix_B = np.random.randint(2, size=(size, size))
75          start_time = time.time()
76          encoded_A = encode_columns_by_first_element(matrix_A)
77          encoded_B = encode_rows_by_first_element(matrix_B)
78          compressed_A = compress_matrix_by_first_row(matrix_A)
```

```
79          compressed_B = compress_matrix_by_first_column(matrix_B)
80          fast_result = compressed_matrix_multiply(compressed_A, encoded_A, compressed_B, encoded_B
                )
81          fast_time = time.time() - start_time
82          fast_times.append(fast_time)
83          start_time = time.time()
84          standard_result = standard_binary_matrix_multiply(matrix_A, matrix_B)
85          standard_time = time.time() - start_time
86          standard_times.append(standard_time)
87          error_matrix = fast_result - standard_result
88          error_norm = compute_frobenius_norm(error_matrix)
89          print(f"Matrix size {size}x{size} -> Frobenius norm error: {error_norm:.2f}")
90
91          plt.figure(figsize=(10, 6))
92          plt.plot(matrix_sizes, fast_times, marker='o', label='Fast (Compressed)', color='blue')
93          plt.plot(matrix_sizes, standard_times, marker='s', label='Standard', color='red')
94          plt.xlabel('Matrix size (n x n)')
95          plt.ylabel('Time (seconds)')
96          plt.title('Binary Matrix Multiplication Performance')
97          plt.legend()
98          plt.grid(True)
99          plt.tight_layout()
100         plt.savefig("matrix_times.png")
```

Listing 1: Cardinality sparsity vs Standard Binary Matrix Multiplication

```
1           import numpy as np
2           import matplotlib.pyplot as plt
3           import time
4           import math
5
6           def custom_unique(row):
7           unique_elements = []
8           first_indices = []
9           counts = []
10          for idx, value in enumerate(row):
11          if value not in unique_elements:
12          unique_elements.append(value)
13          first_indices.append(idx)
14          counts.append(1)
15          else:
16          element_index = unique_elements.index(value)
17          counts[element_index] += 1
18          return np.array(unique_elements), np.array(first_indices), np.array(counts)
19
20          def encode_first_matrix(matrix_A):
21          num_rows = len(matrix_A)
22          num_cols = len(matrix_A[0])
23          encoding_matrix = np.zeros((num_rows, 1))
24          unique_blocks = []
25          flat_indices = []
26          for col in range(num_cols):
27          unique_vals, first_indices, _ = custom_unique(matrix_A[:, col])
28          block_values = [matrix_A[index, col] for index in sorted(first_indices)]
29          unique_blocks.append(block_values)
30          for block_idx, value in enumerate(block_values):
31          matching_indices = [idx for idx, val in enumerate(matrix_A[:, col]) if val == value]
32          encoding_matrix[matching_indices, :] = block_idx
33          flat_indices = np.append(flat_indices, encoding_matrix)
34          index_matrix = np.reshape(flat_indices, (num_cols, num_rows))
35          return unique_blocks, index_matrix
36
37          def encode_second_matrix(matrix_B):
38          num_rows = len(matrix_B)
39          num_cols = len(matrix_B[0])
40          encoding_matrix = np.zeros((1, num_cols))
41          unique_blocks = []
42          flat_indices = []
```

```
43          for row in range(num_rows):
44          unique_vals, first_indices, _ = custom_unique(matrix_B[row, :])
45          block_values = [matrix_B[row, idx] for idx in sorted(first_indices)]
46          unique_blocks.append(block_values)
47          for block_idx, value in enumerate(block_values):
48          matching_indices = [idx for idx, val in enumerate(matrix_B[row, :]) if val == value]
49          encoding_matrix[:, matching_indices] = block_idx
50          flat_indices = np.append(flat_indices, encoding_matrix)
51          index_matrix = np.reshape(flat_indices, (num_rows, num_cols))
52          return unique_blocks, index_matrix
53
54          def compute_outer_product(vector_a, vector_b):
55          rows = len(vector_a)
56          cols = len(vector_b)
57          result = [[0] * cols for _ in range(rows)]
58          for i in range(rows):
59          for j in range(cols):
60          result[i][j] = vector_a[i] * vector_b[j]
61          return result
62
63          def compressed_matrix_multiply(blocks_A, index_A, blocks_B, index_B):
64          num_blocks = len(index_A)
65          num_rows = len(index_A[0])
66          num_cols = len(index_B[0])
67          compressed_products = []
68          for k in range(num_blocks):
69          outer = compute_outer_product(blocks_A[k], blocks_B[k])
70          compressed_products.append(outer)
71          result_matrix = np.zeros((num_rows, num_cols))
72          for block in range(num_blocks):
73          block_rows = len(compressed_products[block])
74          block_cols = len(compressed_products[block][0])
75          for i in range(block_rows):
76          row_indices = [idx for idx, val in enumerate(index_A[block, :]) if val == i]
77          for j in range(block_cols):
78          col_indices = [idx for idx, val in enumerate(index_B[block, :]) if val == j]
79          result_matrix[np.ix_(row_indices, col_indices)] += compressed_products[block][i][j]
80          return result_matrix
81
82          def standard_matrix_multiply(matrix_A, matrix_B):
83          result = np.zeros((len(matrix_A), len(matrix_B[0])))
84          for i in range(len(matrix_A)):
85          for j in range(len(matrix_B[0])):
86          for k in range(len(matrix_B)):
87          result[i][j] += matrix_A[i][k] * matrix_B[k][j]
88          return result
89
90          def frobenius_norm(matrix):
91          return math.sqrt(sum(element ** 2 for row in matrix for element in row))
92
93          fast_times = []
94          standard_times = []
95          matrix_sizes = [200, 250]
96          value_bias = np.random.normal(loc=3, scale=2)
97
98          for size in matrix_sizes:
99          matrix_A = np.random.randint(10, size=(size, size)) - value_bias
100         matrix_B = np.random.randint(10, size=(size, size)) - value_bias
101         start_time = time.time()
102         compressed_A, index_A = encode_first_matrix(matrix_A)
103         compressed_B, index_B = encode_second_matrix(matrix_B)
104         result_compressed = compressed_matrix_multiply(compressed_A, index_A, compressed_B,
                  index_B)
105         elapsed_fast = time.time() - start_time
106         fast_times.append(elapsed_fast)
107         expected_result = np.dot(matrix_A, matrix_B)
108         error_matrix = expected_result - result_compressed
```

```
109          print(f"Matrix size {size}x{size} -> Frobenius norm error: {frobenius_norm(error_matrix)
                 :.2f}")
110          start_time = time.time()
111          result_standard = standard_matrix_multiply(matrix_A, matrix_B)
112          elapsed_standard = time.time() - start_time
113          standard_times.append(elapsed_standard)
114
115          # Plotting
116          plt.figure(figsize=(10, 6))
117          ax = plt.gca()
118          ax.set_facecolor("#f0f0f0")
119          plt.plot(matrix_sizes, fast_times, color='blue', marker='o', label='Fast multiplication')
120          plt.plot(matrix_sizes, standard_times, color='red', marker='s', label='Standard
                 multiplication')
121          plt.xlabel('Time (seconds)')
122          plt.ylabel('Matrix size (n x n)')
123          plt.title('Non-Binary Matrix Multiplication Performance')
124          plt.grid(True, linestyle='--', linewidth=0.5)
125          plt.legend(loc='upper left')
126          plt.tight_layout()
127          plt.show()
128
129          print("fast_times:", fast_times)
130          print("standard_times:", standard_times)
```

Listing 2: Cardinality Sparsity for Non-Binary Matrix Multiplication

