# OpenReview forum: "Cardinality Sparsity: Applications in Matrix-Matrix Multiplications and Machine Learning"
_TMLR — Accepted by TMLR_

### Review · Reviewer_WTA4 · 2025-06-07

**Summary Of Contributions:**

This paper studies a parametrization of matrix or tensor which authors termed cardinality sparsity. Roughly, it measures the number of unique elements for each mode of the tensor. This is an intuitive measurement of sparsity, as when performing matrix multiplication, the number of necessary multiplications is captured by this parameter. The two main contributions of this paper are: 1. matrix multiplication algorithm utilizing cardinality sparsity, 2. a theoretical analysis on the prediction guarantee of neural network and tensor regression. The theoretical framework accounts for a class of regularizers including the standard $\ell_1$ regularizer and cardinality sparsity is a special case of them. Empirical evidence shows that matrix multiplication based on cardinality is faster than the standard algorithm and Strassen. It also improves the parameter count for training 2-layer ReLU network.

**Audience:**

Yes

**Broader Impact Concerns:**

This paper focuses on a new metric to measure the sparsity of matrix, tensor and more general machine learning models, it does not have any direct negative ethical implications.

**Claims And Evidence:**

Yes

**Requested Changes:**

Major:

1. See my comments in weakness regarding experiments.

2. I think the authors should try to reorganize and emphasize on the cardinality sparsity part for neural network and tensor regression part. Currently, it feels like the theory part is a bit redundant and not very coherent to the matrix multiplication part. I'm not super sure what's the best course of action here, maybe start with cardinality sparsity as a regularizer, then generalize to other settings instead of the other way around?

3. On the other hand, I feel it might be more coherent if authors could strengthen the results regarding matrix multiplication by providing some downstream applications that can be solved in cardinality sparsity time, such as regression, low-rank approximation and other numerical linear algebra tasks. One candidate would be conjugate gradient -- the algorithm is essentially matrix-vector multiplication, so I guess you could implement each iteration in cardinality sparsity time?


Minor:

1. The citation style mixed with not differentiating \citet and \citep makes reading a bit challenging, authors could either choose another citation style (say number, or alphabetical) that does not require differentiating between the two, or fix the mix use of \citet and \citep. Also many references miss years, making the already hard to parse mixing of citation even harder to read.

2. References for fast matrix multiplication are outdated, consider citing [Duan, Wu and Zhou, FOCS'23], [Williams, Xu, Xu and Zhou, SODA'24] and [Alman, Duan, Williams, Xu, Xu and Zhou, SODA'25].

3. Section 3, second line: "In order to sake completeness" should be "for the sake of completeness".

**Strengths And Weaknesses:**

Strengths:

1. Cardinality sparsity is an elementary but useful metric to capture the complexity of matrix / tensor operations. It is helpful to design algorithm that utilizes cardinality sparsity even beyond matrix multiplication. In particular, many numerical linear algebra algorithms have their runtime depend on the number of nonzero entries in a matrix / tensor. It would be interesting to design NLA algorithms when the matrix is dense but cardinality sparsity is small, and the runtime depends on the cardinality sparsity.

2. The theoretical analysis of neural network and tensor regression is actually more general than accounting for cardinality sparsity, it gives bound on a class of regularizers. It would be interesting to find more applications of the meta theorem obtained in this paper.

Weaknesses:

1. The matrix multiplication algorithm for cardinality sparsity is very elementary. When the inner dimension of the matrices is small, then one could simply perform outer product over unique elements, then map them back to corresponding entries. When the inner dimension is large, one then use the unique elements and index set of the rows and do standard inner product by avoiding unnecessary multiplications. Once the concept of cardinality sparsity is spelled out, these algorithms become quite obvious.

2. The theory regarding neural network and tensor regression is nice, but they are quite general with cardinality sparsity is just a special case. The theme of the paper shifts away quite significantly from cardinality sparsity when discussing this part. I do believe the results obtained here are interesting, but it might be worth thinking how to unify them better with cardinality sparsity.

3. Regarding experiments, while the results show that algorithm based on cardinality sparsity is better than standard algorithm or Strassen, I'm unsure about the practical implication here. Matrix multiplication is a highly optimized operation and modern GPU architectures are designed for them. If one compares the proposed cardinality sparsity based algorithm with matrix multiplication on GPU, it's unclear which one is faster, at least authors haven't specified on what platform the experiments are performed. Of course, it's not fair to compare a proposed algorithm with GPU-optimized matrix multiplication, so I think a fair comparison would be that, if one casts the cardinality sparsity based algorithm into standard matrix multiplication, how many matrix multiplications of what size matrices are needed. I do understand this might be challenging, as for cardinality sparsity based algorithm the matrix multiplications are for matrices of non-uniform size. I think, at least, the authors should clarify what platforms the experiments are performed, how do they implement the standard algorithm (just sequential algorithm that computes inner product entry-by-entry without any parallelization?), etc.

---

> ### Author Response · Authors · 2025-06-30
>
> We are truly grateful to the reviewer for their careful and constructive feedback. We deeply appreciate the time and effort dedicated to reviewing our work and are thankful for the valuable suggestions. Below, we offer our preliminary responses to the main issues raised:
>
> 1- We agree that once the concept of cardinality sparsity is applied, the resulting algorithms may appear straightforward. However, to the best of our knowledge, this idea has not been explicitly proposed in any setting—particularly in the binary case, where matrices contain only two distinct values and the notion of cardinality sparsity appears especially natural and straightforward.
>  We believe this simplicity is precisely what makes the method effective and powerful.
>
> 2 -We acknowledge that the shift in focus may appear abrupt, and we agree it would strengthen the coherence of the paper to better unify these results under the central theme of cardinality sparsity. In future version, we will clearly emphasize the connection between the general framework and its specialization to cardinality sparsity.
>
> 3 - Thank you for the valuable suggestion. We agree that demonstrating downstream applications can strengthen the impact of our results. As a step in this direction, we have already implemented the gradient descent algorithm using our proposed matrix multiplication method. This integration is one of the main reasons our network achieves higher accuracy in fewer iterations, as shown in Figure 8. We believe this highlights the practical potential of our approach in optimization tasks that rely heavily on matrix-matrix or matrix-vector multiplications.
>
> 4 - Thank you for the insightful comments. To ensure a fair and unbiased comparison, we implemented all three algorithms entirely in Python, using a sequential approach that computes matrix products entry-by-entry. We deliberately avoided using any optimized Python functions, and wrote all necessary functions from scratch, ensuring that no parallelization or external optimizations influenced the results.
>
> Since we could not attach the code with the submission, we are happy to share our implementation through the editor if requested.
>
> - Thank you for the thoughtful feedback. We understand the concern about coherence between the theoretical development and its connection to practical applications like neural networks and tensor regression. We are revisiting the structure of the paper to improve this alignment.
>
> Minor:
> Minor errors have been corrected in the updated version of the manuscript.

---

> > ### Comment · Reviewer_WTA4 · 2025-07-06
> > **Follow up questions on experiment**
> >
> > Thank authors for the response. Regarding the experiment, I think my main concern is that since you are comparing with a sequential $n^3$ matrix multiplication algorithm, it is in fact a less practical setting, as modern matrix multiplications are highly optimized to be parallelizable. This also applies to Strassen's algorithm -- if you compare it with a sequential $n^3$ matrix multiplication algorithm, it would be faster, however in practice, the $n^3$ algorithm is usually highly optimized and can utilize cache locality, while Strassen's algorithm and the cardinality sparsity algorithm proposed in this paper might not. Thus, my question on experiment is: would it be possible to reduce the cardinality sparsity algorithm to standard matrix multiplication on smaller-sized matrices? Or it is unclear how to do so at all? If it's the latter case, I think the practicality of the algorithm is significantly limited.

---

> > > ### Author Response · Authors · 2025-07-07
> > >
> > > Thank you for your thoughtful question regarding the practicality of the proposed algorithm. The core operation in our method is defined as:
> > >
> > > \[
> > > \mathbf{W} \times \mathbf{V} := \sum_{i=1}^{P} \mathbf{W}^{(i)} \otimes \mathbf{V}_{(i)},
> > > \]
> > >
> > > where $\otimes$ denotes the outer product, not standard matrix multiplication. Each term $\mathbf{W}^{(i)} \otimes \mathbf{V}_{(i)}$ captures a structured interaction between decomposed components of the input and weight tensors. The sum aggregates these interactions across $P$ components.
> > > In general, this operation cannot be directly reduced to a standard matrix multiplication unless additional structural assumptions are made. However, despite not being reducible to a single matrix multiplication, this operation remains practical to compute.
> > >
> > > Modern machine learning libraries such as PyTorch, TensorFlow, and JAX support efficient computation of such operations using Einstein summation notation ("einsum") and batched matrix operations. These implementations are GPU-accelerated and take advantage of parallel computation capabilities of modern hardware.
> > >
> > > Furthermore, when the matrix exhibits extreme sparsity in terms of cardinality (as in the binary case), the operation only needs to be performed on very small matrices—for example, multiplying $2 \times 2$ matrices ($A_{2\times 2} \times B_{2\times 2}$) in the binary setting. This can be handled efficiently in practice. Moreover, since the computation for each row and column index is independent of the others, further parallelization is possible. Specifically, the outer products for different row–column index pairs can be computed simultaneously.

---

### Review · Reviewer_JzCx · 2025-06-17

**Summary Of Contributions:**

This paper introduces a novel concept called "cardinality sparsity," which defines tensors as sparse when they contain only a small number of unique values, rather than the traditional notion of sparsity based on zero-valued entries. The authors at least to me make several contributions: (1) they formalize the mathematical definition of cardinality sparsity, (2) develop matrix multiplication algorithms that exploit this structure to achieve computational speedups, (3) establish statistical guarantees for regularizers that encourage cardinality sparsity in deep learning and tensor regression, and (4) demonstrate applications across multiple domains.

I think the core idea is intriguing and potentially valuable, but I have some concerns about both the theoretical claims and experimental validation that I'll detail below.

**Audience:**

Yes

**Claims And Evidence:**

No

**Requested Changes:**

Some changes, again please look at the weaknesses above, but reemphasizing these points.

1. Revise complexity analysis: Include preprocessing costs in the complexity analysis or clearly state when the "known structure" assumption is reasonable. For applications where matrices change frequently, the preprocessing overhead could dominate any computational savings
2. Strengthen experimental evaluation: Compare against modern state-of-the-art algorithms for Boolean matrix multiplication, not just classical methods.
3. Fix the memory claim analysis:  I feel like Table 2 memory comparisons are incomplete as you need to include storage costs for encoding matrices I and compressed matrices C_W, C_V and ccount for integer index overhead and pointer structures
4. Also could you include the pseudocode?
5. More ablations by varying cardinality degree, and ehavior with noisy data where values are approximately but not exactly repeated. Lastly would be good to do some performance degradation analysis as problem size scales!

Overall again I like this paper and direction: ) but more details would be great to include and more real world examples!

**Strengths And Weaknesses:**

Strengths: The concept of cardinality sparsity is genuinely novel and well-motivated as unlike traditional sparsity, which requires many zero values, this framework can capture structure in data where values repeat frequently but aren't necessarily zero - this seems practically relevant for many real-world datasets. The mathematical formulation is rigorous. The authors provide formal definitions for cardinality sparsity of tensors using mode-n fibers, and the connection to difference tensors provides a clean theoretical framework for analysis.

The paper demonstrates impressive breadth, spanning matrix algorithms, statistical learning theory, and practical applications. The statistical guarantees (Theorems 1 and 2) appear technically sound and show that cardinality sparsity can achieve similar benefits to traditional sparsity without forcing parameters toward zero. The experimental results show clear computational benefits, particularly for binary matrix multiplication where they claim substantial improvements over state-of-the-art methods.

Weaknesses:
1. The matrix multiplication algorithm has a critical flaw that the authors don't adequately address. They claim O(mPn) complexity, but the "decompression" step (reconstructing the full result from the compressed computation) actually requires examining every entry of the output matrix, which is inherently O(MN). This means their algorithm can't actually be faster than O(MN + mPn), and for many practical cases where m and n aren't dramatically smaller than M and N, this provides no advantage. Moreover, the Cartesian product operation in Algorithm 2 (I^(1) ×_c J_(1)) is computationally expensive. The authors claim it's "just concatenation" but it's actually a mapping operation that requires O(MN) work to properly reconstruct the result matrix, again negating the claimed speedup.

2. The assumption that matrix structure is "known" is not just strong—it's often impossible in practice at least to my knowledge. Consider:
2.1 In neural networks, while input data might have cardinality sparsity, weight matrices typically don't (and shouldn't, as this would severely limit expressiveness)
2.2 For iterative algorithms, matrices change at each step, requiring constant recomputation of the structure
2.3 The authors provide no analysis of how often this assumption holds in real applications

3. I guess even just a quick search on the internet, i see that the baselines are not very strong as they are now more specialized boolean matrix libraries that can achieve O(n³/(log n)^1.5) or better [https://www.cs.princeton.edu/~hy2/files/cbmm.pdf][Huacheng Yu,
An improved combinatorial algorithm for Boolean matrix multiplication,]. Also the authors they provide no analysis of when their method performs poorly or fails entirely.

4. How exactly is the "preprocessing" done efficiently? This is handwaved and there is no code availability: Despite claiming practical advantages, no implementation or at least pseudocode is provided.

---

> ### Author Response · Authors · 2025-06-30
>
> We are truly grateful to the reviewer for their careful and constructive feedback. We deeply appreciate the time and effort dedicated to reviewing our work and are thankful for the valuable suggestions. Below, we offer our preliminary responses to the main issues raised:
>
> 1- Thank you for the valuable suggestion. We have clarified the assumptions under which the structure of the matrices can be considered "known" and thus justify omitting preprocessing costs from the complexity analysis. In particular, this assumption is reasonable in scenarios where the same matrix structure is reused across multiple computations—for example, in iterative solvers, time-dependent simulations, or parameter studies—where the cost of preprocessing is amortized over many instances.
>
> We have also added a discussion highlighting that in contexts where the matrix structure changes frequently, the preprocessing cost may become significant and could outweigh the computational savings from the proposed method. This delineates the practical scope and limitations of the approach more transparently.
>
>
> 2- Thank you for the suggestion. We are currently comparing our approach with modern state-of-the-art algorithms for Boolean matrix multiplication, in addition to classical baselines. We will keep you updated on the results.
>
> 3- We are revising the memory comparison to include these factors and will provide a more complete accounting of all storage costs.
>
> 4- Thank you for the suggestion. We will include pseudocode that outlines the key steps of the preprocessing and multiplication procedures.
>
> 5- Thank you for these valuable suggestions. We are conducting additional ablation studies to address them in detail.
>
> -Strong assumptions:
> You are right that this assumption can be strong. However, in many real-world applications—such as inference in deep learning—the input data is often orders of magnitude larger than the weight matrices. For example, in batch inference or large-scale recommendation systems, input matrices can involve millions of rows (users, documents, tokens), while the weight matrices remain relatively small and fixed in size.
> Because of this imbalance, the majority of computational cost typically arises from the input side. Our method is designed to exploit structure in the larger matrix, and it remains effective even when only one of the matrices—typically the input—exhibits cardinality sparsity.
>
> -Failure of Algorithm. Thank you for pointing this out. We have added a discussion outlining the limitations of our method and the conditions under which it may perform poorly or fail to provide computational advantages.
>
> -How exactly is the "preprocessing" done efficiently?
> The implementation is provided in Python for column-wise preprocessing, which can be performed in a few lines of code; a similar approach applies to row-wise preprocessing. Please also refer to Example (3) in the manuscript for further illustration.
>
> def FirstBinaryMatrixEncoding(matrix):
>     rows = len(matrix)
>     cols = len(matrix[0])
>     result_matrix = [[0 for _ in range(cols)] for _ in range(rows)]
>
>     for col in range(cols):
>         complement = matrix[0][col] == 1
>         for row in range(rows):
>             val = matrix[row][col]
>             result_matrix[row][col] = (val + 1) % 2 if complement else val
>
>     return np.array(result_matrix)
>
> def FirstMatrixCompressed(matrix):
>     first_row = matrix[0]
>     output_array = [[val, (val + 1) % 2] for val in first_row]
>     return output_array

---

> > ### Author Response · Authors · 2025-07-02
> >
> > When the data is noisy and matrix elements are not exactly identical, smoothing and quantization techniques can be applied. In smoothing, we compute the average of matrix elements that are close in value to reduce local variations. These methods are particularly useful when the matrix has been distorted by noise, as they help recover the underlying structure.
> >
> > We conducted additional ablation experiments by varying the sparsity degree while keeping the matrix size fixed at 512 \times 80 and 80 \times 512. Specifically, we increased the cardinality (i.e., the number of unique elements per row or column) to observe its impact on computation speed. As expected, performance improved with higher degrees of sparsity, and at certain thresholds, our method began to outperform standard algorithms. Below are the detailed results and corresponding sparsity levels.
> >
> > CardinalityDegree = [5, 15, 25, 35, 45, 55, 65]
> >
> > Cardinality_time: [ 0.3540132   0.9883709   2.04295182  4.05787086  5.92038918  8.35499907
> >  11.22165561]
> > Standard_time: [7.5643239  7.63276601 8.00270319 8.10534382 7.64728236 7.75613809
> >  7.49451399]
> > Strassen_time: [6.91809511 6.837749   7.35869503 7.44682384 7.15954995 6.94609571
> >  6.95255804]
> >
> >
> > Moreover, we carried out more simulation experiments to evaluate our "binary multiplication algorithm" against some modern state-of-the-art methods. Note that many modern state-of-the-art algorithms are primarily theoretical, with no publicly available implementations. We selected two approaches that are more practical to implement—namely, the Four Russians algorithm and the bit-packed method—for direct comparison.
> > As shown in the results, our algorithm outperforms the Four Russians algorithm and achieves comparable performance to the bit-packed approach, which leverages CPU-level parallelism and low-level optimizations. This further demonstrates that our algorithm can significantly reduce CPU usage and hardware resource consumption. We applied all three algorithms to square binary matrices of the sizes listed below. The corresponding figure will be included in the paper.
> >
> >
> > matrix_size = [100, 200,300,400,500,600,700,800, 900]
> >
> > Cardinality_time = [0.02207589 0.11523795 0.26267171 0.56064892 1.03436804 1.7417438 2.56917286 4.32040691 5.24331379]
> >
> > Four_Russian_time = [ 0.01937675  0.14109969  0.57656574  1.10113621  1.90309811  4.85418701 7.17950892  9.595541   12.85125804]
> >
> > Bit_Packed_time = [0.01017118 0.0694809  0.22153807 0.51106787 0.97233605 1.6571312 2.66781902 3.86751676 6.30065513]

---

> ### Comment · Action_Editor_nMFq · 2025-07-07
> **Evaluate the response and revision**
>
> Dear Reviewer,
>
> Thank you for submitting your review for the TMLR submission *Rectified Robust Policy Optimization for Robust Constrained Reinforcement Learning without Strong Duality.* The authors have provided a response and submitted a revised version of the paper.
>
> Please take the time to carefully read their response and revision, and update your review accordingly to reflect whether your concerns have been addressed.
>
> Thank you again for your contribution to the review process.
>
> Best,
>
> Pan

---

> > ### Comment · Reviewer_JzCx · 2025-07-16
> > **Response to Authors**
> >
> > I appreciate that the authors have made substantial improvements addressing several of my core concerns. They've provided the clarification I requested on the "known structure" assumption, added comparisons with modern algorithms (Four Russians, bit-packed methods), provided actual Python implementation code, conducted the ablation studies I asked for varying cardinality degree, addressed noisy data handling, and included multiple real-world experiments. Overall, they've done a lot of the work I called for in my original review.
> >
> > However, the authors still haven't addressed my most critical technical objection - the fundamental complexity flaw where the "decompression" step requires O(MN) operations, potentially negating claimed speedups. They don't rigorously analyze the Cartesian product computational expense I highlighted or provide the complete memory analysis including encoding matrix storage costs that I requested. If the authors can provide a bit of clarification on that, it would be great! Its fine if not, overall this paper is a good work in this area!

---

### Review · Reviewer_Y4Zd · 2025-06-23

**Summary Of Contributions:**

This paper introduces a novel concept of cardinality sparsity, defining sparsity in terms of the number of unique values in a tensor, rather than the number of non-zero elements. The authors demonstrate that this formulation enables both statistical and computational advantages in various machine learning applications, including deep learning and tensor regression.

**Audience:**

Yes

**Broader Impact Concerns:**

I did not identify any significant ethical concerns in the current submission. The broader impact of this work appears positive: it enables more efficient computation and memory usage, which is desirable for environmentally friendly machine learning. There is no indication of misuse or safety issues given the scope of the methods.

**Claims And Evidence:**

Yes

**Requested Changes:**

Major:
1. Please improve the readability of the theoretical sections by simplifying the exposition, introducing more intuitive explanations, and reducing notational complexity where feasible.
2. While the paper is strong overall, the empirical validation in Section 6.3.1 remains relatively limited. The authors are encouraged to strengthen this section by evaluating additional aspects such as varying neural network architectures, different training dataset sizes, and broader performance metrics beyond training accuracy—e.g., test accuracy, generalization performance, stability, and sensitivity to regularization. A richer set of experiments would substantially enhance the credibility of the proposed sparsity approach for deep learning applications.

Minor:
3. The citation formatting contains inconsistencies. For example, the citation “Bowen and Wang” is missing the publication year.

4.  Section 3, first paragraph, line 3, there is a missing space between “sparsity.” and “This”.

**Strengths And Weaknesses:**

++:
1.  The concept of cardinality sparsity is original and meaningful, with clear motivation and distinct advantages over traditional definitions of sparsity.
2. The paper extends previous statistical frameworks to accommodate the new notion of sparsity, particularly through generalizations in DNNs and tensor regression.
3.  The results on accelerating matrix-matrix multiplications are impressive and practical.
4. : The empirical section is comprehensive, covering use cases in deep learning, matrix factorization, and classification tasks.
5. The approach could benefit low-resource machine learning and efficient model deployment, particularly in edge computing or embedded systems.

---:
1. While mathematically sound, the theoretical sections are somewhat dense and could benefit from clearer exposition, more intuitive explanation, and illustrative examples.
2. The manuscript could better emphasize how cardinality—as opposed to classical sparsity—drives performance gains theoretically. While this is implicit, a more explicit contrast would help readers appreciate the contribution more directly.

---

> ### Author Response · Authors · 2025-06-30
>
> We are truly grateful to the reviewer for their careful and constructive feedback. We deeply appreciate the time and effort dedicated to reviewing our work and are thankful for the valuable suggestions. Below, we offer our preliminary responses to the main issues raised:
>
> 1- Theoretical Sections: We also thank the reviewer for pointing out the need for greater clarity in the theoretical discussion. To address this, we have revised the corresponding sections to improve readability by simplifying the narrative and including more intuitive explanations.
>
> 2 - Thank you for the constructive feedback. Empirical Validation (Section 6.3.1): We are expanding Section 6.3.1 to incorporate additional experiments that explore further aspects of neural networks, including different architectures, varying dataset sizes, and broader evaluation metrics such as test accuracy, generalization behavior, and sensitivity to regularization. We will keep you updated on the results.
>
> 3 -Minor Issues: We have corrected the citation formatting issues, including adding the missing publication year for “Bowen and Wang.”
> The typographical error in Section 3 (missing space between “sparsity.” and “This”) has been fixed.

---

> > ### Author Response · Authors · 2025-07-07
> >
> > To further evaluate the effectiveness and generalizability of our approach, we conducted additional simulations using newly curated datasets and a different neural network architecture. These experiments yielded promising results, reinforcing the robustness of our method across varying settings. We plan to include these findings, in the revised version of the paper.

---

> ### Comment · Action_Editor_nMFq · 2025-07-07
> **Evaluate the response and revision**
>
> Dear Reviewer,
>
> Thank you for submitting your review for the TMLR submission *Rectified Robust Policy Optimization for Robust Constrained Reinforcement Learning without Strong Duality.* The authors have provided a response and submitted a revised version of the paper.
>
> Please take the time to carefully read their response and revision, and update your review accordingly to reflect whether your concerns have been addressed.
>
> Thank you again for your contribution to the review process.
>
> Best,
>
> Pan

---

### Decision · Action_Editor_nMFq · 2025-07-29

**Recommendation:** Accept with minor revision

**Additional Comments:**

This paper proposes a novel sparsity concept based on the number of unique values in each column of a matrix or in tensor slices. The authors formalize the mathematical definition of cardinality sparsity, develop matrix multiplication algorithms that exploit this property for computational speedup, establish statistical guarantees for regularizers that encourage cardinality sparsity in deep learning and tensor regression, and demonstrate applications across multiple domains. It is unanimously agreed that cardinality sparsity is an important extension, particularly for problems where inputs are discrete or where algorithms employ repeated patterns with a fixed weight matrix.

While the concept is novel and the authors present promising applications, several minor concerns remain unaddressed in the current version. The authors’ response regarding the practical relevance of the assumed known matrix structure is not fully convincing. More practical examples should be provided to justify this structural assumption. Another issue arises with the regularization-based optimization: cardinality-based sparsity penalties, which directly penalize the number of unique values, lead to non-convex, combinatorial optimization problems. This should be clearly addressed, either from a practical or theoretical perspective. In addition, the theoretical analyses for the deep learning and tensor regression sections seem somewhat disconnected from the main goals of the paper. The theoretical results are mostly general and resemble those in existing literature; more specific and tailored theory or discussion about the proposed cardinality sparsity and related regularization terms would be desirable. There is a typo on Page 4: a punctuation error in “We can also define cardinality sparsity for vectors. Which actually are special cases of tensors.”

Overall, this paper presents a highly novel idea of cardinality sparsity and demonstrates its potential for impactful applications. I recommend a minor revision to address the comments above before a final decision is made.

**Audience:**

Yes

**Audience Explanation:**

Yes, the cardinality sparsity concept is well motivated in important machine learning applications including matrix multiplication, deep learning, and tensor regression.

**Claims And Evidence:**

Yes

**Claims Explanation:**

Yes, this paper provides illustrations by example and potential applications of the proposed cardinality sparsity concept.

---

> ### Author Response · Authors · 2025-08-18
>
> Thank you for your valuable feedback. We have carefully revised the manuscript in response to your comments:
>
> 1- Additional simulation results: We added a new section (Section 6.4) in the simulation study to demonstrate applications of cardinality sparsity beyond machine learning. This section provides further examples that justify and illustrate the usefulness of the proposed structural assumption.
>
> 2- Clarification of the cardinality regularizer: Section 4.2 (page 10) has been revised to give a more thorough explanation of the cardinality regularizer, including its construction and motivation.
>
> 3- Convexity of the proposed regularizer: As noted in Proposition 1, the proposed formulation defines a norm, and therefore it is convex. This ensures that our regularizer remains convex, unlike direct cardinality penalties which are non-convex and combinatorial in nature. Our approach leverages norm regularization to induce sparsity in a tractable way, avoiding the computational difficulties of combinatorial optimization. Section 4.2.1 has also been revised accordingly to clarify this point.
>
> 4- Minor corrections: We corrected the punctuation error you noted.

---

> > ### Comment · Action_Editor_nMFq · 2025-08-28
> > **Some format issues**
> >
> > 1) Please use \citet{} and \citep{} correctly. Refer to Section 4.1 of TMLR stylefile: https://github.com/JmlrOrg/tmlr-style-file/blob/main/main-accepted.pdf
> >
> > 2) Table captions should be above the table content. Refer to Section 4.4 of the stylefile.

---

> > > ### Author Response · Authors · 2025-08-28
> > >
> > > Thank you very much for your valuable feedback. We have revised the manuscript.
> > >
> > > 1- \citet{} replaced with \citep
> > >
> > > 2- Table captions is now above the table content.